# Competition Dynamics Shape Algorithmic Phases of In-Context Learning

**Core Francisco Park**[1,2,3,*]**, Ekdeep Singh Lubana**[1,3,*]**, Itamar Pres**[4]**, Hidenori Tanaka**[1,3]

[1]CBS-NTT Program in Physics of Intelligence, Harvard University
[2]Department of Physics, Harvard University
[3]Physics & Informatics Laboratories, NTT Research, Inc.
[4]EECS Department, University of Michigan, Ann Arbor

## ABSTRACT

In-Context Learning (ICL) has significantly expanded the general-purpose nature of large language models, allowing them to adapt to novel tasks using merely the inputted context. This has motivated a series of papers that analyze tractable synthetic domains and postulate precise mechanisms that may underlie ICL. However, the use of relatively distinct setups that often lack a sequence modeling nature to them makes it unclear how general the reported insights from such studies are. Motivated by this, we propose a synthetic sequence modeling task that involves learning to simulate a finite mixture of Markov chains. As we show, models trained on this task reproduce most well-known results on ICL, hence offering a unified setting for studying the concept. Building on this setup, we demonstrate we can explain a model's behavior by decomposing it into four broad *algorithms* that combine a fuzzy retrieval vs. inference approach with either unigram or bigram statistics of the context. These algorithms engage in a competition dynamics to dominate model behavior, with the precise experimental conditions dictating which algorithm ends up superseding others: e.g., we find merely varying context size or amount of training yields (at times sharp) transitions between which algorithm dictates the model behavior, revealing a mechanism that explains the transient nature of ICL. In this sense, we argue ICL is best thought of as a mixture of different algorithms, each with its own peculiarities, instead of a monolithic capability. This also implies that making general claims about ICL that hold universally across all settings may be infeasible.

## 1 INTRODUCTION

In-Context Learning (ICL)—the ability to perform novel tasks by merely using the inputted context—has substantially expanded the general-purpose nature of large language models (LLMs) (Brown et al., 2020; Wei et al., 2022), allowing them to solve a broader spectrum of problems than they may have been initially trained for (Gemini Team, 2023; Qin et al., 2023; Huang et al., 2022; Bai et al., 2022). To better understand the mechanisms underlying ICL, a series of papers have designed toy, synthetic domains that are amenable to rapid experimentation and can offer precise hypotheses into how this capability operates. This line of work has established a rich phenomenology, demonstrating, e.g., the importance of specialized attention heads (aka induction heads) (Olsson et al., 2022; Edelman et al., 2024; Singh et al., 2024), change in ICL abilities as a function of data diversity (Raventós et al., 2023; Lu et al., 2024; Kirsch et al., 2022), the non-monotonic trend in test performance as context is increased (Min et al., 2022; Lin & Lee, 2024), and the transient nature of ICL with training time (Singh et al., 2023; Anand et al., 2024).

Despite the substantial progress highlighted above, we note a unified account of ICL is still lacking. This can be partially attributed to the fact that prior work often focuses on rather disparate setups to develop its findings—e.g., linear regression (Garg et al., 2023; Von Oswald et al., 2023; Akyürek et al., 2023; Bai et al., 2024), classification (Chan et al., 2022a;b; Reddy, 2023; Singh et al., 2023; 2024), and probabilistic automata (Akyürek et al., 2024; Edelman et al., 2024; Bigelow et al.,

---

*Equal contribution. Contact: `corefranciscopark@g.harvard.edu`, {`ekdeeplubana`, `hidenori_tanaka`}`@fas.harvard.edu`, `presi@umich.edu`.

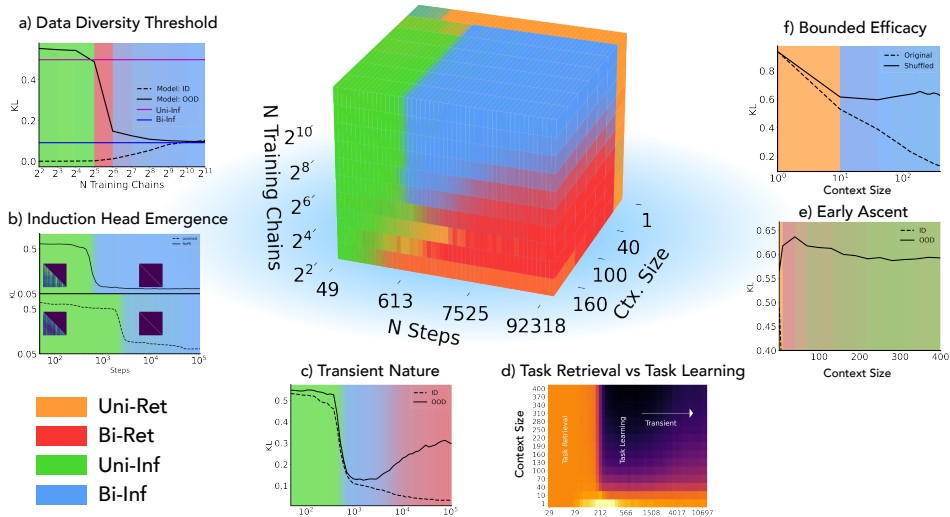

Figure 1: **Algorithmic phase diagram for a finite Markov mixtures task.** We propose to study ICL phenomena through a minimal experimental system: Transformers trained on sequence data generated by a finite mixture of Markov chains. This setup turns out to be extremely rich, capturing most (if not all) known phenomenology of ICL, but still being simple enough to be amenable to theoretical modeling. We identify four distinct, interpretable algorithmic solutions and characterize the transitions between them as functions of data diversity, optimization steps, and context size— labeled **algorithmic phases**. Considering the corresponding phase diagram (middle panel), we find part of the rich phenomenology of ICL emerges from competing algorithmic strategies promoted or suppressed by broader experimental configurations. Specifically, our framework captures an array of known phenomena: a) Data diversity threshold (Raventós et al., 2023); b) Emergence of induction heads (Edelman et al., 2024); c) Transient nature (Singh et al., 2023); d) Task retrieval and task learning phases (Min et al., 2022); e) Early ascent of risk (Xie et al., 2021); and f) Bounded efficacy (Lin & Lee, 2024). See App. C for a concise summary of these findings and propositions.

2023)—leaving it unclear precisely which ICL phenomena are universal, important, and worth investigating to develop a unified theory. To address this issue, we argue a novel experimental setup is needed that is rich enough to capture most (if not all) known phenomenology of ICL, but is also simple enough to be amenable to modeling, hence offering fertile ground for a unifying account. We aim to fill this gap in the current paper.

**This work.** We propose a novel sequence modeling task that involves learning to simulate a finite mixture of Markov chains. We train Transformers on this task via a standard autoregressive loss under different amounts of compute budget (training iterations and model size) and data diversity (number of Markov chains in the mixture), while evaluating them with different amounts of context seen at inference. Systematically varying these factors, we show our proposed task turns out to be extremely rich, *reproducing most known phenomenology of ICL and hence offering a unified setup for studying the concept.* Building on this, we start to deconstruct how a model trained on our task performs ICL, finding that there exist (at least) four broad algorithms that can explain the model's behavior. These algorithms combine a fuzzy retrieval vs. inference approach with either unigram or bigram statistics of the context, and, as we show, engage in a *competition dynamics* with each other to dictate model behavior. Interestingly, we find the precise experimental conditions (e.g., amount of training and context-size) decide which algorithm wins the competition, hence yielding several *phases* in the model's ability to perform ICL: varying experimental conditions elicits different algorithmic behaviors (at times rather abruptly), making it difficult for understanding of ICL derived in one configuration to help predict model behavior in another one. This picture also helps us better understand several existing phenomena of ICL, e.g., why it can be transient in nature, hence enabling a step towards a unified account. Our contributions follow.

- **A finite Markov mixtures task captures ICL's phenomenology.** We introduce a synthetic sequence modeling task wherein a model is trained to simulate a finite mixture of Markov chains (Sec. 2). As we show, models trained on this task reproduce most (if not all) known phenomenology of ICL (see Fig. 1, 3), hence offering a unified, controlled setting for studying the concept.

Figure 2: **Data generation and evaluation protocol with finite Markov mixtures.** (a) **Data generation.** We first sample a finite set $\mathcal{T}_{\text{train}} = \{T_1, T_2, \ldots, T_N\}$ of random transition matrices to define our set of Markov chains. We then randomly select a chain from this set and sample a training sequence from it. We repeat this process at every step of training, sampling a fresh batch of sequences from by randomly selecting a chain from our predefined set. (b) **Model training.** We train a Transformer (Karpathy, 2022) on this sequence data with a standard autoregressive training loss. (c) **Evaluation.** A novel sequence of states is sampled from the test transition matrix, $T^*$, for evaluation. Here, $T^*$ is either (i) selected from the finite set $\mathcal{T}_{\text{train}}$ (for in-distribution tests), or (ii) newly sampled (for OOD tests). We subsequently compute the KL divergence between the model's empirical transition matrix $\hat{T}$ vs. ground truth transition matrix $T^*$. See App. A.1 for details.

- **Systematic experiments identify different algorithmic phases of ICL.** By varying the amount of training steps, data diversity (number of chains), and context size in a systematic manner, we find a model trained on our proposed task transitions between (predominantly) four *phases* of algorithmic solutions that are characterized by use of *unigram vs. bigram* context statistics in a *fuzzy retrieval vs. inference* manner (Fig. 4, 5). Furthermore, the scale of the model interacts with the boundaries of these phases, e.g., by shifting the critical amount of data diversity needed for transitioning between different algorithms (Fig. 8). These results indicate ICL is best regarded as an umbrella term for a spectrum of algorithms, instead of a monolithic model capability, and any identified phenomenology of ICL should *not* be deemed universal unless shown otherwise.

- **A competition of algorithms picture underlies ICL's phenomenology.** To further develop a precise understanding of our identified algorithmic phase diagram, we decompose our model's behavior at any given time into a linear interpolation of the four algorithms mentioned above (Fig. 6). This interpolation turns out to be surprisingly accurate, achieving approximately zero KL with respect to the trained model's next token probabilities, and thus suggesting that different algorithms compete with each other to dictate model behavior. We then show these competition dynamics can explain part of ICL's phenomenology, e.g., offering an explanation for the transient nature of ICL (Singh et al., 2023; Anand et al., 2024) and its non-monotonic out-of-distribution performance curves (Kirsch et al., 2022) (Fig. 7).

## 2    PROBLEM SETUP: SIMULATING A FINITE MIXTURE OF MARKOV CHAINS

We begin by proposing a task that unifies (most) known phenomenology of ICL into a singular setup: *learning to simulate a finite mixture of Markov chains*. This task captures the sequence modeling nature of LLMs by applying a stochastic map to every token (similar to the probabilistic automata setting of Akyürek et al. (2024)), while also offering a knob (number of chains) that helps assess the impact of data diversity on ICL (similar to the linear regression setting of Raventós et al. (2023)).

**Sequence modeling with finite mixture of Markov chains.** As illustrated in Fig. 2, our proposed task involves modeling of a predefined set of $N$ Markov chains ($N \in \{2^2, 2^3, \ldots, 2^{11}\}$). A chain has a unique Transition matrix $T_n \in \mathbb{R}^{k \times k}$ associated with it, where $k$ denotes the number of states ($k = 10$, unless noted otherwise). Each row of $T_n$ is sampled from a Dirichlet distribution, with $T_{[i,j]}$ denoting the $(i,j)^{\text{th}}$ element of the transition matrix, i.e., the probability of transition from state $i$ to state $j$. The overall set of transition matrices is denoted $\mathcal{T}_{\text{train}} = \{T_1, T_2, \ldots, T_{N-1}, T_N\}$. The data-generating process (DGP) involves first randomly selecting a matrix $T_n \in \mathcal{T}_{\text{train}}$ following a prior $\mathbf{p} \in \mathbb{R}^N$, defining a Markov chain using $T_n$, and then sampling a sequence of length $l = 512$ from it. Overall, we note the DGP is characterized by three key hyperparameters: (i) $N$: the number of chains (a measure of data diversity in this work); (ii) $k$: the number of states; and (iii) $l$: the sequence length. See App. A.1 for further experimental details.

**Model Training.** We train a 2-layer Transformer (Karpathy, 2022) on sequences sampled from the above DGP via the standard, autoregressive sequence modeling objective (see App. A.2 for model

Figure 3: **Finite Markov mixture setup captures rich phenomenology of in-context learning (ICL).** (a) KL divergence (OOD evaluation) as a function of training steps and data diversity (Number of Training Chains). (b) As the data diversity of the training data is increased (see ruby vertical dashed line in panel (a)), we reproduce the data diversity threshold for "task learning" ICL, similar to Raventós et al. (2023); Kirsch et al. (2022). (c) At high task-diversity regime with $N = 2^7$ (see green horizontal dashed line in panel (a)), we reproduce non-monotonic performance dynamics in a sequence modeling setup. This phenomenon was previously reported as "transient nature of ICL" in Singh et al. (2023). See App. F.1 for more plots from these experiments.

details and hyperparameters). The sampling process is repeated every step of training, i.e., the model is unlikely to see the same sequence twice during training (a.k.a. online training).

**Evaluation.** We evaluate trained models on In-Distribution (ID) and Out-Of-Distribution (OOD) chains. For ID evaluations, we select the transition matrix $T^*$ from the training set $\mathcal{T}_{\text{train}}$ and for OOD evaluations, we sample a novel $T^*$ using a Dirichlet prior. We then draw many sample sequences from these transition matrices and compute the average KL divergence between the model's predicted transition matrix $\hat{T}$ and $T^*$. Formally, we compute:

$$\left\langle KL(\hat{T}\|T^*) \right\rangle = \left\langle \sum_i \pi^*_{[i]} \sum_{j=1}^k \hat{T}_{[i,j]} \log \frac{\hat{T}_{[i,j]}}{T^*_{[i,j]}} \right\rangle. \tag{1}$$

See Appendix A.3 for further detail about evaluation.

## 2.1 REPRODUCING ICL'S PHENOMENOLOGY

We next demonstrate our proposed task reproduces several known results on ICL, yielding fertile ground for developing a unified account of this capability. While in the main paper we present only a few salient phenomena that are of interest to our discussion later, we refer the reader to Fig. 1 and App. C for a more comprehensive list of captured phenomenology.

**Transition via Data Diversity.** In Fig. 3 (a), we present a heatmap of the KL divergence between the model's predicted transition matrix ($\hat{T}$) and an OOD chain's transition matrix ($T^*$) as a function of training steps (x-axis) and the number of Markov chains used to generate the training dataset (y-axis) (See App. F.1 for ID evaluation). Similar to Raventós et al. (2023), who argue the model *transitions from a "Bayesian averaging" approach to an "in-context learning" one* as the amount of data diversity is increased, we find that given sufficient training steps, there is a sudden drop in KL on OOD evaluations: when diversity is low, we find the model performs well on ID chains but poorly on OOD chains; meanwhile, when diversity is high, we find the model performs well on OOD chains as well. This phenomenon is explicitly shown in Fig. 3 (b), where we show the KL of ID and OOD chains at 839 steps of training for different data diversity values. As data diversity increases, the ID KL slightly increases since the task gets relatively more complex. The OOD KL drops slowly with data diversity until $N = 2^6$, where we see an abrupt decrease and for $n > 2^6$ there is nearly no gap between the ID and OOD performance.

**Transient nature of ICL.** We first highlight that, given enough data diversity ($n > 2^6$), there is always an **emergence of induction heads** once a critical number of training steps is reached, similar to Edelman et al. (2024) (see Fig. 1 (b)). Fig. 3 (c) shows KL as a function of training steps at a high data diversity ($N = 2^7$). Again, we observe the emergence of the induction head dropping both the KL of ID and OOD chains at $\sim 6 \times 10^2$ steps. *Strikingly, after this drop, the KL divergence begins to increase again, but only for OOD chains.* As we show later, this behavior corresponds to the *transient nature of ICL* proposed by Singh et al. (2023): the model transitions back from

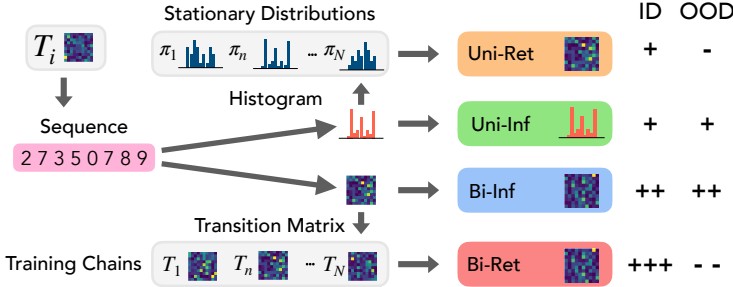

Figure 4: **Proposed algorithms for the finite Markov mixture task.** (a) Unigram based Retrieval (`Uni-Ret`): Given a sequence, `Uni-Ret` involves computing a histogram of token frequencies in the sequence and then creating a new transition matrix that is a weighted average of chains in $\mathcal{T}_{\text{train}}$. Weight associated to a chain is based on the distance between the computed histogram and a chain's steady-state distribution. (b) Bigram based Retrieval (`Bi-Ret`): Similar to `Uni-Ret`, but uses observed *transitions*, i.e., bigrams, to weight the chains. The resulting likelihood is much sharper, making this algorithm better *for the training data*. (c) Unigram Inference (`Uni-Inf`): This algorithm infers a histogram from the given context and draws subsequent tokens from this histogram directly. (d) Bigram Inference (`Bi-Inf`): This algorithm infers the *transition matrix* from the given context and draws subsequent tokens from this transition matrix directly. This approach achieves best OOD generalization among considered algorithms. The $+$ and $-$ indicate the performance expected on ID chains and OOD chains, where a $+$ indicates better performance.

using an algorithm that performs well OOD to one that performs well solely ID; the latter relies on memorized information that is akin to what the authors in their paper call an "in-weights" solution.

## 3 ALGORITHMIC PHASES IN FINITE MIXTURE OF MARKOV CHAINS

Having ascertained the value of our proposed task by reproducing known phenomenology, we now aim to take a step towards developing a unified account of ICL. To this end, we must better understand how a model trained on our task performs ICL. As we show, we can identify (at least) *four broad algorithms* that explain the trained model's behavior (i.e., its next token predictions) for different subsets of experimental configurations. We call these subsets **algorithmic phases**: continuous ranges of experimental configurations where the trained model's behavior is explainable by a predefined algorithm. Building on this analysis, we show in the next section that ICL's phenomenology, as it manifests in our setup, is partly driven by a competition between our identified algorithms. We also offer preliminary mechanistic evidence for our analysis in App. E.

### 3.1 ALGORITHMS TO SIMULATE FINITE MIXTURE OF MARKOV CHAINS

Broadly, the axes that help characterize our algorithms (see Fig. 4) are (i) what *statistics of the inputted context* are used by the model (unigram vs. bigram), and (ii) whether the approach involves a fuzzy *retrieval* of the most relevant Markov chains seen during training to make the next-token prediction (akin to a Bayesian averaging operation where a discrete prior is defined over $\mathcal{T}_{\text{train}}$, i.e., chains seen during training), versus an *inference* of the Markov chain parameters based solely on the sequence seen in context (akin to a Bayesian averaging operation where a continuous prior, i.e., the Dirichlet distribution is used)[1]. A retrieval approach will generally achieve better performance on ID evaluations; however, its performance on OOD evaluations will be worse, especially with increased context length (see App. D for a detailed discussion). Below, we use $\pi_n$ to denote the stationary distribution of a chain $T_n$ and $\delta$ is the Kronecker delta.

**Retrieval Approach.** Similar to *task-retrieval* notions of ICL (Min et al., 2022), we define "retrieval" algorithms as the accumulation of some relevant statistics of the input sequence to compute

---

[1]We note we primarily use distinct names for the two approaches for clarity, but both approaches are in fact Bayesian inference protocols with priors that depend vs. not on $\mathcal{T}_{\text{train}}$. See App. B.1 for further discussion.

a likelihood function that depends on the Markov chains underlying our training data, i.e., $\mathcal{T}_{\text{train}}$. Specifically, the algorithm utilizing unigram statistics of the input, which we call `Uni-Ret` (Unigrams based Retrieval), uses the following likelihood function.

$$\text{Unigram Likelihood: } \mathcal{L}_U(T_n|\mathbf{x}_{1:t}) = \Pi_{j=1}^t \pi_{n[x_j]}, \tag{2}$$

where $\mathbf{x}_{1:t}$ is the sequence of all states 1 to $t$. Meanwhile, the algorithm utilizing bigram statistics of the input, which we call `Bi-Ret` (Bigrams based Retrieval), uses the following likelihood function.

$$\text{Bigram Likelihood: } \mathcal{L}_B(T_n|\mathbf{x}_{1:t}) = \Pi_{j=1}^t T_{n[x_{j-1}, x_j]}. \tag{3}$$

Given the likelihood functions above, the posterior predictive distribution to predict how likely a given next state is can be computed as follows.

$$\text{Retrieval approach: } p(x_t|\mathbf{x}_{1:t-1}) \propto \sum_n p_n \, \mathcal{L}(T_n|\mathbf{x}_{1:t-1}) \, T_{n[x_{t-1}, x_t]}. \tag{4}$$

**Inference Approaches.**  Similar to "task-learning" notions of ICL (Raventós et al., 2023; Lu et al., 2024), we define "inference" algorithms as the computation of relevant statistics from the inputted sequence to infer a probability distribution over the next feasible states. The precise Markov chains seen during training *play no role in this computation* (unlike the fuzzy retrieval approaches discussed above). Consequently, these algorithms exhibit no performance disparity between ID and OOD evaluations, as they do not incorporate any information from the training dataset. Formally, one uses either the frequency of token occurrences, i.e., the unigram distribution, or the frequency of pairwise token occurrences, i.e., the bigram distribution, to define a transition matrix that encodes the predicted next-token probabilities. We call the former algorithm `Uni-Inf` and the latter `Bi-Inf`, denoting their transition matrices $T^U$ and $T^B$ respectively.

$$\texttt{Uni-Inf:} \ T_{[i,j]}^U(\mathbf{x}_t) = \frac{\sum_{k=1}^t \delta_{x_k, j}}{t}; \tag{5}$$

$$\texttt{Bi-Inf:} \ T_{[i,j]}^B(\mathbf{x}_{1:t}) = \frac{1 + \sum_{k=1}^{t-1} \delta_{x_k, i} \delta_{x_{k+1}, j}}{k + \sum_{k=1}^{t-1} \delta_{x_k, i}}. \tag{6}$$

## 3.2 Isolating Algorithmic Phases

We now demonstrate the four algorithms proposed above delineate models trained on our task into broad **algorithmic phases** based on the train / test configuration. To this end, we define the following two evaluations protocols that assess whether the model utilizes bigram statistics and whether it follows a fuzzy retrieval approach, i.e., relies on the chains seen during training.

**Assessing Bigram Utilization.**  We quantify a model's reliance on bigram statistics of the sequence shown in context by exploiting a key difference between our proposed algorithms: unigram-based methods depend solely on steady-state distributions ($\pi^*$), while bigram-based methods consider state transitions ($T^*$). Thus, to distinguish between these approaches, we can simply shuffle the positions of all tokens in the input sequence. This perturbation preserves the stationary distribution, but disrupts any order-sensitive information, e.g., information about bigram transitions. We can then measure change in KL between the empirical transition matrix inferred from the model's predicted next-token distributions and the ground truth matrix used for sampling the sequence: a large change would suggest the model relies on bigram statistics to perform the task, while a small change would indicate a unigram-based approach is at play. See App. A.3.3 for implementation details.

**Proximity to a Retrieval Approach.**  Assessing how much a model relies on the Markov chains seen during training, e.g., by internalizing their transition matrices (see also App. E.1), helps distinguish between solutions that *solely* leverage context statistics (`Bi-Inf` and `Uni-Inf`) and those that do not (`Bi-Ret` and `Uni-Ret`). Motivated by this, we first sample a *new* set of transition matrices, denoted $\mathcal{T}_{\text{random}}$, with the same number of matrices as the train set, $\mathcal{T}_{\text{train}}$. We then define a chain using a transition matrix $T^*$ that *does not belong* to either $\mathcal{T}_{\text{train}}$ or $\mathcal{T}_{\text{random}}$. Using a sequence sampled from this chain, we compute the empirical transition matrix $\hat{T}$ based on the model's next token predictions, and then check whether this matrix is closer (in terms of KL) to the set $\mathcal{T}_{\text{train}}$ or to the set $\mathcal{T}_{\text{random}}$. If the model does not have a preference for the seen transition matrices, i.e., it is not utilizing a retrieval approach, then $\hat{T}$ should be (approximately) equally close to the two sets; else, it should be closer to $\mathcal{T}_{\text{train}}$. See App. A.3.3 for further discussion and implementation details.

Figure 5: **Algorithmic phases.** (a) **Bigram Utilization**: We shuffle the order of all states in a sequence and measure the KL (App. Eq. 9) before and after the perturbation to quantify the bigram utilization of a model. The shuffling should only affect algorithms sensitive to higher-order statistics. (b) **Proximity to Retrieval**: A model is labeled "closer" to a *retrieval* approach when its next-token probabilities are closer to matrices seen in the training set. We evaluate this by sampling an unseen set of transition matrices, and measuring if the model's next-token probabilities have a lower KL w.r.t. transition matrices seen in training or if it is similar to the freshly sampled set (App. Eq. 11). (c) **Algorithmic Phases**: The product of bigram utilization and proximity to retrieval scores delineates four distinct algorithmic phases. (d) **Validating phases**: KL between model's and predefined algorithms' next-token probabilities provides validation to our identified phase diagram.

**Results.** See Fig. 5. We find the evaluation protocols defined above clearly delineate experimental configurations into regions where the solution is (i) unigram-dependent vs. bigram-dependent, and (ii) closer to retrieval vs. inference (Figs. 5 (a,b)). These results divide Fig. 3 into four distinct phases, each in accordance with the four algorithms proposed in Sec. 3.1: Uni-Ret, Bi-Ret, Uni-Inf, and Bi-Inf (see Fig. 5 (c)). We confirm the validity of these phases by comparing KL between the model and the predefined algorithms' next-token probabilities (Fig. 5 (d)). Importantly, we observe that with enough training steps and data-diversity, bigram dependence consistently emerges. Meanwhile, if the data-diversity is large (small), the model is closer to an inference (retrieval) approach. Medium data-diversity however sees an interesting learning dynamic, wherein the model starts off with a retrieval approach, transitions to an inference approach with enough training, but then *slowly rolls back to a retrieval approach!* We also perform several other experiments to corroborate these findings, such as providing preliminary mechanistic evidence for these algorithms; e.g., *we find we can reconstruct transition matrices from $\mathcal{T}_{train}$ via MLP neurons in retrieval phases! (see App. E).* We also report additional metrics and attention analysis in App. F.

Overall, *we conclude there are (at least) four* **algorithmic phases** *in the dynamics of learning to simulate finite mixture of Markov chains:* a model uses (predominantly) one of the four algorithms identified above to perform our task, with the experimental configuration dictating which precise algorithm is finally used. Next, we will use these identified algorithmic solutions to better understand various phenomena associated with ICL. We will especially focus on investigating the non-monotonic nature of OOD generalization dynamics, i.e., the transient nature of ICL.

## 4    LINEAR INTERPOLATION OF ALGORITHMS: A COMPETITION PICTURE OF NON-MONOTONIC GENERALIZATION DYNAMICS IN ICL

In Sec. 3, we identified four algorithms that decompose the learning dynamics of a model trained on finite mixture of Markov chains into broad algorithmic phases. We now show these algorithms consistently compete with each other to dictate a trained model's behavior (Sec. 4.1), partially driving ICL's phenomenology. Specifically, we analyze the non-montonic generalization dynamics of ICL in Sec. 4.2 (e.g., its transient nature), and how model design (e.g., width, tokenization) affect the algorithmic phases in Sec. 4.3.

### 4.1    LINEAR INTERPOLATION OF ALGORITHMS (LIA)

To begin, we first show that a simple linear interpolation of the four algorithms described in Sec. 3 captures a trained model's behavior, i.e., its next-token predictions, extremely well. Formally, let $\mathcal{A}$ denote the set of our four algorithms [Uni-Ret, Bi-Ret, Uni-Inf, Bi-Inf], then the Linear Interpolation of these Algorithms (LIA) is identified by solving the following problem.

$$\text{LIA: } \underset{w_a, a \in \mathcal{A}}{\arg\min} \, \mathbb{E}_{\mathbf{x}_{1:t}} \left[ p_{\text{model}}(\mathbf{x}_{1:t}) - \sum_{a \in \mathcal{A}} w_a * p_a(\mathbf{x}_{1:t}) \right]^2 , \text{ where } \sum_{a \in \mathcal{A}} w_a = 1 \, \& \, w_a \geq 0. \quad (7)$$

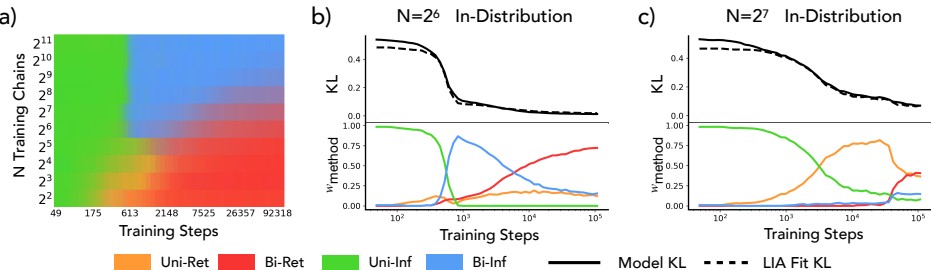

Figure 6: **Linear Interpolation of Algorithms (LIA)** (a) Algorithmic Phases extracted by LIA. We color each location by combining individual colors associated to an algorithm, weighted by their weights $w_a$. (b) LIA weights across checkpoints extracted using ID data for $N = 2^6$. KL of the empirical transition matrix fit by LIA from the ground truth matrix is shown as a black dotted line. The model KL is shown as a black solid line. The evolution of the weights assigned to each algorithm is shown in the lower plot. (c) LIA weights across checkpoints for $N = 2^7$. *This model was trained with learned positional embeddings. See App. F.2, Fig. 36 for the phase diagram.* LIA reveals transitions between algorithms which were seemingly hidden due to the smooth evolution of the ID KL divergence (top panel).

Here, $p_{\text{model}}$ and $p_a$ respectively denote the next-token predictions of the model and individual algorithms from set $\mathcal{A}$, given the sequence $\mathbf{x}_{1:t}$ as input; meanwhile, $w_a$ denotes the weight associated with algorithm $a \in \mathcal{A}$ in the interpolation. We optimize the interpolation weights by minimizing Eq. 7 over multiple ID sequences $\mathbf{x}_{1:t}$, i.e., sequences sampled from Markov chains that constitute $\mathcal{T}_{\text{train}}$ (see App. A.3.4 for further details). Fits are almost perfect for all settings (see App. I, Fig. 39).

**Results.** See Fig. 6. We run the LIA analysis for different amounts of *training steps* and *data diversity*, hence analyzing the dynamics of how the algorithms underlying our identified phases evolve to dictate a model's behavior. Crucially, this fine-grained analysis helps us better understand the model at different phases' boundaries, where we find algorithms may possibly co-occur.

- Fig. 6 (a) shows that LIA qualitatively finds the same dominant algorithm in each phase as ones illustrated in Fig. 5, where we used the bigram utilization and retrieval proximity tests.

- Fig. 6 (b) shows LIA applied across checkpoints for $N = 2^6$, a moderately high data diversity setting. Per panel (a), this setting is the first to not have a particularly dominant `Bi-Inf` phase. We see herein an intriguing dynamic occurs as the model undergoes training: since the bigram solution's KL divergence is lower, the model transitions from `Uni-Inf` to `Bi-Inf` as it undergoes training. *However, after $10^3$ steps, we start to witness transience:* the model slowly cross-overs to utilizing the `Bi-Ret` solution, which performs better than `Bi-Inf` on ID sequences.

- Fig. 6 (c) finally shows that depending on experimental conditions, the order in which different algorithms come to explain the model behavior can be different. For $N = 2^7$ and when using learned positional embeddings (please see App. F.2 for further details), we show that both `Bi-Inf` and `Bi-Ret` are delayed to long after `Uni-Ret` is used as a solution by the model. We emphasize that the evolution of the ID KL, which is essentially the training loss, is smooth; however, LIA detects interesting underlying dynamics that indicate a persistent competition between different algorithms to supersede one another.

## 4.2 UNDERSTANDING NON-MONOTONIC OOD PERFORMANCE WITH LIA

Our results above show that LIA is a useful tool to probe how a model transitions between different algorithms to converge on a solution for the task. Next, we discuss how these transitions shape the evolution of OOD performance, explaining the transient nature of ICL (Singh et al., 2023). We again use LIA on the ID sequences for this analysis—we emphasize that these experiments amount to *predicting OOD performance of a model by merely using the ID data*. See App. A.3.4 for further explanation, and App. F for more experiments in this vein.

**Results.** Fig. 7 shows that the weights extracted via LIA can predict the evolution of OOD performance during training. Fig. 7 (a, b) correspond to the models in Fig. 6 (b) and (c), but this time we are plotting OOD performance (unlike before, when we analyzed the ID performance).

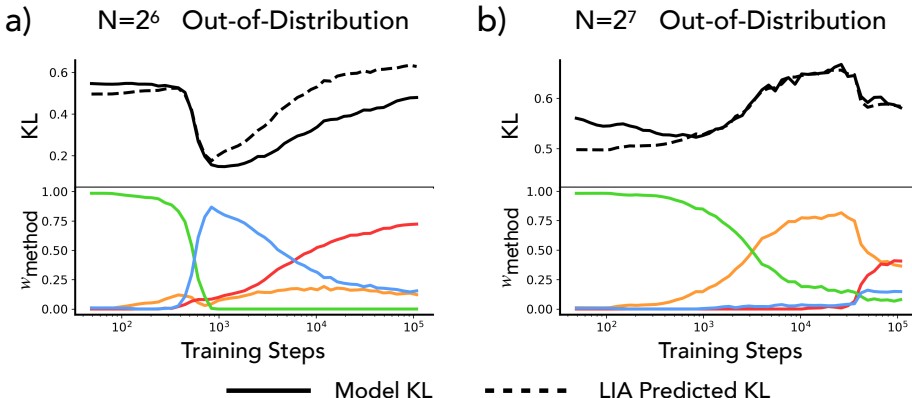

Figure 7: **Algorithmic phase transitions drive non-monotonic OOD performance.** Note that each algorithm's KL in colored dotted lines has changed from Fig. 6 because of the distribution change. (a) Predicting out-of-distribution (OOD) performance corresponding to Fig. 6 (b) using LIA weights. By applying the LIA weights fit on *ID* data, we predict the model's *OOD* performance, explaining the transient nature of ICL. (b) Predicting out-of-distribution (OOD) performance corresponding to Fig. 6 (c) using LIA weights. By applying the LIA weights fit on *ID* data, we predict the model's *OOD* performance, explaining non monotonic OOD performance and sudden changes.

- Fig. 7 (a) shows that as the model undergoes training, the `Bi-Inf` solution, which generalizes extremely well OOD, suddenly comes to dictate the model behavior. This algorithm is thus similar to what prior work calls "task-learning" ICL, since the algorithm is entirely reliant on input context (Raventós et al., 2023). However, as we saw in Fig. 6 (b), the `Bi-Ret` solution slowly takes over because of its superior performance on ID sequences. This causes the OOD performance to degrade since the `Bi-Ret` solution does not generalize well to OOD sequences, as seen in Sec. 3.1. We argue this dynamic underlies the broader ICL phenomenon demonstrated by Singh et al. (2023), who claim ICL can be transient in nature. Specifically, LIA demonstrates that an algorithm that heavily relies on internalized knowledge of the train distribution (i.e., a retrieval solution) consistently competes with the better OOD-generalizing algorithm. *Since the former will ultimately achieve a better loss on ID data, it slowly but steadily will supersede the better generalizing solution, manifesting as the transient nature of ICL.* See also App. C.2.1 for a further detailed analysis and discussion, and App. E for mechanistic evidence in support of these claims: we show we can reconstruct transition matrices from MLP neurons after the model returns to the `Bi-Ret` phase, but not in the `Bi-Inf` phase.

- Fig. 7 (b) shows that the emergence of `Bi-Inf`, which only changes the ID performance slightly, affects the OOD performance more drastically. This demonstrates that certain changes in OOD performance can be predicted by carefully decomposing the model's strategy for performing a task. This result also complements the results in (a), demonstrating that *both ascents and descents in OOD performance can be explained with algorithmic transitions on the training set.*

Overall, the results above show that training dynamics of sequence modeling tasks can be thought as a competition of algorithms on the training set; the generalization performance of the model is a reflection of the current combination of algorithms used.

## 4.3 MODEL ANALYSIS USING ALGORITHMIC PHASE DIAGRAMS

The core finding of Sec. 4.2 is that *algorithmic transitions characterize model behavior under different experimental configurations*. An extremely crucial component of this configuration is the precise set of design decisions made to define the model we are training. For example, as shown in prior work (Kirsch et al., 2022), scaling the width of the model can impact its ICL abilities. Building on this, we now analyze the effects of model design choices on ICL, specifically evaluating the effects of model size, data complexity, and tokenization. See App. F for more experiments.

**Results.** See Fig. 8, which yields the following observations.

- Fig. 8 (a) shows the phase diagram when using a model with an embedding dimension of 256, which is 4 times bigger than our baseline experiments. We observe that the data diversity re-

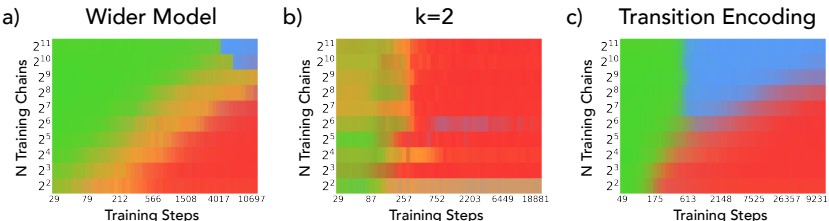

Figure 8: **Algorithmic Phase Diagrams Explain the effects model size, data *complexity*, and tokenization.** (a) Phase diagram for a wider model with an embedding dimension of 256. Here, we find that the data diversity required to *observe* `Bi-Inf` is increased to $2^{10}$. (b) A decreased state space ($k = 2$, decrease data complexity) enhances the retrieval solutions and causes `Bi-Inf` to almost never appear. (c) Encoding the transitions into the tokens removes the `Uni-Ret` solution by making it the more complex and higher train loss solution compared to `Bi-Ret`.

quired to observe `Bi-Inf` is increased $16\times$. Since there is no *a priori* reason to believe a wider model cannot implement the `Bi-Inf` algorithm, our conclusion is that `Uni-Ret` and `Bi-Ret` algorithms are relatively faster to appear in a bigger model, hence impeding `Bi-Inf`'s ability to succeed in the overall competition. This finding is intuitive as a bigger model will have more parameters to memorize the training set transition matrices. We also note that this result is in contrast to Kirsch et al. (2022)'s, who claim width scaling leads to (faster) emergence of "task-learning" ICL. We provide further discussion of this point in App. C.2.2, C.2.3.

- Fig. 8 (b) shows the case where the state space of the DGP is set to $k = 2$, reducing the complexity of the data (note that this is independent of the data diversity; see App. D for a discussion). In this case, we find that the model can easily internalize the transition matrices needed for the `Bi-Ret` solution—even $N = 2^{11}$ chains do not allow the `Bi-Inf` solution to win the competition.

- Fig. 8 (c) shows the effect of expanding the *token space* to allow each token to represent the last state *and transition* (See App. A.2 for details). This experiment is motivated to understand the effect of a tokenization on downstream abilities. We find tokenization allows the model to be able to count transitions without a formation of a complex attention head, which we suspect is one of the reasons `Bi-Ret` is slower to learn than `Uni-Ret` (see App. J for a discussion and App. E.3 for attention head visualizations). As expected, we find the `Uni-Ret` phase disappears in this case, since `Bi-Ret` is both superior on the training set and (now) the simpler solution.

Overall, our results shows that the downstream effect of design choices can be well understood at the algorithm level. Please see App. F for additional perturbations, including positional embeddings, model depth, and number of attention heads.

## 5    CONCLUSION

In this study, we introduced *finite Markov mixtures* as a model system of ICL which reproduces a myriad of phenomena discovered in recent studies of ICL, hence offering a unified setting for studying the concept. This setup also allowed us to write down four algorithmic solutions, each with their peculiarities, that can explain the trained model's behavior. We then decomposed trained models into a combination of these solutions, revealing a competition dynamics between the algorithms to dictate model behavior. These dynamics result in an *algorithmic phase diagram of in-context learning* spanning data diversity and optimization, and can be interpreted as the model finding the best algorithm on the training data, leading to both sudden and slow transitions towards better solutions. These transitions of algorithms can offer insights into ICL's phenomenology, e.g., offering a mechanism that leads to the transient nature of ICL (Singh et al., 2023). More broadly, we claim our findings challenge the traditional "more is better" view of scaling laws by showing that ICL emerges from competing algorithmic behaviors rather than a single mechanism. This insight suggests a fresh perspective on how we should approach model development: instead of solely focusing on reducing the loss through scaling, *can we **promote** desired algorithms over competing alternatives that may achieve lower training loss but generalize poorly*? Through careful design of data composition, model architecture, and training duration, we can potentially guide models toward implementing more robust and generalizable algorithms.

## ACKNOWLEDGMENTS AND DISCLOSURE OF FUNDING

CFP and HT gratefully acknowledge the support of Aravinthan D.T. Samuel. CFP acknowledges the support of Cecilia Garraffo and Douglas P. Finkbeiner. IP was supported by BERI. The computations in this paper were run on the FASRC cluster supported by the FAS Division of Science Research Computing Group at Harvard University. The authors thank Gautam Reddy, Pulkit Gopalani, Robert Kirk, Andrew Lee, Daniel Wurgaft, Maya Okawa, Cole Gibson, Alex Nguyen, Sadhika Malladi, Zechen Zhang, Helena Casademunt, Yongyi Yang, and Wei Hu for useful discussions.

## CONTRIBUTIONS

CFP and ESL independently defined the setup, ran preliminary investigations, and converged on a bulk of the experiments, evaluation protocols, and phenomenology; CFP ran extensive experiments to verify the latter. CFP ran the final experiments in Section 3 in collaboration with ESL to identify the algorithmic phases and uncover transience of ICL. This formed the basis of the paper. ESL and CFP proposed the LIA protocol in Section 4, which they used to develop explanations for ICL's phenomenology; CFP led the experiments. ESL and CFP performed a first version of the mechanistic interpretability analysis, which IP significantly expanded. IP also proposed the memorization analysis and the analysis of attention maps at phase boundaries. ESL defined the project narrative and wrote the paper, with inputs from all authors. CFP, ESL, and HT designed the conceptual diagrams. HT supervised the project.

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

APPENDIX

CONTENTS

## A  EXPERIMENTAL DETAILS

### A.1  DATA GENERATING PROCESS

The Data Generating Process (DGP) is defined by 3 crucial hyperparameters:

1. $k$: The number of states in the Markov chain, fixed to 10 unless mentioned otherwise.
2. $l$: The length of the sequence generated, fixed to 512 unless mentioned otherwise.
3. $N$: The number of training matrices, $N \in \{2^2, 2^3, \ldots, 2^{11}\}$ for the main experiments.

The set of Markov chains used for training is constructed by drawing $N$ transition matrices. We draw each row of each transition matrix from a Dirichlet distribution with parameter $\alpha = \mathbf{1}_k$. We index individual transition matrices via a subscript, i.e., $T_n$ and use square brackets to index the matrix elements, e.g., $T_{[i,j]}$. Thus, the next state probability is $p(x_{t+1} = j | x_t = i) = T_{[i,j]}$. The probability of selecting a chain is also drawn from a Dirichlet distribution with parameter $\alpha = \mathbf{1}_N$. We denote this prior as $p_n$ in equation Eq. 4, although its precise values did not yield any interesting effects in our experiments. To generate data, we first choose a transition matrix from the prior probability $p_n$, define a Markov chain with that transition matrix, and then sample a sequence of length $l$.

Each transition matrix $T_n$ naturally defines a single stationary distribution we denote as $\pi_n$. We draw the first state from this stationary distribution to initialize a sequence. By definition, this distribution should remain the same when moving forward in the chain, i.e., it satisfies the relation $\pi_n = T_n \pi_n$; we calculate the stationary distribution analytically from this relation. Please see the function `get_stationary_distribution` in the code snippet below.

To make the dataset clear we attach a minimal PyTorch (Paszke et al., 2019) implementation. We generated this code with the help of GPT4o (OpenAI, 2024). The code is for clarity only and is not optimized for speed or compatibility.

```python
import torch
from torch.utils.data import IterableDataset
import numpy as np

class MarkovChainDataset(IterableDataset):
    def __init__(self, k, l, n, seed=None):
        self.k = k  # Number of states
        self.l = l  # Output sequence length
        self.n = n  # Number of transition matrices
        self.seed = seed  # Seed for reproducibility

        if seed is not None:
            np.random.seed(seed)
            torch.manual_seed(seed)

        # Generate n transition matrices, each with k states
        self.transition_matrices = []
        for _ in range(n):
            matrix = np.array([np.random.dirichlet([1] * k) for _ in
    range(k)])
            self.transition_matrices.append(matrix)

        # Prior distribution over transition matrices
        self.prior = np.random.dirichlet([1] * n)

    def get_stationary_distribution(self, matrix):
        # Compute the stationary distribution of the transition matrix
        eigvals, eigvecs = np.linalg.eig(matrix.T)
        stationary = np.real(eigvecs[:, np.isclose(eigvals, 1)])
        stationary = stationary[:, 0]
        stationary /= stationary.sum()
        return stationary

    def sample_chain(self, transition_matrix):
```

```
34          # Sample the first state from the stationary distribution
35          stationary_distribution = self.get_stationary_distribution(
     transition_matrix)
36          first_state = np.random.choice(self.k, p=stationary_distribution)
37
38          # Generate the sequence
39          sequence = [first_state]
40          for _ in range(1, self.l):
41              current_state = sequence[-1]
42              next_state = np.random.choice(self.k, p=transition_matrix[
     current_state])
43              sequence.append(next_state)
44
45          return sequence
46
47      def __iter__(self):
48          while True:
49              # Choose a transition matrix based on the prior
50              matrix_index = np.random.choice(self.n, p=self.prior)
51              chosen_matrix = self.transition_matrices[matrix_index]
52
53              # Generate a sequence using the chosen transition matrix
54              sequence = self.sample_chain(chosen_matrix)
55              yield torch.tensor(sequence)
56
57  # Example usage:
58  # k = 3 (states), l = 10 (sequence length), n = 5 (transition matrices),
       seed = 42
59  dataset = MarkovChainDataset(k=3, l=10, n=5, seed=42)
60  iterator = iter(dataset)
61
62  # Get a sample sequence
63  sample_sequence = next(iterator)
64  print(sample_sequence)#e.g. tensor([8, 8, 9, 5, 3, 4, 8, 8])
```

Listing 1: *Markov Mixtures* Data Generating Process

## A.2 TRAINING DETAILS: MODEL & OPTIMIZATION

**Model Architecture**   We train a Transformer model with softmax attention (Vaswani et al., 2023) on sequence data generated by the DGP described above in Sec. A.1. We adapted code from nanoGPT (Karpathy, 2022), and implemented Rotational Positional Embedding (RoPE) (Su et al., 2023) instead of the default learning positional embedding, which significantly delayed the emergence of Bi-Inf. Further discussion about these results are in App. F.2. All matrix weights are initialized as $\mathcal{N}(0, 0.02)$ except residual projections which are initialized as $\mathcal{N}(0, 0.02/\sqrt{2 * N_{\text{layer}}})$. All biases are initialized as zero. The embedding layers for Sec. F.2 are initialized from $\mathcal{N}(0, 0.02)$. We trained our models on NVIDIA A100 80 GB GPUs, running 5 experiments in parallel on the same GPU, and hence yielding a wallclock time of 3.5 hours.

**Tokenization**   We tokenize each state as a single token. Since we do not have task tokens or separator tokens, the model is trained on $k$ tokens. The only exception is the transition encoding tokens experiments in in Sec. 4.3 and Fig. 8, where we include the last transition into all tokens. In this case, we define $k^2 + k$ tokens where $k^2$ tokens are used to represent all transitions while $k$ tokens are used only to indicate the first state, for which the last transition is not defined.

**Optimization**   We trained the model above using sequences from the DGP in Sec. A.1 with an autoregressive next-token prediction cross-entropy loss. We used the AdamW optimizer (Loshchilov & Hutter, 2019) with learning rate $6 \times 10^{-4}$. We kept the *estimated* FLOPs constant to $1 \times 10^{16}$, where we estimate the compute by $6DN$ ($D$ denotes the number of tokens seen during training and $N$ denotes the number of model parameters). This compute estimate is not proportional to wall time, especially since we operate with small models, but are kept for consistency when scaling the model

in order to normalize for faster optimization of bigger models (Kaplan et al., 2020; Hoffmann et al., 2022; Bordelon et al., 2024). See App. C.2.3 for further details on scaling. Using a batch size of 128 resulted in a training of 107978 steps. Changing the batch size to 64 or 256 had no significant changes to results to the best of our knowledge. We experimented with a learning rate warmup and cooldown, sometimes pointed out to be crucial (Liu et al., 2019; Gotmare et al., 2018), but found no significant difference.

### A.3 EVALUATION DETAILS

Our evaluation process involves two scenarios: In-Distribution (ID) and Out-Of-Distribution (OOD). For ID evaluations, we select a transition matrix $T^*$ from the set $\mathcal{T}_{\text{train}}$, i.e., the set of transition matrices used to sample sequences for training; for OOD evaluations, we sample a *novel* transition matrix $T^*$, using again the Dirichlet prior over the row elements[2]. We next define a Markov chain using $T^*$ and sample a sequence of length $l_{\text{eval}}$ to feed as input to the trained model; unless mentioned otherwise, $l_{\text{eval}} = 400$. The model then computes a probability distribution over possible next tokens given this sequence as context. Repeating this process, we can collect pairs of last tokens from the in-context sequence and the model's predicted next token probabilities. This allows us to construct an **empirical transition matrix**, $\hat{T}$, that denotes the model's inferred state transitions based on the provided context and the prior knowledge it may have internalized during training (e.g., $\mathcal{T}_{\text{train}}$). To assess how accurate this predicted transition matrix is, we calculate the *expected* KL divergence between $\hat{T}$ and $T^*$ by marginalizing over the stationary state distribution of $T^*$ (denoted $\pi^*$). Details are below.

#### A.3.1 KL DIVERGENCE

We compute the Kullback-Leibler divergence (Kullback & Leibler, 1951) between a model or a method's predicted probabilities and the GT probabilities from a transition matrix rows to quantify performance. We draw an evaluation transition matrix $T^*$ from either the training set or the Dirichlet prior respectively to quantify the ID and OOD performance. Given sequences drawn from $T^*$, we estimate the transition matrix $\hat{T}$ inferred by either a model checkpoint or one of the two retrieval approaches proposed in Sec. 3.1 (which depend on the context). Given this $\hat{T}$, the average KL divergence for a distribution of transition matrices $\mathcal{T}$ is then quantified by:

$$\left\langle KL(\hat{T}\|T^*)\right\rangle_{T^*\in\mathcal{T}} = \left\langle \sum_i \pi^*_{[i]} \sum_{j=1}^k \hat{T}_{[i,j]} \log \frac{\hat{T}_{[i,j]}}{T^*_{[i,j]}} \right\rangle_{T^*\in\mathcal{T}}. \tag{8}$$

#### A.3.2 ESTIMATING TRANSITION MATRICES

To estimate a transition matrix, we need sequences which end with all states in the state space, so that every row of the estimates transition matrix $T$ can be filled. Thus, for each context length, we generate $k$ sequences which end with the states 1 to $k$. The exact way the estimated transition matrices $\hat{T}$ for the model and the methods are computed is as follows.

- $\hat{T}_{\textbf{Model}}$: We draw $k$ sequences from $T^*$ ending in states from 1 to $k$, controlling for the desired context length $l_{\text{eval}}$ ($= 400$, unless mentioned otherwise). Given a sequence ending with a certain state, $x_t = i$, we can evaluate the next state probability $p(x_{t+1} = j|\mathbf{x}_{1:t})$ by a forward pass through the model and taking the softmax of the logits to obtain an estimate of the matrix element $\hat{T}[i,j]$ for all $i$ simultaneously. We iterate this process for all last states $j \in [1, \cdots k]$ to estimate the whole transition matrix.

- $\hat{T}_{\texttt{Uni-Ret}}$: We use the same procedure to draw evaluation sequences as above, and use Eq. 2 and Eq. 4 to estimate the next state probability.

- $\hat{T}_{\texttt{Bi-Ret}}$: We use the same procedure to draw evaluation sequences as above, and use Eq. 3 and Eq. 4 to estimate the next state probability.

---

[2]We use the Dirichlet prior to define OOD Markov chains primarily for consistency with the training prior. We do note that our preliminary experiments show that there are certain phases of training configuration wherein the model generalizes to essentially arbitrary prior distributions (see App. G and Fig. 37).

- $\hat{T}_{\texttt{Uni-Inf}}$: We use equation Eq. 5 to estimate the transition matrix directly.
- $\hat{T}_{\texttt{Bi-Inf}}$: We use equation Eq. 6 to estimate the transition matrix directly.

We repeat this process $n_{\text{rep}} = 30$ times for statistical power. For ID evaluations, we choose $T^*$ from the training set $\mathcal{T}_{\text{train}}$ using the task prior $p_n$ defined in App. A.1. For OOD evaluations, we draw each row of $T^*$ directly from the Dirichlet distribution as seen in App. A.1.

### A.3.3 BIGRAM UTILIZATION AND RETRIEVAL PROXIMITY

We quantify Bigram Utilization and Retrieval Proximity using the following procedure:

**Bigram Utilization** We quantify Bigram Utilization as described in the main text: we shuffle the context and measure the increase in KL. We normalize this value by the stationary KL divergence given by simply predicting the stationary distribution *over the whole training set.*

$$\texttt{Utilization} = \text{clip}\left(\left\langle \frac{KL(\hat{T}_{\text{model}}^{\text{Shuffled}}||T^*) - KL(\hat{T}_{\text{model}}||T^*)}{KL(T_{\text{Stationary}}^*||T^*)} \right\rangle, 0, 1\right) \tag{9}$$

where $<>$ is an ensemble average over different samples of ID sequences, $\hat{T}_{\text{model}}^{\text{Shuffled}}$ is the model estimated transition matrix when we shuffle all states in the context randomly, and $T_{\text{Stationary}}^*$ describes the transition matrix corresponding to the perfect stationary solution, i.e.,

$$T_{\text{Stationary}[i,j]}^* = \pi_j^*. \tag{10}$$

**Proximity to a Retrieval Approach** We quantify proximity to a retrieval approach by constructing a set of transition matrices, denoted $\mathcal{T}_{\text{random}}$, with the same number of matrices as the training set. If a model has no bias towards the training set, $\hat{T}_{\text{model}}$ inferred from a sequence drawn from $T^*$ (which itself has no bias towards the training set) should be closer to $\mathcal{T}_{\text{random}}$ with 50% chance, and thus the expected value of the fraction in Eq. 11 should be unity, yielding null proximity. If the model generates a sequence precisely from one of the training set matrices, the demoninator will vanish yielding unity proximity.

$$\texttt{Proximity} = \text{clip}\left(\left\langle 1 - \frac{\min_{T \in \mathcal{T}_{\text{train}}} KL(T||\hat{T}_{\text{model}})}{\min_{T \in \mathcal{T}_{\text{random}}} KL(T||\hat{T}_{\text{model}})} \right\rangle, 0, 1\right) \tag{11}$$

$<>$ is an ensemble average over different samples of ID sequences.

### A.3.4 LINEAR INTERPOLATION OF ALGORITHMS

Recalling the main text, we defined LIA as:

$$\text{LIA: } \underset{w_a, a \in A}{\arg\min} \, \mathbb{E}_{\mathbf{x}_{1:t}} \left[ p_{\text{model}}(\mathbf{x}_{1:t}) - \sum_{a \in A} w_a * p_a(\mathbf{x}_{1:t}) \right]^2, \text{ where } \sum_{a \in A} w_a = 1 \, \& \, w_a \geq 0. \tag{12}$$

where we are interested in extracting $w_a$ from the model output probabilities and individual algorithm's probabilities. The positivity constraint exists so that methods can not destructively compensate. We used the squared error in probability space instead of, e.g. KL divergence, since we found similar results when the fit was successful, while avoiding numerical subtleties.

In practice, we used 300 independent chains to optimize for LIA and find the method weights. To use the method weights for OOD prediction, we simply apply:

$$\text{LIA prediction: } p_{\text{pred}}(\mathbf{x}_{1:t}) = \sum_{a \in A} w_a * p_a(\mathbf{x}_{1:t}) \tag{13}$$

where $\mathbf{x}_{1:t}$ is the context sequence.

We note that analyzing the deviation of model outputs to a single analytic solution has been explored in a linear regression setting (Raventós et al., 2023). In contrast, LIA extracts the weights of each member solution in a set of solutions using Eq. 7.

## A.4 Learning Curves

Fig. 9 shows the learning curves (loss) and ID / OOD KL divergence through training. The data is the same as in Fig. 3, but shown as a plot.

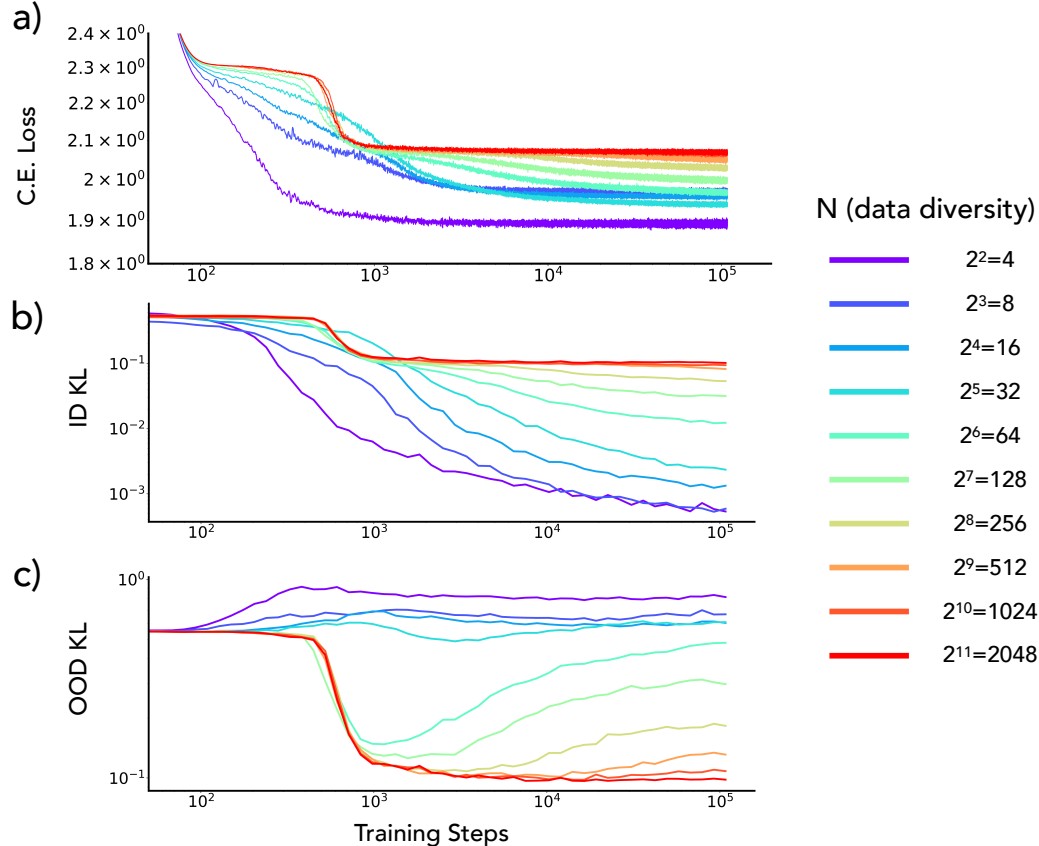

Figure 9: **Online Training loss, ID KL divergence and OOD KL divergence** We show the loss and the ID/OOD KL divergence. **a) The next-token cross entropy loss** averaged over the whole sequence. At high data diversity, we find a sudden loss drop with the formation of an induction head. After loss drops, sometimes we find a slower further decrease of the loss. We have shown that this dynamics is behind the transient nature of ICL. **b) In-Distribution KL divergence.** The ID KL divergence is monotonically decreasing as expected. This is a simple re-parametrization of the loss. **c) Out-of-Distribution KL divergence.** The OOD KL divergence simply rises for low data diversity (See Sec. D) while it suddenly drops for high data diversity. However, we find that this drop can be transient, as discussed in Sec. 2.1 and explained in Sec. 4.2.

## B    MORE ABOUT SOLUTIONS

### B.1    INFERENCE APPROACHES ARE "BAYESIAN" WITH A RELAXED PRIOR

The inference approaches, i.e., `Uni-Inf` and `Bi-Inf`, can be interpreted as a Bayesian inference operation with a relaxed hypothesis space. This relaxed hypothesis space is the infinite dimensional space of all transition matrices drawn from a Dirichlet distribution. This relaxed prior is in fact required to set up the `Uni-Inf` and `Bi-Inf` transition matrices, since without it some matrix elements will be strictly zeros for small context length (specifically, for unobserved transitions). This interpretation thus sets a prior distribution so that all transition matrix elements are non-zero even if a transition is not observed. This can be seen by setting the second term of the numerator to zero in Eq. 6. The solution is still Bayesian, but only has a "memory" of *the distribution of transition matrices* and not the distribution of the sequences itself (i.e., the precise transition matrices seen during training). However, we expect the context dependent part of Eq. 5 and Eq. 6 to very quickly dominate the estimate of $T$ as we increase the context, hence yielding a frequentist solution. Specifically, this can be seen in the distributional OOD example in Fig 37, where the context is drawn from a distribution extremely far from the Dirichlet prior.

### B.2    UNIGRAM POSTERIOR SOLUTIONS

Other than the four algorithms discussed in the main text (see Sec. 3), we formalized two more algorithms but found that they did not significantly contribute to the model explanation in the linear combination of algorithms analysis. Thus we focused our analysis on the four solutions in the main text. Nevertheless, we describe these two solutions and their properties here.

**Unigram based Retrieval with Unigram Posterior.** This algorithm is similar to `Uni-Ret` in Sec. 3.1, however it draws the *next token* from the *stationary* distribution corresponding to the chosen training set chain. This algorithm is expected to be less expressive than `Uni-Ret`, but we nevertheless observe a small contribution from it (overlapping with `Uni-Inf`), especially at high $N$. This is expected as a large number of $N$ can easily span the full distribution of stationary distributions from a Dirichlet distributed transition matrix.

**Bigram based Retrieval with Unigram Posterior.** This algorithm is similar to `Bi-Ret` in Sec. 3.1, however it draws the next token from the *stationary* distribution corresponding to the chosen training set chain. Although the likelihood is sharper than unigram posterior retrieval, exactly the way `Bi-Ret`'s likelihood is sharper than `Uni-Ret`, this algorithm is again not expressive due to the nature of its output distribution.

## C  PHENOMENOLOGY OF IN-CONTEXT LEARNING: SUMMARY OF REPRODUCED RESULTS AND IMPROVED UNDERSTANDING

### C.1  LIST OF PHENOMENA WE REPRODUCE

We first provide a short summary of ICL phenomenology we reproduce in this work.

1. **Emergence of Non-Bayesian ICL with data diversity**: Prior work exploring linear regression tasks shows that ICL performance drastically improves on OOD data with increase in data diversity Raventós et al. (2023); Kirsch et al. (2022); Lu et al. (2024). In specific, Raventós et al. (2023) shows that there is a threshold needed for a "non-Bayesian" ICL to emerge. Here, we reproduce this data diversity threshold in We reproduce this phenomena as seen in Fig. 1 (a), Fig. 3 (a,c) and Fig. 27 (a). We show that this transition happens because of an "inference" approach (see Sec. 3.2) becoming better than the available "retrieval" approach at this data diversity, echoing the results of Lu et al. (2024). Furthermore we show that in order to observe this transition we need a certain optimization threshold to be reached before a Bayesian circuit (`Bi-Ret`) dominates. We also show that this threshold can further increase with more optimization – the transpose effect of the transient nature of ICL.

2. **Formation of induction heads and variants**: Induction heads are a specialized attention head that help infer next-token predictions in a context-conditioned manner Elhage et al. (2021); Olsson et al. (2022); Reddy (2023); Edelman et al. (2024); Akyürek et al. (2024). Often, there is a sudden loss drop that correlates with induction head formation, and we find consistently find this drop occurs in our results with training time: this drop, in fact, is the cause of sudden transition from unigram to bigram dependence in the model. We reproduce this phenomena as seen in Fig. 1 (b), Fig. 3 (a,c) and Fig. 27 (b). Interestingly, we find that data-diversity has almost no effect to the formation of this circuit. In App. K, we discuss that this transition is likely an "Optimization Limited Emergence" as opposed to the `Bi-Inf` vs. `Bi-Ret` transition which is mostly data limited.

3. **Transient Nature of In-Context Learning**: Recently researchers have found that ICL can be transient during pre-training Singh et al. (2023); Anand et al. (2024); Panwar et al. (2024). In this work we show that this happens when a solution performing better on the training set emerges later in optimization, likely due to its complexity. We reproduce this phenomena when there is enough data diversity to allow `Bi-Inf` to emerge before `Bi-Ret` is implemented, largely between $N = 2^5$ and $N = 2^{10}$. This is demonstrated in Fig. 1 (c), Fig. 27 (c). See also Fig. 10) for a direct comparison with Singh et al. (2023).

4. **Task Retrieval to Task Learning transition**: Researchers have classified the operation of ICL as either "task retrieval" or "task learning" (Min et al., 2022; Pan et al., 2023). To this end, such papers explore effects of, e.g., shuffling next-token predictions in few-shot tasks and finding that one can still achieve almost similar performance as the scenario when exemplars are matched with the correct labels (Min et al., 2022; Lu et al., 2021). In our work, the unigram retrieval approach to ICL captures this phenomenon: since shuffling the tokens does not affect stationary distribution of a sequence, we can retrieve the relevant transition matrix regardless of the states being matched to next-state transitions with the correct probabilities. In other words, *in context learning performance might not need correct labels if the circuitry to correctly use the question-answer relation has not developed during training*. We show that this happens in our setting when the model is under-trained and thus the bigram circuit has not formed yet (see Fig. 5 a).

5. **Early Ascent**: Early ascent describes the phenomena where the error/risk on an ICL task initially increases before decreasing. This phenomena is empirically observed in Xie et al. (2021) while Lin & Lee (2024) suggests an explanation with the linear regression setting. In this work, we show that this phenomena can be reproduced when training with an intermediate data diversity $N = 2^5$ chains with sufficient optimization to enable `Bi-Ret` to dominate at small context length. In these settings, the model uses a context length dependent superposition of `Bi-Ret` and `Uni-Inf`, as seen in Fig. 1 (e), Fig. 27 (e).

6. **Bounded Efficacy**: Bounded efficacy of biased label ICL is observed empirically in Min et al. (2022) and coined as a term in Lin & Lee (2024). In this work, we show that this happens when the model's algorithm is a superposition of a retrieval solution and an inference solution, as seen in Fig. 1 (f), Fig. 27 (f). In this case, at short context length the model mostly uses `Uni-Ret`,

and thus retrieving the right task even though the context is shuffled (biased labels) while at long context length it uses `Bi-Inf`, thus inferring a wrong transition matrix

## C.2 IMPROVED UNDERSTANDING OF ICL: INSIGHTS INTO TRANSIENCE AND EFFECTS OF MODEL SCALING

We highlight two specific insights on understanding ICL drawn from our experiments: the dynamics of transience and the effects of model size scaling.

### C.2.1 TRANSIENT NATURE OF IN-CONTEXT LEARNING

We reproduce the transient nature of ICL in Fig. 10, making a clear comparison to Singh et al. (2023). Then, applying LIA (see Sec. 4.1), we find a precise dynamic that underlies transience: an algorithm that performs better on the training set (and hence ID), but perhaps is more complex to represent, slowly and steadily comes to dominate the algorithm that performs well OOD. This yields severe performance degradation on OOD data and leads us to the conclusion: *if the best solution on the training set is one that does not generalize OOD, but is slowly learned due to learning signal from the training loss, ICL can be transient.* We also find an interesting memorization dynamic underneath this result. Specifically, as we show in App. E, we can find neurons which are responsible for encoding specific state transitions (i.e., they have low KL divergence with respect to transition probabilities) in the `Bi-Ret` phase of learning in Fig. 10, but not in the `Bi-Inf` phase of learning!

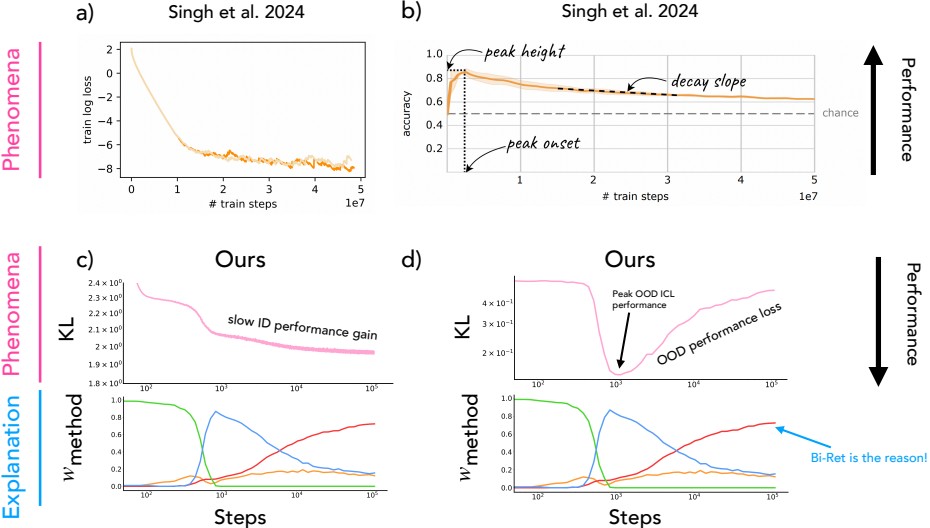

Figure 10: **Explaining the Transient Nature of In-Context Learning** Our setup and LIA allows a clear understanding of the transient nature of In-Context Learning. a,b) are panels from Singh et al. (2023). a) shows that the training loss slowly decreases after a initial drop. b) shows that during the slow drop of training loss the ICL accuracy decreases. c) We find the same phenomenology in our experiments. The ID KL divergence, directly related to the model's loss (see App. A.4), slowly decreases after an initial drop. Our setup has two initial drops due to the formation of induction heads. d) Just like b), we find a performance loss (KL increase) in the OOD evaluations during the slow decrease of ID KL. We have a very clear explanation to this phenomena in the panels below. The ID KL decreases as the `Bi-Ret` solution, optimal on the training set, slowly replaces `Bi-Inf`. This change causes the OOD performance to degrade.

### C.2.2 ICL AND MODEL-SIZE SCALING

Kirsch et al. (2022) analyzed Transformer models trained to meta-learn, i.e., to learn tasks in-context (aka ICL). Therein, the authors run experiments by changing the model size and find that bigger models can develop a "General Purpose ICL" solution. Here, we produce similar results, as shown in

Fig. 11. However, on further investigation, we found evidence that our results are *confounded* by the faster learning speed of bigger models (Kaplan et al., 2020; Hoffmann et al., 2022; Bordelon et al., 2024). **We thus properly normalized the training by FLOPs, as shown in the next subsection, and found no model scale dependent emergence.** We believe this finding also explains the results by Kirsch et al. (2022), i.e., we claim their results are confounded by lack of FLOPs-normalization!

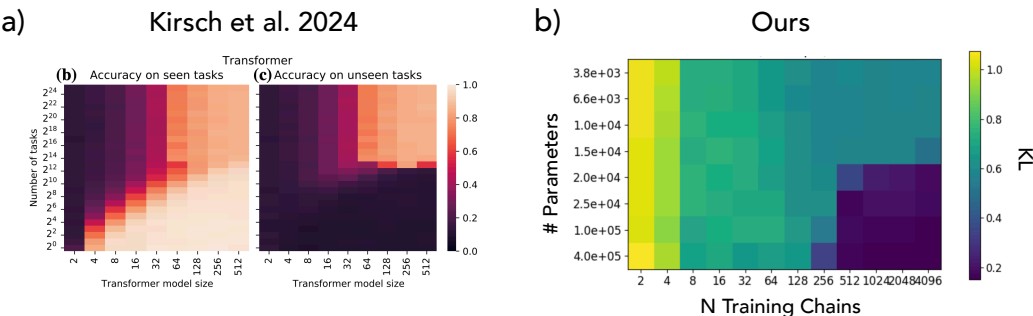

Figure 11: **Effect of Model Scaling?** We find results similar to Kirsch et al. (2022), where we observe both a data diversity and model size threshold. a) Figure from Kirsch et al. (2022). b) OOD KL depending on model scale, when training for 10000 steps. However, we believe that these results are caused by keeping the number of *steps* constant and not FLOPs. See Sec. C.2.3.

### C.2.3   EFFECTS OF MODEL-SIZE SCALING, REVISITED

As argued above in App. C.2.2, we found a model size threshold to ICL emergence. However, here we refute our own findings (and hence those of Kirsch et al. (2022)). To this end, we study the effects of model-size scaling at the algorithmic phase diagram level (see Sec. 3).

**Effects of equi-FLOPs training while varying model size.** First, we train models with equal estimated FLOPs, calculated as $6DN$ (Hoffmann et al., 2022), where $D$ is the number of tokens passed through the model and $N$ is the number of parameters in the model. This results in a much bigger wall-time for smaller models as they will run for vastly more steps. However, we proceed with this normalization for an analogy to real systems where training is bottlenecked by actual GPU compute unlike our smallest models. The resulting phase diagrams for widths of $\{32, 48, 64, 96, 128, 192, 256\}$ are shown in Fig. 12. Generally, we find that `Bi-Inf`, the generalizing ICL solution, is in fact suppressed up to higher data diversity. This is well aligned with the intuition that bigger models have more memorization capacity and thus develop a memorizing solution more easily.

**Effects of equi-FLOPs training while varying model size.** Next, we construct a diagram spanning data diversity and model width, and show it for different amount of FLOPs. This result is shown in Fig. 13. We now see that the effect of model size threshold for `Bi-Inf` is entirely removed! In fact, smaller models are more robust at developing the `Bi-Inf` solution given the same amount of FLOPs. *This highlights that normalizing for FLOPs and steps can yield very different qualitative conclusions about the role of model scaling for ICL.*

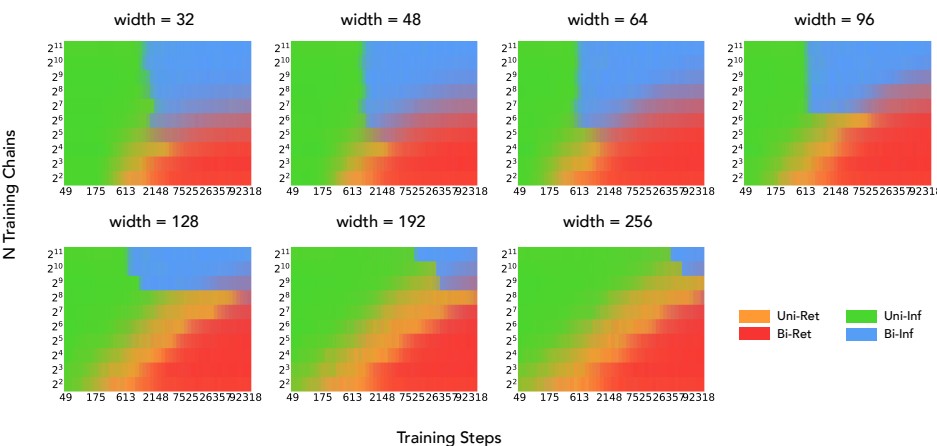

Figure 12: **Algorithmic Phase diagrams depending on model width** We show algorithmic phase diagrams as we increase the model's embedding dimension from 32 to 256. Larger models seem to enhance the memorization solutions.

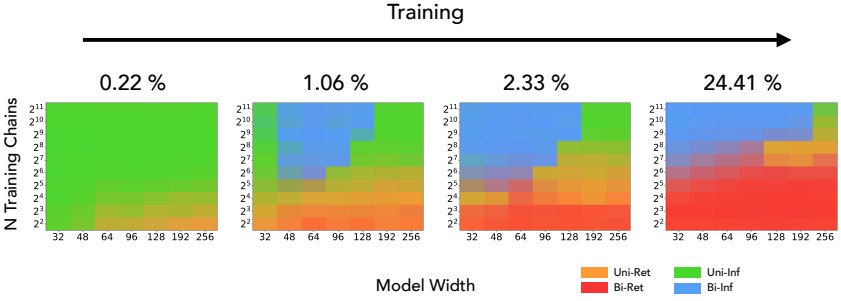

Figure 13: **Algorithmic Phases spanning data diversity and *model scale*.** We see the algorithmic phases of models depend on data diversity and their width. We find that smaller models form `Bi-Inf` solution more efficiently, when properly normalizing for FLOPs.

## D HIGH DIMENSIONAL DISTANCES

At first glance, it could be confusing why a retrieval approach would perform worse with more context on OOD chains. Both retrieval approaches discussed in the main paper (see Sec. 3) achieve a lower KL on ID sequences when given more context, as they can determine the transition matrix with higher precision, i.e., sharper likelihood. However, not only their performance on OOD sequences is worse in general, the KL becomes higher with more context. This can be confusing, as even for OOD sequences they are still choosing the "closest" transition matrix—a process that should become precise with increasing context.

This counterintuitive result originates from properties of distances in higher dimensions. Fig. 14 compares the KL of a freshly drawn $T^*$ to the distributional mean and the nearest neighbor from a big set of draws, representing the training set. In 2 dimensions, as seen in Fig. 14 (a), it is clear that the nearest neighbor will be closer to a novel draw compared to the distributional mean. However in higher dimensions, it is increasingly the case that the distance to the distributional mean is closer than the nearest neighbor. We quantify this in Fig. 14 (b,c,d) using 30 different seeds. For different values of $N$, i.e., number of Markov chains in training data, we find consistently that as $k$ increases, the KL to the distributional mean increases much more moderately than the nearest neighbor KL. This is the reason why a retrieval approach could perform worse with more context: as more context is added, a retrieval approach chooses a transition matrix with a higher precision. However, as seen from the tests in Fig. 14, *choosing the nearest neighbor transition matrix is in fact worse than averaging over all transition matrices seen in training uniformly.*

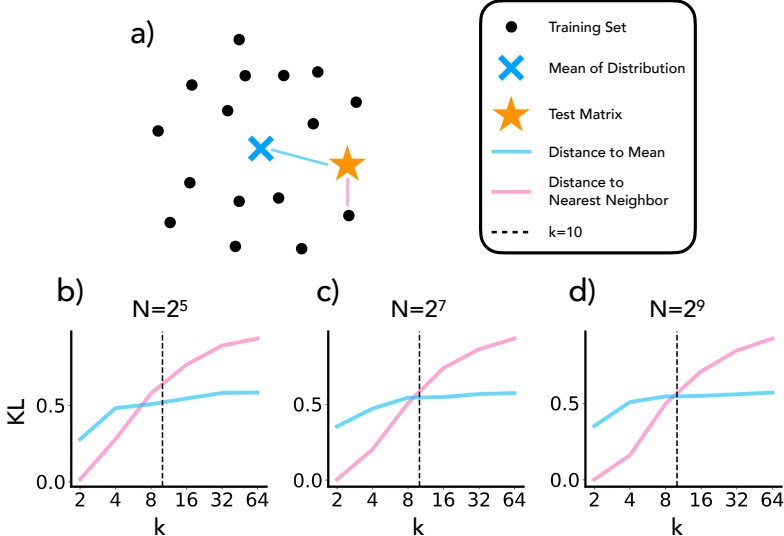

Figure 14: **High Dimensional Distances** We show that the distance of a point to the nearest neighbor within a set of points increases faster than the distance to the distributional mean as the dimension increases. (a) Schematic of the distances. In 2 dimensions, it is much more intuitive to think that the nearest neighbor of a set of points is closer than the distributional mean. (b, c, d) However, we show that in higher dimensions the nearest neighbor distance increases quickly while the distributional mean stays relatively constant. This relation is only mildly affected by changes in the number of elements in the nearest neighbor pooling set ($n$).

# E   MECHANISTIC ANALYSIS

In this section, we perform a preliminary mechanistic analysis to provide further evidence for the claims from main paper. Specifically, we perform the following experiments.

- **Reconstructing Markov Chains from MLP Neurons in Retrieval Phases.** In Sec E.1, we attempt to directly retrieve the transition matrices of Markov chains used during training by analyzing the MLP weights of the model, demonstrating successful retrieval in low diversity settings.

- **Dynamics of Memorization.** In Sec E.2, we quantify the extent to which individual neurons memorize transition matrices by measuring the minimum KL divergence between neuron outputs and in-distribution transitions across training, showing greater memorization in retrieval phases.

- **Attention Maps: Aggregating Context Statistics.** In Sec E.3, we further visualize attention maps to infer the specific algorithms implemented by the model during different training phases.

- **Attention Maps' Evolution Corroborates LIA.** In Sec E.4, we investigate the evolution of attention maps throughout training, finding evidence for shifts in algorithms predicted by LIA.

## E.1   RECONSTRUCTING MARKOV CHAINS FROM MLP NEURONS IN RETRIEVAL PHASES

In Sec. 3.2, we claim that for certain experiment configurations, the model relies on a fuzzy retrieval approach to perform the finite mixture of Markov chains task. Specifically, this corresponds to the phases wherein the model is involved in the `Uni-Ret` and `Bi-Ret` algorithms. To provide further evidence towards these algorithms explaining model behavior in these phases, we try to reconstruct the Markov chains seen during training (denoted $\mathcal{T}_{\text{train}}$ in the main paper) from its internals—specifically, from the MLP neurons. This analysis builds on the approach for knowledge localization by Geva et al. (2021).

**Model Setup.** Based on our analysis in Sec. 4.1, we know that fully trained models with data diversities $N = 2^2$ and $N = 2^6$ should possess a mechanism that is behaviorally equivalent to the `Bi-Ret` algorithm; meanwhile, for $N = 2^{11}$, the model should implement something akin to the `Bi-Inf` solution. Ideally, this implies, we can reconstruct the Markov transition matrices used to generate training sequences directly from the model weights in the retrieval phases, whereas for inference-based phases we should yield less accurate (if any) reconstructions. Note the reconstructions may still be feasible in the inference-based phases since the model generally arrives at them after going through a retrieval-based phase.

**Approach.** In the following, we use the term **neuron** to refer to an entry in the second fully connected layer of an MLP of the second (i.e., last) Transformer block. For each neuron, we compute its next-token distribution (called **neuron output**) by applying the final LayerNorm, multiplying by the Unembedding matrix, and applying a Softmax. We then randomly sample a Markov chain seen during training, and compare the rows of its transition matrix to neuron outputs. We then select the neurons with the smallest KL divergence to form a row in the "reconstructed matrix".

**Results.** We provide visualizations of the reconstructions in Fig. 15. Note that the Markov chains are redefined for different experiments, and hence the transition matrices targeted for reconstruction are different for different experimental settings. To assess the accuracy of the reconstruction more quantitatively, we also report the *average KL divergence* between the reconstructed and the targeted ground truth transition matrix, averaging over 100 randomly selected rows from different transition matrices.[3] We also report the average KL between the stationary distributions of a randomnly sampled chain's transition matrix and its reconstruction (see Table 1). To contextualize these quantitative experiments, we report two baselines as well.

1. Random model's ability to reconstruct a transition matrix: We take a randomly initialized model and report the KL it achieves when trying to reconstruct a transition matrix.

2. Trained model's ability to reconstruct an unseen transition matrix: We take a trained model, sample an unseen transition matrix, and analyze whether we can reconstruct such an unseen

---

[3]Performing this analysis over all seen matrices can be prohibitively expensive, requiring analysis of as many as 20480 transitions, and hence we focus on only 100 randomly-sampled transitions instead of all seen matrices.

matrix. Since a random model's neurons are likely to be arbitrary in their weights, we believe this baseline offers a more meaningful comparison.

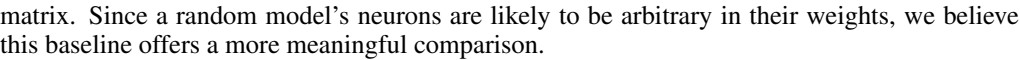
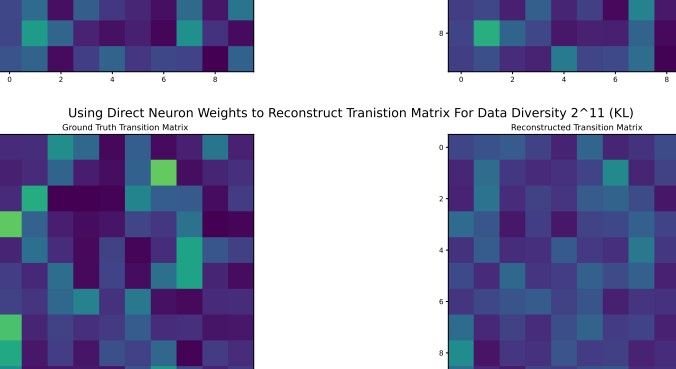

Figure 15: **Reconstructing Markov Chains from Neuron Weights.** We show that transitions from Markov chains seen in training (**left column**) can be directly reconstructed from neuron weights (**right column**). As expected, results are especially good in low data diversity settings (**top row**). Reconstructions in the medium data-diversity are structurally sound (more or less the same transitions have highest probability mass as ground truth), but the precise magnitudes are off (**middle row**). Meanwhile, the reconstructions for high data-diversity are quite poor, barely capturing the structure of the transition matrix and almost uniformly spread magnitudes (**bottom row**).

Table 1: Reconstruction similarities for ID and OOD transitions.

| Category | Avg KL | Stationary KL |
|---|---|---|
| **Random model** | 0.34 | 0.30 |
| **Seen chains** | | |
| $N = 2^2$ | 0.09 | 0.09 |
| $N = 2^6$ | 0.11 | 0.11 |
| $N = 2^{11}$ | 0.17 | 0.17 |
| **Unseen chains** | | |
| $N = 2^2$ | 0.17 | 0.16 |
| $N = 2^6$ | 0.13 | 0.13 |
| $N = 2^{11}$ | 0.18 | 0.22 |

### E.2 DYNAMICS OF MEMORIZATION

In the previous section, we posthoc analyze trained models to assess signatures of memorization. We now perform this analysis over time, *hence yielding the dynamics of memorization*.

**Approach.** To demonstrate that neurons store transition matrices as part of retrieval solutions, we evaluate their outputs for both in-distribution and out-of-distribution transitions. Specifically, we repeat the analysis from previous section across time: we randomly select 100 in-distribution state transitions from the training data (sampling chains and transitions with replacement) and compute the outputs of all neurons in the second MLP layer. For each transition, we identify the neuron with the minimum KL divergence from the target transition distribution and record the average minimum KL across all 100 transitions. This process is repeated at every training checkpoint.

**Baseline.** As a baseline, we apply the same procedure to 100 unseen transitions. Additionally, we create a second baseline by sampling 256 random vectors (matching the number of neurons in the MLP layer) from Dirichlet distributions with $\alpha = \mathbf{1}_k$ ($k = 10$). For each of these random vectors, we compute the minimum KL divergence with the 100 in-distribution transitions. This baseline assesses whether the observed neuron behaviors reflect structured learning of training-specific distributions or merely random mappings.

**Results.** As shown in Fig. 16, neurons closely align with the specific transitions observed during training in lower data diversity settings. This is especially the case when the model enters the `Bi-Ret` phase, where the KL from seen transitions starts to substantially diverge from that of unseen transitions. This pattern partially shows up for medium data diversity $N = 2^6$, consistent with LIA's prediction that the model continues to perform bigram retrieval. However, for large data diversity $N = 2^{11}$, the neurons show greater dissimilarity to the training transitions compared to random transitions, suggesting that the model is no longer memorizing transitions in this regime.

### E.2.1 SUPERPOSITION OF NEURONS WITH INCREASE IN DATA DIVERSITY

In Sec. 3, we claim models in high data diversity scenarios rely on inference-based algorithms. Consequently, as data-diversity increases and the ability to rely on a retrieval approach goes down, we hypothesize that predicting the correct next-token distribution likely requires a complex linear combination of neuron outputs. If this were not the case, we would need to scale the number of neurons in proportion with data diversity to witness a retrieval based solution. To circumvent this, the model then likely represents transitions across several neurons—akin to the superposition effects discussed by, e.g., Henighan et al. (2023); Elhage et al. (2022).

**Results.** To quantitatively investigate the hypothesis, we analyze neuron activity across different data diversity regimes. As shown in Fig 17a, the distribution of neuron activity is sharply concentrated in lower data diversity regimes and is smoother as data diversity increases (with the most active distribution observed at data diversity $2^{11}$). This supports our hypothesis that representations are distributed across neurons. In panel (b), we examine linear combinations of neuron outputs and their similarity to the ground truth transition. Neurons are first sorted by their GeLU activations on in-distribution samples. For the $k$-th linear combination, the top $k$ neuron outputs are aggregated, weighted by their respective GeLU activations, and summed to form a composite output. Our findings reveal that as data diversity increases, more neurons are required to reconstruct the ground truth transition. This suggests that higher data diversity leads the model to adopt more distributed representations, providing more support for our hypothesis.

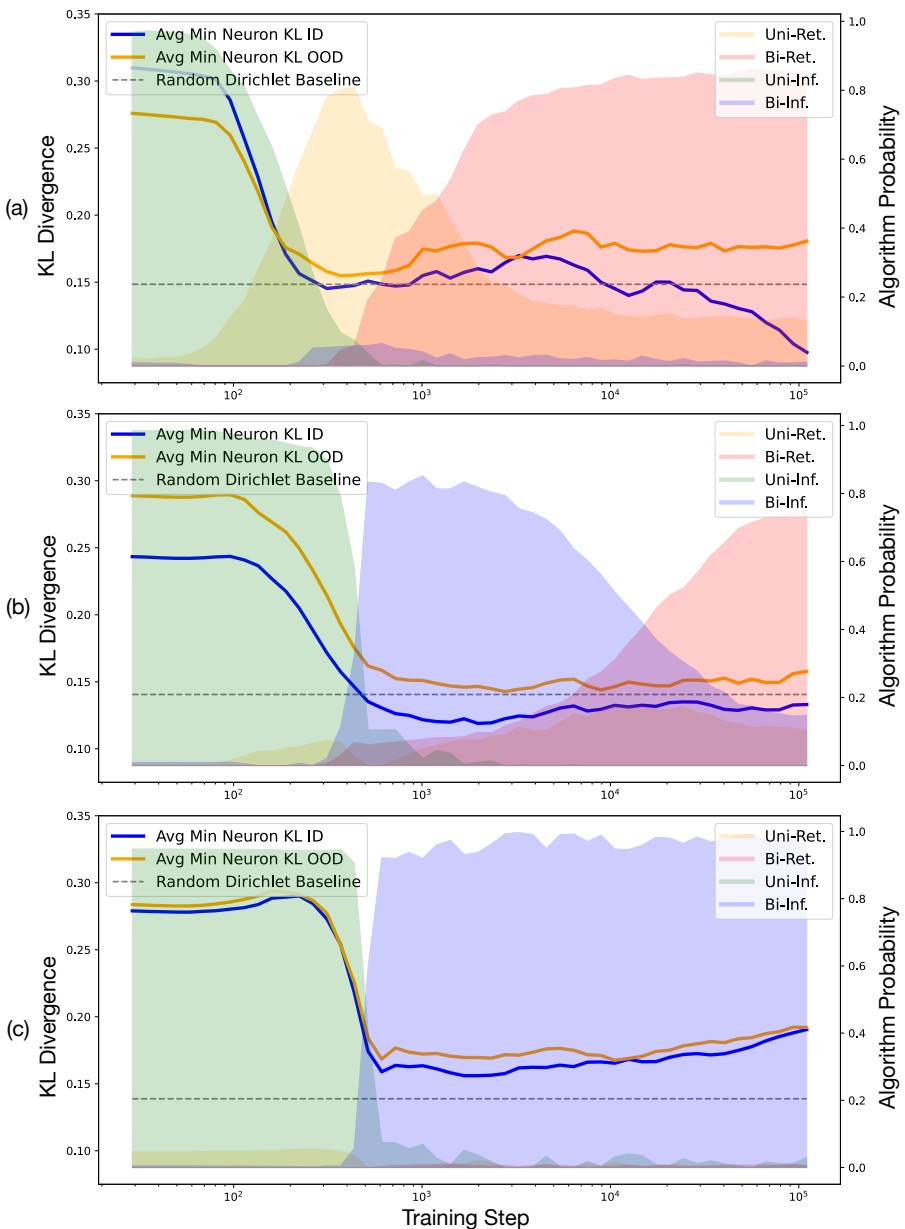

Figure 16: **Memorization as Training Progresses.** Minimum KL of transitions from training and neuron outputs, averaged across 100 randomly selected transitions. a) $N = 2^2$. b) $N = 2^6$. c) $N = 2^{11}$. Lower data diversity settings display much greater degree of similarity with training transitions than higher data diversity settings.

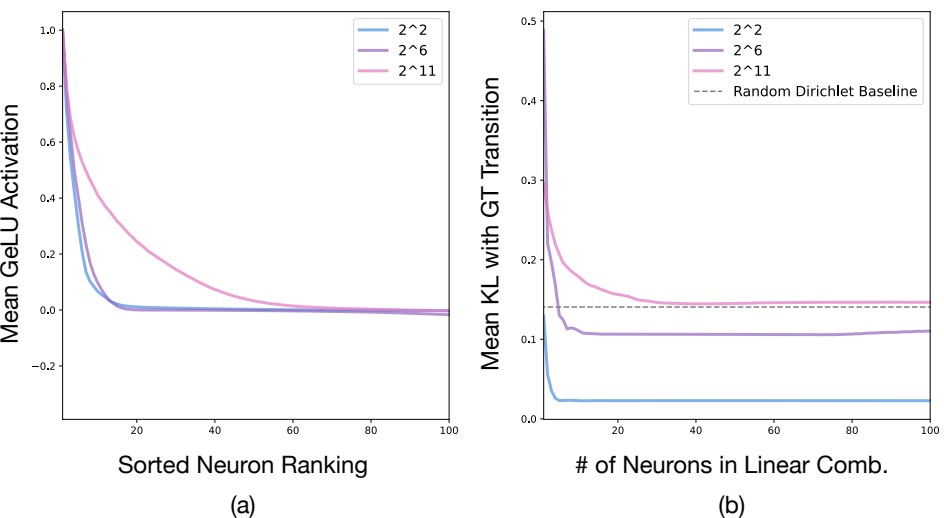

(a)  (b)

Figure 17: **Neural Activity Across Increasing Data Diversities.** a) Sorted GeLU activations of neurons, normalized by their maximum activation, are shown for increasing levels of data diversity. In the `Bi-Ret` regime, significantly fewer neurons are active compared to the `Bi-Inf` regime. b) Similarity of linear combinations of top neuron outputs to the ground truth transition. As data diversity increases, a larger number of neurons are required to reconstruct the correct ground truth transition.

### E.3 ATTENTION MAPS: IMPLEMENTING INFERENCE APPROACHES

We visualize attention maps to provide further evidence for our algorithmic solutions. Specifically, we train a model with a single attention head to avoid the ambiguity of choosing the most relevant attention head for model behavior or an algorithm being implemented over different heads. While in this section we focus on only a single value of data-diversity ($N = 2^6$), in App. E.4, we will show the effects of data-diversity as a function of number of iterations, contextualizing the results in light of different algorithmic phases to provide evidence in support of the validity of LIA.

**Analysis of first layer's attention head.** Fig. 18 shows the first layer attention matrix when a context sequence is input to the model. This model is trained using $N = 2^6$ training chains, and the dominant algorithm changes from `Uni-Inf` to `Bi-Inf` to `Bi-Ret`.

- Fig. 18 (a) shows that the attention map is uniform at model initialization.
- Fig. 18 (b) shows the attention pattern at step 189, corresponding to a `Uni-Inf` solution. Each position attends to most tokens in the context, enabling a frequency count. This is *consistent* with the hypothesis that the model is simply computing the unigram distribution.
- Fig. 18 (c) shows the attention pattern at step 855, corresponding to a `Bi-Inf` solution according to our analysis in Sec. 3. Each position attends almost only to the previous token. This is a characteristic of an induction head (Elhage et al., 2021; Olsson et al., 2022; Edelman et al., 2024; Akyürek et al., 2024) for copying or gathering statistic from a context. This is *consistent* with the hypothesis that the model is performing a statistical induction to enable computation of the bigram statistics, as theoretically characterized by Edelman et al. (2024).
- Fig. 18 (d) shows the attention pattern at step 110133, corresponding to a `Bi-Ret` solution. Each position now attends to the previous token *and* partially the current token. This is *consistent* with the hypothesis that this head is forwarding transitions to upper layers of the model, where upper layers are expected to count them.

**Analysis of second layer's attention head.** Fig. 19 shows the second layer attention maps for the same sequence input as in Fig. 18.

- Fig. 19 (a) shows the uniform attention at initialization.
- Fig. 19 (b) shows that the attention pattern for `Uni-Inf` is mostly uniform, again consistent with the hypothesis that it computes a histogram. Note that it is unclear as of yet how much of the histogram computation is delegated to layer 1 or 2.
- Fig. 19 (c) shows that the second layer attention pattern for the model performing `Bi-Inf` is *consistent* with our hypothesis. We draw a red cross wherever the query state matches the previous state of the key state. These red crosses precisely overlap where the attention pattern peaks: alongside the first layer head, this head combines to form the induction head as in Edelman et al. (2024).
- Fig. 19 (d) Interestingly, the second layer attention again becomes uniform as the model drifts from `Bi-Inf` to `Bi-Ret`. This is again consistent with the hypothesis that the first layer attention presents transitions into the residual stream so that the second layer attention can count these transitions.

By visualizing the attention patterns, we find that the observed patterns are *consistent* with how we would expect the 2 layer transformer model to implement the solutions in Sec. 3.1. While these results are strongly suggesting our algorithms are implemented, causality experiments (Nanda et al., 2023; Li et al., 2023a; Hazineh et al., 2023) or attention patching could help confirm these results in the future.

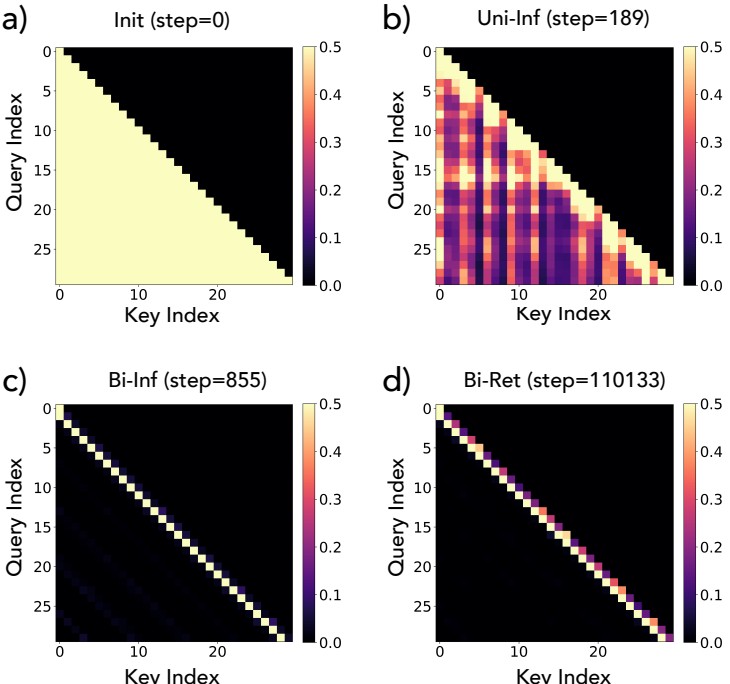

Figure 18: **First layer attention maps from the $N = 2^6$ run.** We visualize the first layer attention maps from different points in the checkpoint. Each query is normalized by the maximal value present in the row for visualization purposes. The title shows the number of training steps elapsed as well as the dominant strategy at the checkpoint. a) The attention map at initialization. The attention pattern is mostly uniform. b) The attention map during `Uni-Inf`. The attention map attends to most tokens in the context. c) The attention map during `Bi-Inf`. The attention pattern clearly implements an induction head, where the first layer attends to the previous token $x_{t-1}$ and thus provides potential to copy the current token $x_t$ given the same token as $x_{t-1}$ appears. d) The attention map during `Bi-Ret`. The attention pattern develops non-zero values for the diagonal entries. This suggests that this head might be combining information from two subsequent tokens for it to be available for the next layer to use. We trained a model with a single attention head to avoid the ambiguity of choosing the most relevant attention head for model behavior or an algorithm being implemented over different heads.

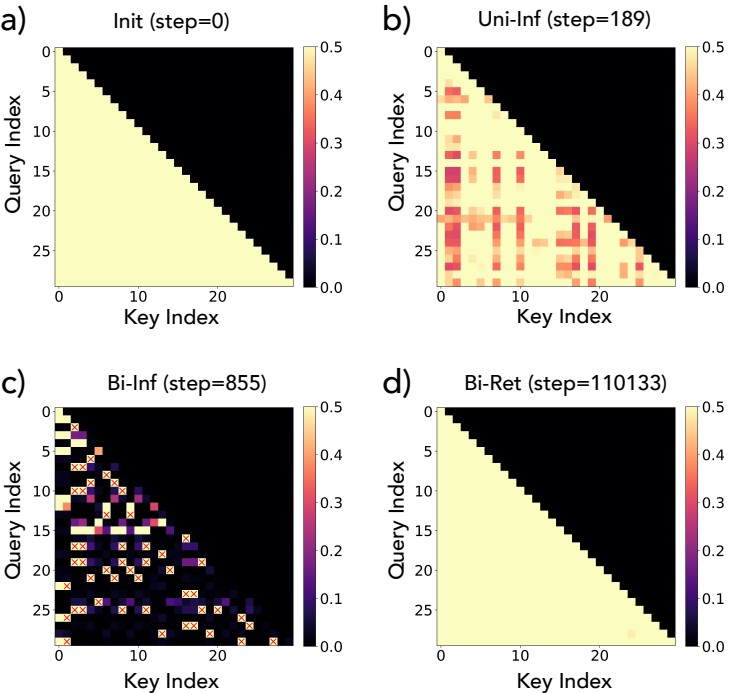

Figure 19: **Second layer attention maps from the** $N = 2^6$ **run.** We visualize the second layer attention maps from different points in the checkpoint. Each query is normalized by the maximal value present in the row for visualization purposes. The title shows the number of training steps elapsed as well as the dominant strategy at the checkpoint. a) The attention map at initialization. The attention pattern is mostly uniform. b) The attention map during `Uni-Inf`. The attention map attends to most tokens in the context. c) The attention map during `Bi-Inf`. The red crosses are locations where we find that the query state matches the previous state of the key state. We find that the attention pattern clearly implements this logic. d) The attention map during `Bi-Ret`. The attention pattern is again mostly uniform.

### E.4 ATTENTION PATTERN EVOLUTION CORROBORATES LIA

Our analysis of MLP neurons in App. E.1, E.2 helps provide mechanistic evidence in support of retrieval-based approaches by demonstrating the memorization of Markov chains seen during training. Meanwhile, the analysis in App. E.3 demonstrates different attention head patterns that provide evidence for unigram vs. bigram strategies, especially ones that finally aid an inference-based algorithm. These analyses thus demonstrate the model possesses components necessary for implementing four of our identified algorithms. We now contextualize these results by analyzing the evolution of attention heads for different data-diversity values as the model makes its way through different algorithmic phases, providing evidence for LIA's validity. To this end, we first plot results at points where the model is in different algorithmic phases, and then when the model is at the phase boundaries, where, if LIA is an accurate methodology, we would expect a simple linear interpolation of attention maps from different phases would approximately match the actual attention head retrieved from running a forward pass on the model.

### E.4.1 VALIDATING LIA: ATTENTION HEADS IN DIFFERENT PHASES

Figs. 20, 21, and 22 show attention maps for different data diversities and different checkpoints—specifically, checkpoints that correspond to particular phases in the learning dynamics. As we can see, LIA effectively predicts algorithmic shifts in the model across training steps and data diversities. For $N = 2^6$, the model transitions from an induction head configuration to a bigram statistical pattern, consistent with the prediction that it relies on an inference-based solution (Fig. 21). Similarly, for $N = 2^2$, the model begins using bigram statistics only when in the bigram retrieval state (Fig. 20). Finally, for $N = 2^{11}$, the model demonstrates induction behavior in the bigram inference state, which was not evident earlier (Fig. 22).

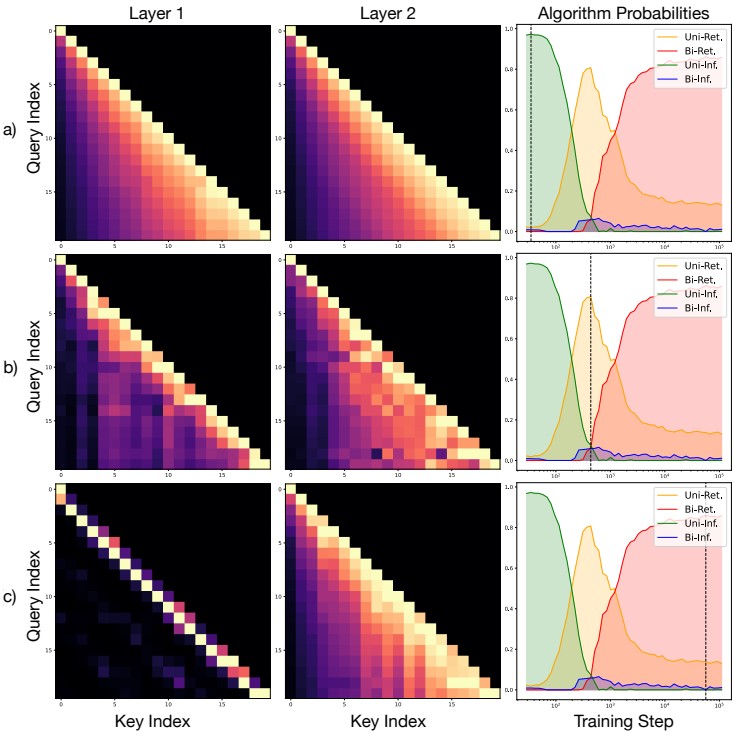

Figure 20: **Attention Maps At Key Checkpoints Across Training For** $N = 2^2$**.** a) Training checkpoint with maximal likelihood of `Uni-Inf` according to LIA. b) Training checkpoint with maximal likelihood of `Uni-Ret` according to LIA. c) Training checkpoint with maximal likelihood of `Bi-Ret` according to LIA.

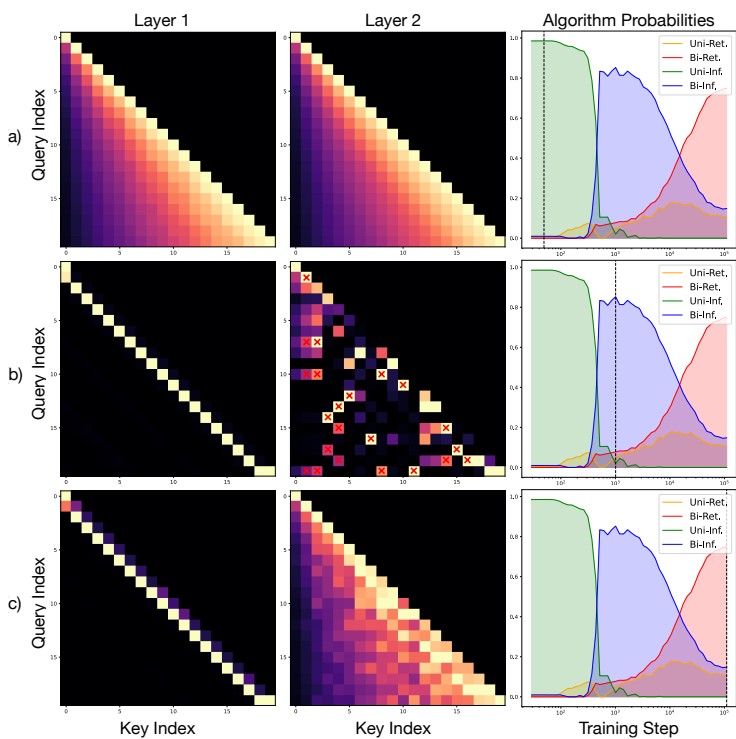

Figure 21: **Attention Maps At Key Checkpoints Across Training For** $N = 2^6$**.** a) Training checkpoint with maximal likelihood of `Uni-Inf` according to LIA. b) Training checkpoint with maximal likelihood of `Bi-Inf` according to LIA. c) Training checkpoint with maximal likelihood of `Bi-Ret` according to LIA. A red $\times$ indicates the position after previous occurrences of the current tokens during `Bi-Inf`.

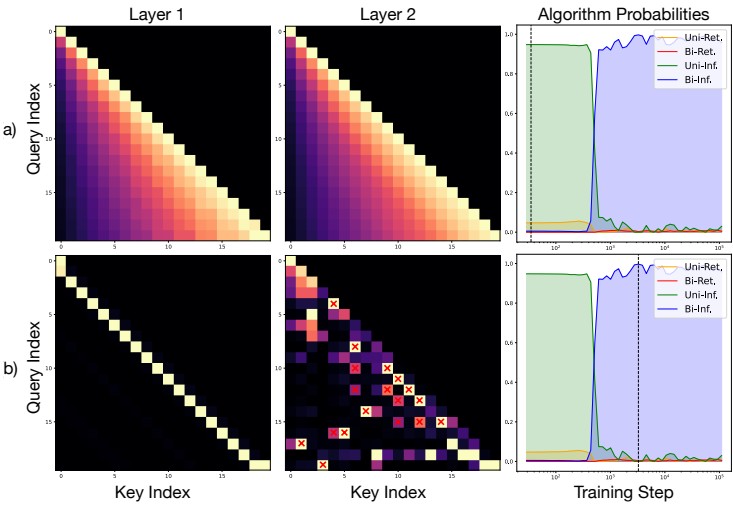

Figure 22: **Attention Maps At Key Checkpoints Across Training For** $N = 2^{11}$**.** a) Training checkpoint with maximal likelihood of `Uni-Inf` according to LIA. b) Training checkpoint with maximal likelihood of `Bi-Inf` according to LIA. A red $\times$ indicates the position after previous occurrences of the current tokens during `Bi-Inf`.

### E.4.2 VALIDATING LIA: ATTENTION HEADS AT PHASE BOUNDARIES

We next want to explore the model's mechanisms when LIA predicts the dominant algorithm is changing. Figs. 23, 24, and 25 show that the attention patterns during these states appear to interpolate between those observed when LIA predicts each algorithm as the most likely. For instance, during the transition from `Bi-Inf` to `Bi-Ret`, the attention pattern resembles an interpolation of the two.

To validate this observation, we manually interpolate between the attention patterns of the last two algorithms the model implements during training across data regimes. As shown in Fig 26, the interpolated attention maps are highly correlated with the ground truth attention maps, supporting the hypothesis that the model transitions between the algorithms at this point.

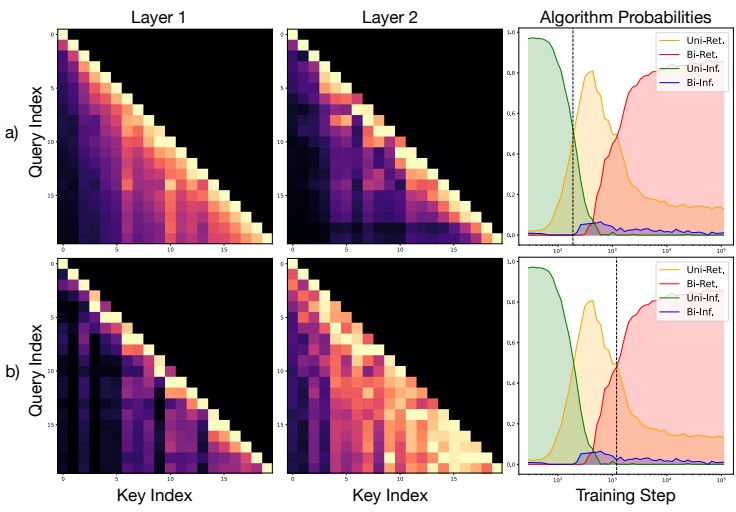

Figure 23: **Attention Maps At Predicted Transition Points For** $N = 2^2$. a) First predicted transition checkpoint according to LIA. b) Second predicted transition checkpoint according to LIA.

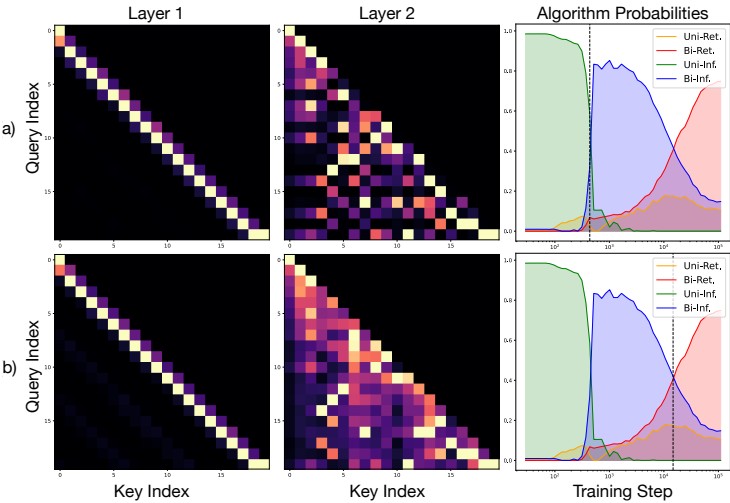

Figure 24: **Attention Maps At Predicted Transition Points For** $N = 2^6$. a) First predicted transition checkpoint according to LIA. b) Second predicted transition checkpoint according to LIA.

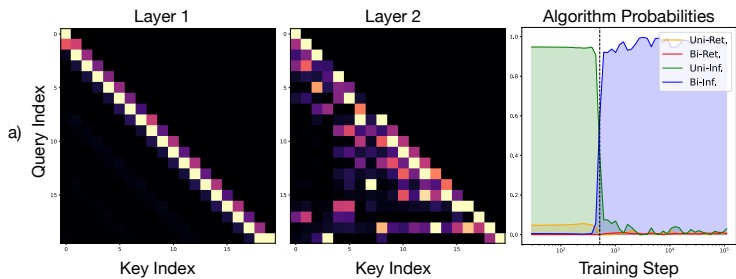

Figure 25: **Attention Maps At Predicted Transition Points Across Training For** $N = 2^{11}$.

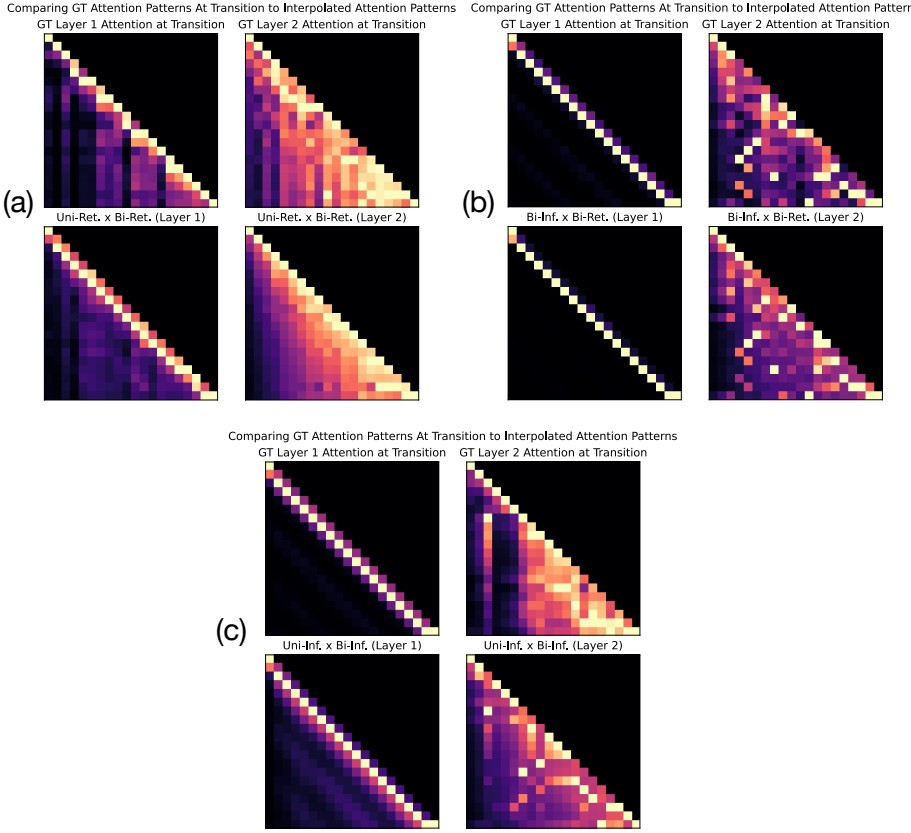

Figure 26: **Attention Interpolation.** Manually interpolating between attention patterns corresponding to different algorithms closely matches the attention patterns observed during training when the model is predicted to be transitioning between these algorithms. This supports the hypothesis that the model is implementing these two algorithms during training. (a) $N = 2^2$ (b) $N = 2^6$ (c) $N = 2^{11}$

# F ADDITIONAL RESULTS

## F.1 ADDITIONAL PLOTS FROM THE MAIN EXPERIMENTS

In this section, we visualize our main experiments in a more detailed manner. Specifically, we first show a high quality version of the subplots in Fig. 1. Next, we show 2D and 3D heatmaps of KL divergence on OOD vs ID sequences. Third, we show different slices our of experiment to show the evolution of the KL divergence as we change the optimization steps, data diversity, and context length independently. Finally, we directly evaluate the KL divergence between the model and each solution to verify our findings in Sec. 3.2 and Sec. 4.1.

### F.1.1 HIGH QUALITY VERSION OF FIG. 1 SUBPLOTS

We reproduce the small subplots of Fig. 1 here in Fig. 27. The color of every subplot corresponds to their algorithmic phases, except (d), where the color represents OOD KL divergence. Each plot has other fixed axes annotated. These 6 subplots corresponds to the 6 phenomena discussed in App. C.1.

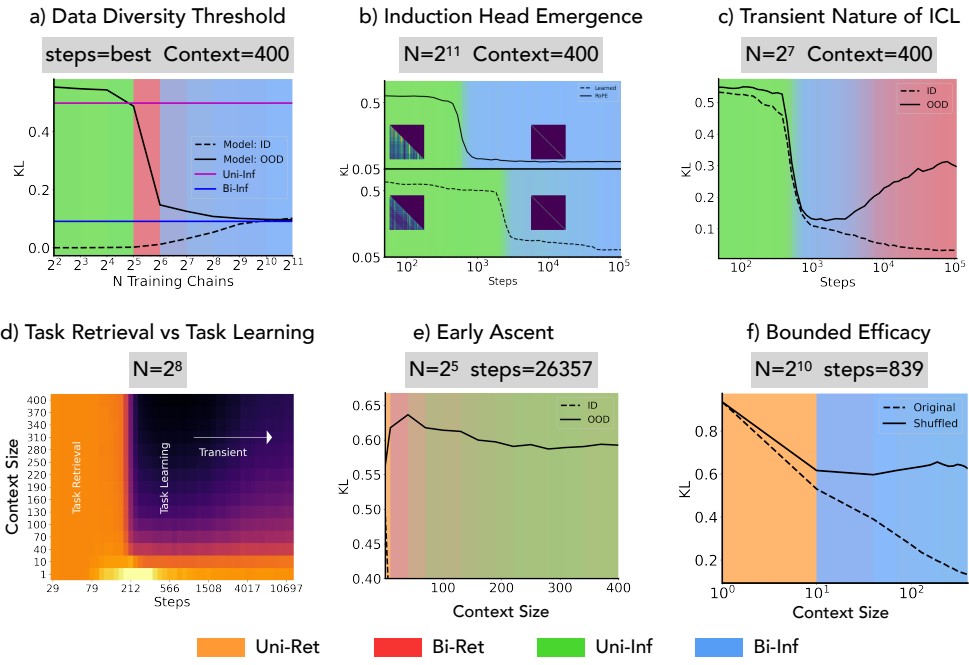

Figure 27: **Subplots of Figure 1** We reproduce the subplots of Figure 1 here for visibility. a) Data diversity threshold (Raventós et al., 2023); b) Emergence of induction heads (Edelman et al., 2024); c) Transient nature (Singh et al., 2023; Anand et al., 2024; Panwar et al., 2024); d) Task retrieval and task learning phases (Min et al., 2022); e) Early ascent of risk (Xie et al., 2021); and f) Bounded efficacy (Lin & Lee, 2024).

Fig. 27 (a). We reproduce the data diversity threshold (Raventós et al., 2023), where a certain number of task (here $2^6$) is required for the model to learn an optimally OOD-generalizing ICL solution (`Bi-Inf`). Note that we selected the checkpoint with the best OOD performance throughout training.

Fig. 27 (b). We reproduce the emergence of induction heads (Edelman et al., 2024), where the model transitions from `Uni-Inf` to `Bi-Inf`.

Fig. 27 (c). We reproduce the transient nature of ICL (Singh et al., 2023; Hoogland et al., 2024), where an OOD-generalizing solution (`Bi-Inf`) is learned, but disappears with more training as a retrieval solution (`Bi-Ret`) slowly replaces it.

Fig. 27 (d). We reproduce the findings on Task Retrieval vs Task Learning. Please see App. C.1 for a detailed discussion.

Fig. 27 (e). We reproduce the *Early Ascent* phenomena where a small amount of exemplars harm the ICL performance. This has been observed in Xie et al. (2021). Our explanation is from the phase diagram. The model implements `Bi-Ret` for small context length and this causes a wrong task retrieval, as explained in App. D. This lost performance is recovered as the model performs `Uni-Inf` when more context is given.

Fig. 27 (f). We reproduce bounded efficacy shown in (Min et al., 2022; Lin & Lee, 2024). When we shuffle labels, an analogy to the perturbation in Min et al. (2022), the OOD KL first decreases since it finds roughly the right algorithm via `Uni-Ret`. However with more context the `Bi-Inf` strategy kicks in, and as the model learns transitions from a wrong context, the KL does not improve (and in fact slightly increases). This is similar to the explanation of bounded efficacy in Lin & Lee (2024).

### F.1.2 KL DIVERGENCE HEATMAPS

In Fig. 28, we show a 3D plot of the KL divergence for ID sequences and OOD sequences. We find that the ID KL always decreases with less data diversity (since the task is simpler), more context (more information), and with more training. OOD KL divergence, on the other hand, shows non-monotonicity in two axes: steps and context. Especially, with more optimization steps the OOD KL divergence shows up-down non-monotonicity ($2^4$, similar to double descent), down-up non-monotonicity ($n = 2^6$, transient ICL) and up-down-up non-monotonicity ($n = 2^3$, explained by 3 mechanisms).

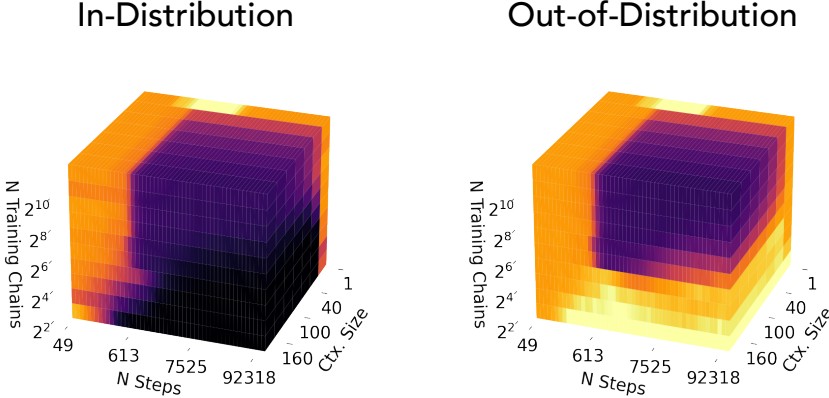

Figure 28: **3D heatmap of ID/OOD KL divergence.** (left) KL divergence on ID sequences. (right) KL divergence on OOD sequences.

In Fig. 29 (a), we show the ID KL corresponding to Fig. 3. We find that the ID KL divergence drops with the induction head formation for high data diversity, but drops earlier for lower data diversity. We explain this by a `Uni-Ret` solution developing for small data diversity, as seen in Sec. 4.1. Fig. 29 (b) is equivalent to Fig. 3 (a). Fig. 29 (c) shows the difference between the ID KL and the OOD KL, clearly highlighting regions where the model follows a retrieval approach.

Fig. 30 illustrates the ID and OOD KL divergence depending on context size and optimization steps for $N = 2^4$, a low data diversity. Fig. 30 (a) shows that the ID KL divergence always decreases with more context and more training. Fig. 30 (b) shows that unlike the ID KL, the OOD KL has an ascent and decent in both optimization steps and context size. As we increase the concext size at step=26357, we find that the KL divergence first rises before falling. This is precisely the *early ascent* phenomenon Xie et al. (2021). Fig. 30 (c) shows the difference between the ID KL and the OOD KL, clearly highlighting regions where the model follows a retrieval approach.

Fig. 31 illustrates the ID and OOD KL divergence depending on context size and optimization steps for $N = 2^6$, a medium data diversity. Fig. 31 (a) shows that the ID KL divergence again always decreases with more context and more training. The only qualitative difference from Fig. 30 (a) is

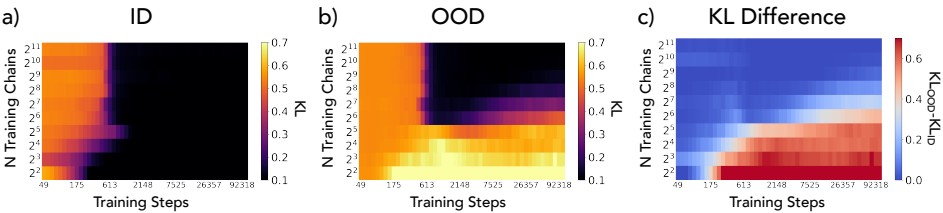

Figure 29: **KL divergence depending on data diversity and optimization.** a) ID KL divergence. b) OOD KL divergence. c) Excess KL divergence on OOD sequences.

$$N = 2^4$$

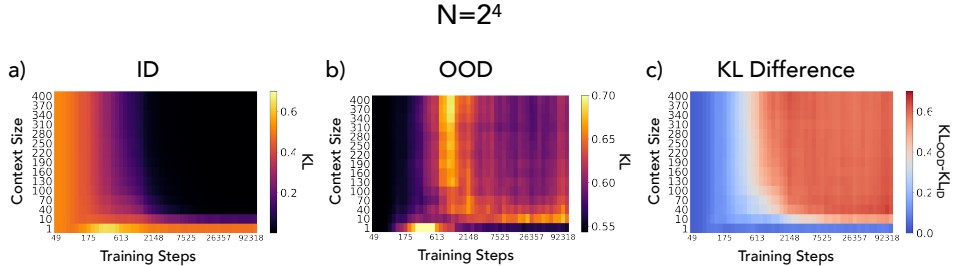

Figure 30: **KL divergence depending on context size and optimization steps for** $N = 2^4$. a) ID KL divergence. b) OOD KL divergence. c) Excess KL divergence on OOD sequences.

that there is a much sharper evolution near step $\sim 6 \times 10^2$. Fig. 31 (b) shows that the OOD KL shares this drop, but only to increase KL again as the training proceeds. This is exactly the transient nature of ICL revisited. Fig. 31 (c) shows the difference between the ID KL and the OOD KL, highlighting regions where the model follows a retrieval approach.

$$N = 2^6$$

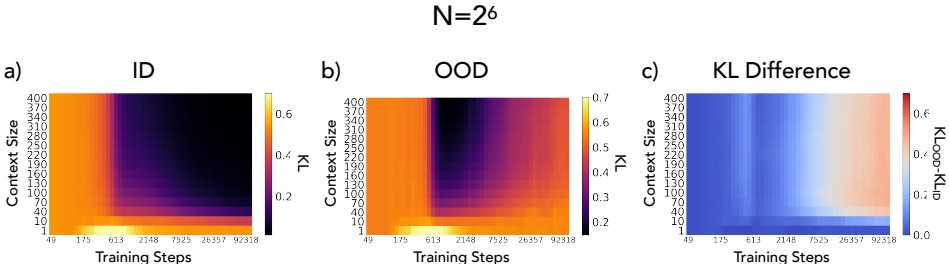

Figure 31: **KL divergence depending on context size and optimization steps for** $N = 2^6$. a) ID KL divergence. b) OOD KL divergence. c) Excess KL divergence on OOD sequences.

### F.1.3 KL DIVERGENCE VS. {STEPS, DIVERSITY, CONTEXT}

Recall that we showed model's ID and OOD KL divergence vs. $N$ at 839 steps of training in Sec. 2.1 (Fig. 3 (b)). This specific value was used as a representative setting as it clearly demonstrates the data diversity threshold "inference" solutions to emerge. For completeness, we now show similar plots for different number of training steps Fig. 32. Additionally, we plot the four different solutions' ID and OOD KL divergence as well. We can clearly observe the transition from `Uni-Inf` to `Bi-Inf` for high $N$, while, at low $N$, the ID and OOD KL divergences split because of retrieval solutions.

Recall also that we showed the model's ID and OOD KL divergence for $N = 2^7$ in Sec. 2.1 (Fig. 3 (c)). This value was chosen as a representative setting since it clearly demonstrates the transient nature of ICL. However, for completeness, we show similar plots for many different values of in Fig. 33. Again, we plot the four different solutions' ID and OOD KL divergence as well. We

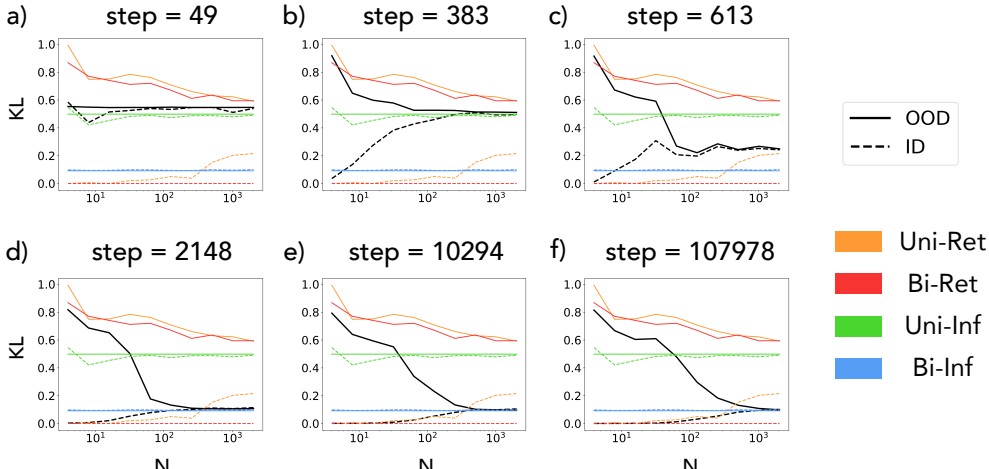

Figure 32: **KL vs. $N$ at fixed step.** We show the effect of data diversity at fixed number of gradient steps. We plot the model's KL divergence with respect to ground truth for both ID and OOD chains in black. We also plot the ID and OOD KL for each of the 4 solutions. Note that while the solutions' expected KL divergence is, in theory, smooth, we estimate them numerically from Eq. 5 and Eq. 6 over 30 transition matrices, thus resulting in some noise. In each plot, the number of training steps is fixed, as denoted in the title, and the KL divergence is averaged over a context size of 400 and over 30 random transition matrices.

can clearly see the emergence of `Bi-Inf` and the transient nature driven by `Uni-Ret` or `Bi-Ret`. Notable phenomena we observe are as follows.

1. Panel (a): We observe a very robust `Bi-Ret` solution for $N = 2^2$, a very low data diversity.

2. Panel (c): We observe a highly non-monotonic OOD KL divergence for $N = 2^5$, a medium data diversity.

3. Panels (d, e): We observe at high data diversity $N = 2^6, 2^8$ the transient nature of ICL. Specifically, the model first finds the `Bi-Inf` solution and then moves to a retrieval solution which harms the OOD KL.

4. Panel (f): At a very high data diversity ($N = 2^{11}$), we observe that the `Bi-Ret` solution does not show up (at least not noticeably) within the compute budget.

We show both the ID and OOD KL divergence of the model and different solutions at fixed $N$, step, and across context length in Fig. 34. Each subplot is marked on the phase diagram. We find that, as expected, the model's KL divergence, both ID and OOD, follows that of the dominant solution in each phase. Notably, the agreement with `Bi-Inf` is very strong.

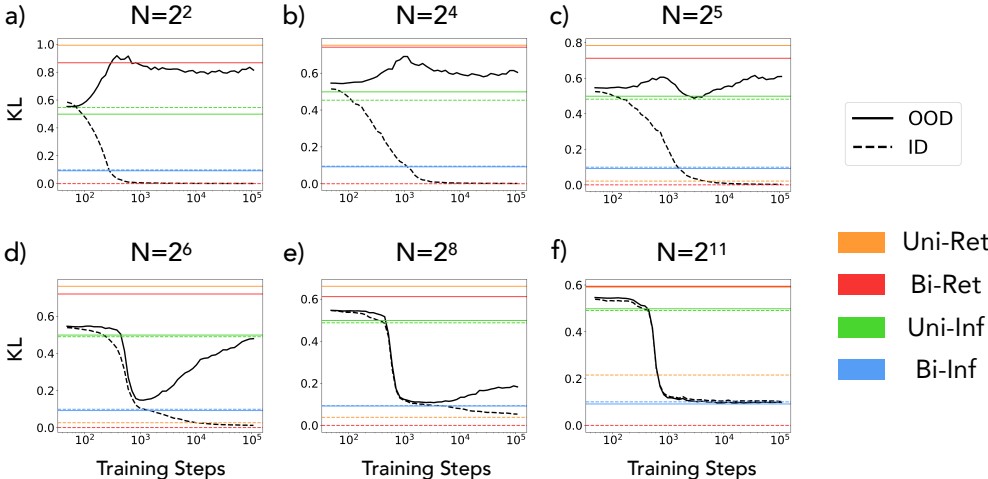

Figure 33: **KL vs. Steps at fixed $N$.** We show the effect of optimization at each fixed data diversity, $N$. We plot the model's KL divergence with respect to ground truth for both ID and OOD chains in black. We also plot the ID and OOD KL for each of the 4 solutions in horizontal colored lines. In each plot, the data diversity $N$ is fixed, as denoted in the title, and the KL divergence is averaged over a context size of 400 and over 30 random transition matrices.

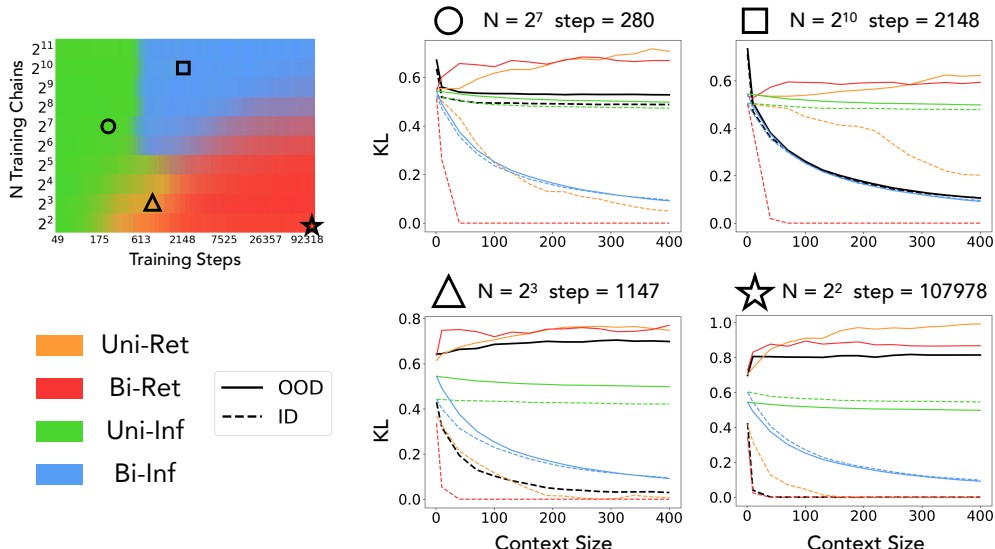

Figure 34: **KL vs. context size at fixed $N$ and step.** We show the effect of context size at fixed data diversity $N$ and gradient steps. We plot the model's KL divergence with respect to ground truth for both ID and OOD chains. We also plot the ID and OOD KL for each of the 4 solutions. As explained in Fig. 32, the solution KLs have a noise contribution. In each plot, the data diversity $N$ and the number of training steps are fixed, as denoted in the title, and the KL divergence is averaged over 30 random transition matrices. In the left side of the figure, we show where in the phase diagram each of these plots live in.

### F.1.4 KL DIVERGENCE BETWEEN THE MODEL AND SOLUTIONS

In Fig. 35, we show the KL divergence between the algorithmic solutions and the model estimated transition matrix. First, we find that $\mathrm{KL}\big(\hat{T}_{\mathrm{Model}}||\hat{T}_{\mathrm{Solution}}\big)$ quantified in Fig. 5 and $\mathrm{KL}\big(\hat{T}_{\mathrm{Solution}}||\hat{T}_{\mathrm{Model}}\big)$ in Fig. 35 are very similar. As discussed in Sec. 3.2, we again find that all four solutions indeed show a small KL divergence from the model exactly where we expect them to from Fig. 5 (c). These findings further confirms our findings in Sec. 3.2 and Sec. 4.1 are indeed accurate characterizations of the model's behavior.

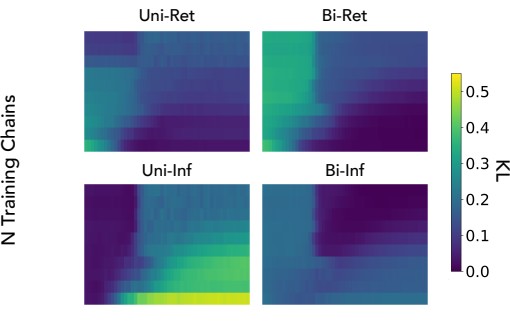

Figure 35: **KL divergence between algorithmic solutions and the model.** We quantify the KL divergence of each algorithmic solution from the model estimate of the transition matrix, across optimization steps and data diversity. We find nearly identical low KL regions as in Fig. 5 (d). The axes are same as in Fig. 5.

## F.2 Architecture Changes

Similarly to our analysis in Sec. 4.3, we analyze the effect of different architectural changes to the algorithmic phase diagram. These results are shown in Fig. 36.

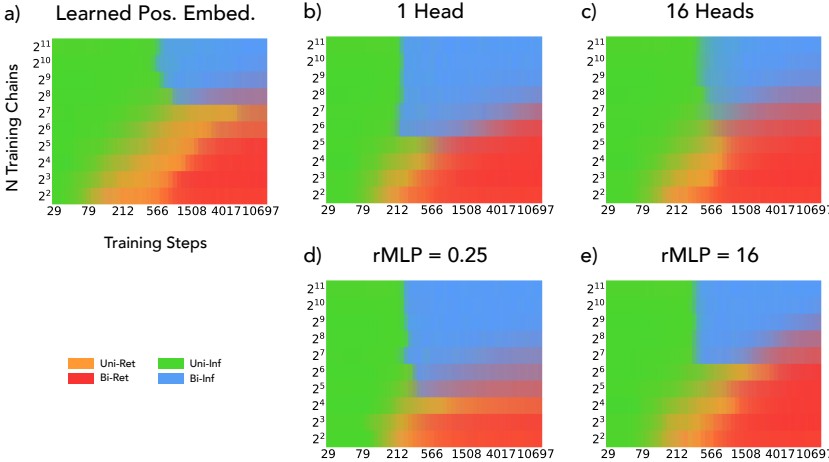

Figure 36: **Algorithmic Phase diagrams for different architectures.** Results with a Transformer using a) learned positional embeddings; b) a single attention head; c) using 16 attention heads; d) using an MLP fan-out ratio of 0.25; and e) using an MLP fan-out ratio of 16.

Fig. 36 (a) shows results when using a learned positional embedding. A learned positional embedding significantly biases the learning dynamics towards the `Uni-Ret` algorithm. We suspect that this is because the formation of a precise induction head for `Bi-Ret` (See App. E.3) is more difficult with learned positional embeddings without passing through an intermediate `Uni-Ret` solution. As a result, the *observed* data diversity threshold is increased from $2^6$ to $2^8$.

Fig. 36 (b,c) shows the phase diagram when using, respectively, a single head and 16 heads. We originally expected that more heads would be able to support more solutions simultaneously and a single head will yield sharper transitions. However this did not result in a significant phase diagram change. We speculate that multiple algorithms can exist within one head by dividing up the residual stream's subspace (similar to the argument by Elhage et al. (2021); Olsson et al. (2022)). An interesting future direction is to mechanistically analyze the checkpoints after the emergence of `Bi-Inf` but where `Bi-Ret` dominated to find out if the `Bi-Inf` circuit still exists but is unused.

Fig. 36 (d,e) shows the phase diagram when changing the MLP fan-out ratio (from the default of 4) to 0.25 or 16. Our prior hypothesis was that MLP layers carry memorization (Geva et al., 2021). This experiment validated our findings. Reducing the MLP layers hidden dimension width significantly suppressed `Uni-Ret` and `Bi-Ret`, resulting in `Bi-Inf` available with $2^5$ chains only. On the other hand, increasing the hidden dimension to 16 caused the run with $N = 2^6$ to never show `Bi-Inf`, as the "memorizing" retrieval solutions are promoted.

## G    GENERALIZATION TO CHAINS FROM A DIFFERENT PRIOR

In the main experiments, we generally drew the test matrix $T^*$ from the same Dirichlet prior generating the training set. As discussed in App. A.1 and App. D, this evaluation is already OOD, since the Dirichlet prior is only shared between the matrices and not the sequences themselves. However, here we verify that our OOD-generalizing solutions can indeed go far out of their training distributions. In particular, we design a context which is highly unlikely to be drawn from a Dirichlet prior and evaluate different model checkpoints.

Fig. 37 illustrates these results. We fed in a context consisting of a repeating pattern of $0, 1, 2$ and we generate a sequence from the model. We generate the sequence with zero temperature, i.e., select the state with the highest predicted probability. We visualize the next token probability as a pixel value intensities. For a model implementing mostly the `Uni-Inf` strategy, we find, as expected, a stationary distribution not depending on the last token, and thus not evolving. However, when the model is in the `Bi-Inf` phase, we find that the model can learn in-context to generate this totally out-of-distribution chain. Finally, when the model is implementing mostly `Bi-Ret`, the model implements a completely different chain, which is the training set chain having the most similar transitions. The model now generates a different sequence. We find that the non-linear evolution of accuracy is reproduced here again, as we see the next token accuracy to rise to 100% when `Bi-Inf` is implemented only to fall back to 1/3 when `Bi-Ret` takes over.

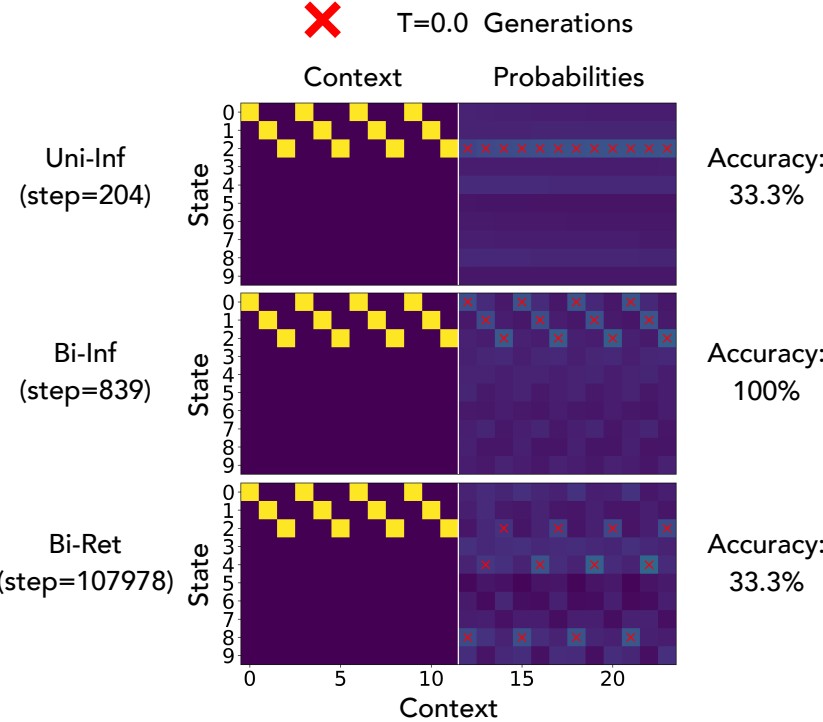

Figure 37: **Testing a model's ability to generalize far out of the training distribution.** We feed a completely out of distribution sequence to the model that consists of repeats of $0, 1, 2$. The left side shows the sequence and the right side shows the output probabilities for different settings. We generate tokens with temperature T=0.0, and continue to show next-token probabilities conditioned on the generated tokens. **(Top)** We find that the model in the `Uni-Inf` dominant phase indeed implements a unigram solution. **(Middle)** We find that the model in the `Bi-Inf` dominant phase learns the transition pattern in the context successfully to generate an adequate sequence. The accuracy rises accordingly. **(Bottom)** As the pre-training phase is continued, the `Bi-Inf` solution gets wiped out by the `Bi-Ret` solution as seen in Sec. 4.1. We find that the model now selects the best Markov chain in the training data, and thus generates a different sequence, resulting in an accuracy drop.

# H IN-CONTEXT LEARNING OF MARKOV CHAINS IN REAL LARGE LANGUAGE MODELS

Real language models trained on complex patterns including human languages, code and structured documents are known to acquire general in context learning abilities (Brown et al., 2020). Thus, we expect these models to deliver non-trivial performance at learning to continue a Markov sequence given in-context even though they haven't been explicitly trained on this task. Furthermore, Wei et al. (2023) showed that larger language models "do in-context learning differently", rougly speaking, in a more context-dependent way. To this end (and also motivated by curiosity) we evaluate the ICL ability of Llama-3.1 (Touvron et al., 2023; Dubey et al., 2024) across model scales: 8B, 70B and 405B. Since LLMs are known to be prompt sensitive (Sclar et al., 2024), we test three different prompts which differ by how precisely the Markovian nature of the sequence is described. We use NNsight and NDIF (Fiotto-Kaufman et al., 2024) to run these experiments.

Fig. 38 illustrates the results of inputting a sequence from a Markov chain as tokens of digits from 0 to 9. More precisely, we drew a transition matrix $T^*$ from the DGP in App. A.1, and generated 300 sequences to collect next token probabilities to construct $\hat{T}$ as in App. A.3. We used a separator token, ",", and we constrained the outputs to the digit tokens.

In Fig. 38 (left) we used a simple prompt without mentioning "Markov process". Interestingly, we found a performance in between `Uni-Inf` and `Bi-Inf`, demonstrating Llama is able to model the sequence better than purely predicting a random probability from the stationary distribution (histogram), without even being instructed that the sequence is Markovian. Furthermore, bigger models (405B > 70B > 8B) are able to model the Markov sequence better (lower KL).

Fig. 38 (center, right) respectively used a prompt which mentions the process is Markovian and a prompt which additionally and explicitly states that next state probabilities only depends on the last state. We find a significant KL divergence improvement on the $70B$ model when adding these specifications, revealing that LLMs can, to some extent parse a natural language instruction and reflect it into its sequence modeling process.

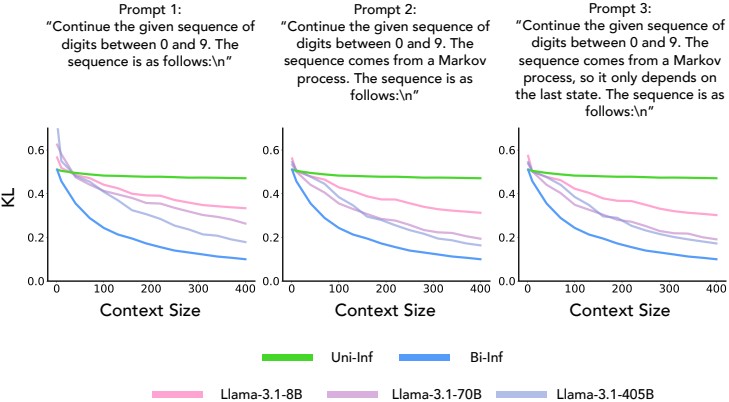

Figure 38: **In-context learning Markov chains on Llama-3.1.** We input sequences generated from the DGP in App. A.1 into LLama-3.1 8B, 70B and 405B and measure the KL divergence on the next state probability between the model predictions and the ground truth transition matrix. Each state is tokenized into a single token, from "0" to "9". The tokens are separated by the "," token to avoid the tokenizer merging states together. The output space was constrained to the state tokens. (left) Prompt 1, not explicitly stating the Markovian nature of the sequence. (center) Prompt 2, stating the Markovian nature of the sequence. (right) Prompt 3, additionally stating that the next state probability will only depend on the last state.

An interesting future direction would be to reverse engineer what algorithmic solutions are implemented by these models to generate the next state, e.g. by decomposing the logit outputs as in Sec. 4.1.

# I FURTHER VALIDATION OF LIA

## I.1 DOES LIA FIT THE MODEL WELL?

Fig. 6 and Fig. 7 showed that the linear interpolation of algorithms from Sec. 3 (i.e., LIA) achieves similar performance as the trained model: the empirical transition matrix identified from either systems yields similar ID KL, and in fact LIA predicts the trained model's OOD performance fairly accurately. Here, we show that in fact both the trained model and LIA produce similar predictions by comparing them to each other.

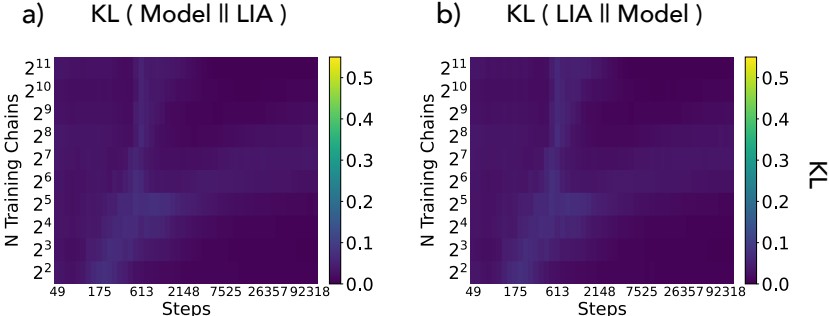

Figure 39: **Residual L2 of LIA fit.** We show the residual probability space L2 (the argument of $\arg\min$ in Eq. 7) for the LIA fit shown in Fig. 6. Note the color-scale, which is much smaller than the order of magnitude of probability vectors, which is unity.

Figure 40: **KL divergence between the model and LIA fit.** **(Left)** KL divergence between model predictions and LIA across data diversity and optimization. **(Right)** KL divergence between LIA and model predictions across data diversity and optimization.

**L2 distance between model and LIA predictions.** The L2 or Euclidean distance between the trained model and LIA predictions defines the optimization problem used to define weights in LIA (see the argument of $\arg\min$ in Eq. 7). To demonstrate LIA identifies an interpolation of algorithms that in fact does minimize this distance, we report the residual (i.e., excess error) from this optimization problem. Results are shown in Fig. 39 across different settings of data diversity and optimization. We clearly see that the residual is consistently low across all settings—specifically, on the order of $10^{-3}$. Note that since the target of this optimization problem is a probability vector (i.e., its elements sum to 1), this low order of an error clearly demonstrates LIA solves the problem almost perfectly in most settings. Moreover, we note that the landscape of this residual error is quite intriguing: the slightly high error (which is still very low in an absolute sense; around $\sim 0.005$) occurs at the phase boundaries! This suggest that the models very slightly deviate from a linear combination during transitions, but the nonlinear effects are relatively ignorable.

**KL between model and LIA predictions.** We also show the KL divergence of the model predictions to the ones predicted by LIA (and vice-versa) in Fig. 40. While Fig. 6 is already suggestive that LIA fits the model's predictions accurately, these KL divergence plots give yet another quantitative validation of the results alongside our L2 / Euclidean distance plots in Fig. 39. In particular, we again find the KL is very low in regions we ascribed to precise phases in our analysis in Sec. 3; meanwhile, the phase boundaries have some (very minimal) deviation from linearity.

## I.2 ROBUSTNESS OF LIA TO ARBITRARY ALGORITHMS

**Does LIA give rises to phases even when the solutions are arbitrary?** In Sec. 4.1, we used Linear Interpolation of Algorithms (LIA) to demonstrate that we can decompose the model's next token probability into a linear combination of the probability predicted from the four algorithms. Here we conduct two simple experiments to assess the robustness of these findings. In Fig. 41 we perform LIA for the same model used in Fig. 6 (a), but with 4 arbitrary solutions, described in the figure caption. We do not see the algorithmic phases seen in Fig. 6 (a) and each solution's weight remains largely constant throughout model training.

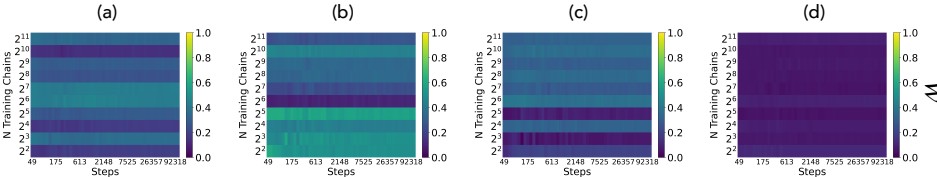

Figure 41: **Applying LIA with 4 arbitrary solutions** LIA is optimized with 4 arbitrarily chosen solutions (a) weight for the first solution, which outputs a next state probability using a frozen transition matrix drawn from the Dirichlet prior. (b) weight for the second solution, which outputs a next state probability using a frozen stationary distribution drawn from the Dirichlet distribution. (c) weight for the third solution, which outputs a 0.2 probability for even states and 0.0 probability for odd states. (d) weight for the fourth solution which outputs a 1.0 probability for state 0 and 0.0 otherwise.

**Are the same algorithmic phases extracted from LIA when arbitrary solutions are mixed with algorithmic solutions?** In Fig. 42, we combine the 4 arbitrary solution with the 4 solutions described in the main text. We find that the 4 algorithmic solutions indeed show high weight in the same way as in Fig. 6 (a). The 4 arbitrary solutions get assigned a near zero weight everywhere. These experiments in Fig. 41 and Fig. 42 confirms that LIA extract meaningful phases only when the solutions are indeed relevant to the task and the model.

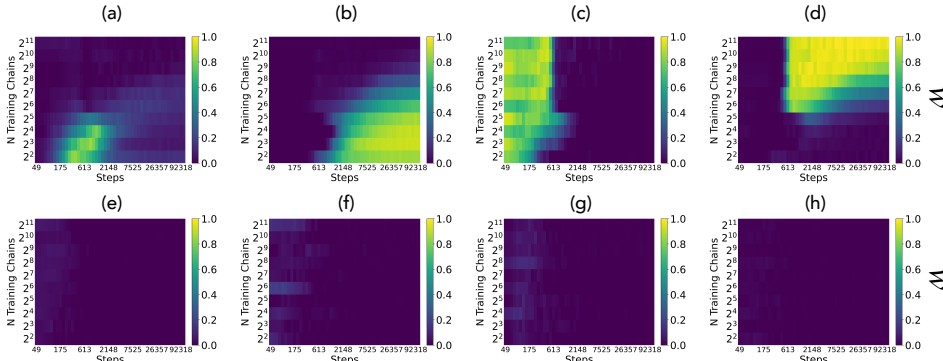

Figure 42: **Applying LIA with the 4 algorithmic solutions and 4 arbitrary solutions** (a) weight for `Uni-Ret`. (b) weight for `Bi-Ret`. (c) weight for `Uni-Inf`. (d) weight for `Bi-Inf`. (e, f, g, h) weights corresponding to, respectively, solutions (a, b, c, d) in Fig. 41.

## J IMPORTANCE OF SEQUENCE-SPACE STRUCTURE TO STUDY TRANSFORMERS

In this work, we focused on a sequence modeling task consisting of a mixture of Markov chains. We argue that there are unique properties of sequence modeling which makes phenomena qualitatively different. Sequence modeling tasks involve the integration of information embedded in individual tokens and their positions. In order to compose this information, a sequence-space computational structure should be learned. This is not the case for many synthetic tasks used to study transformers.

More specifically, many ICL studies focus on **Linear regression** (Garg et al., 2023; Akyürek et al., 2023; Li et al., 2023c; Von Oswald et al., 2023; Mahankali et al., 2023; Raventós et al., 2023; Li et al., 2023b; Lu et al., 2024; Lin & Lee, 2024), where exemplars are presented in pairs as $(x_i, y_i)$. Following (Garg et al., 2023), many works *do not use a tokenized representation*. Thus, the only spatial computational structure needed to be implemented is a fixed template which recognizes pairs of $(x_i, y_i)$ from the context. Furthermore, a recent paper (Tong & Pehlevan, 2024) showed that multi layer perceptrons (MLPs) can learn linear regression and classification tasks in-context, sometimes competitively to transformers.

Motivated by a lack of language structure in synthetic studies of ICL, some recent studies have begun analyzing transformers trained on **probabilistic formal languages** (Edelman et al., 2024; Akyürek et al., 2024). In such a setting, Edelman et al. (2024) found statistical induction heads which is a probabilistic variant of a copying induction head (Elhage et al., 2021; Olsson et al., 2022). Akyürek et al. (2024) study in-context learning of deterministic finite automata (DFA), and find n-gram heads which perform copying similarly to induction heads but based on n-grams. These studies showed that studying ICL on formal languages helps develop rich training dynamics (Edelman et al., 2024) and a complex relation to model architectures (Akyürek et al., 2024). It is also the case that, to the best of our knowledge, there does not exist a study equivalent to Tong & Pehlevan (2024) for linear regression but applied to formal languages. We suspect MLPs will struggle to learn formal languages requiring a dynamic spatial computational structure.

The unigram solutions vs bigram solutions discussed in our work highlights that even the simplest form of spatial computational structure, counting states vs. counting pairs (transitions), introducing a new axis of analysis: the circuit complexity of solutions. In our work, the bigram solutions, `Bi-Ret` and `Bi-Inf`, arguably need a more complex circuit structure. The attention maps visualized in Fig. 18 and Fig. 19 support this argument, though there is substantial work to be done to confirm this intuition (e.g., by properly defining circuit complexity). Nevertheless, well aligned to this intuition, both bigram solutions always follow after a unigram solution in our experiments.

In Sec. 4.2 and Sec. 4.3, we demonstrated how a varying circuit complexity can result in significantly different learning time for different algorithms. We have also demonstrated that different learning speeds can cause an algorithm to be hidden or suppressed (Fig. 8 (a,c)). Additionally, we have demonstrated that a change of tokenization, which does not change the task but only alters how the task is presented to the model, can remove a solution from emerging by closing the circuit complexity gap.

Overall, we believe that studying in-context learning with setups *requiring non-trivial sequence-space computation*, e.g., formal languages, will be crucial to advance our understanding of ICL in LLMs at scale.

## K    DISCUSSION AND FUTURE DIRECTIONS

**Distributional Adaptation**    An interesting phenomenon observed in Sec. 4.1, Fig. 6 is that the predicted OOD KL is always slightly higher (worse) than the model's actual performance. This signifies that our algorithmic decomposition does not explain the model's behavior perfectly, and the model is likely applying a combination of solutions slightly better fit for sequences it has not observed. A possible future direction would be to understand how such behavior is possible. We lay down a hypothesis: `Bi-Ret` *emerges on top of the* `Bi-Inf` *solution, but there is a circuit triggering this solution preferentially for ID sequences*.

**Parallel or Interacting Circuit Evolution?**    An interesting question raised by our analysis is whether different algorithms can evolve in parallel or not. This question is especially motivated by the extremely stable emergence of the `Bi-Inf` solution with respect to optimization steps. As seen in Fig. 6, the `Bi-Inf` solution emerges very stably even for data diversity levels different by $2^5$. One natural question is whether it *could* emerge with data diversity under $2^6$, but is just not observed due to `Uni-Ret` and `Bi-Ret` being found first. Another question with a slightly different implication is whether the circuit supporting the solution did in fact already emerge, yet is not used.

Thus, an explicit and interesting future direction is to try to ablate the `Bi-Ret` solution (via careful design, which we do not know yet) and see if `Bi-Inf` can be observed from data diversity level it is not observed in the current experiment.

**Optimization Limited Emergence vs Data Limited Emergence**    We find many algorithmic transitions in our study. The formation of an induction head (Fig. 1 (b)) and the rise of `Bi-Ret` (Sec. 4.1) are two of them. Here, we argue that these transitions are different in nature. If one carefully looks at Fig. 6, one can notice (as discussed in the paragraph above) that the emergence of `Bi-Inf` is largely independent of data diversity. We thus suggest this emergence is only limited by optimization. However, the emergence of `Bi-Ret` is limited by data diversity (usually limited means that a higher data diversity is needed, however for `Bi-Ret`, a lower data diversity makes it easier to emerge). We thus propose a classification of emergence of mechanisms into two classes: one driven by the introduction of sufficient compute and another driven by a sufficient data diversity.

## L    CODE AVAILABILITY

All code used to run the experiments and analysis are available at

https://github.com/cfpark00/markov-mixtures.

