# OpenReview forum: "Competition Dynamics Shape Algorithmic Phases of In-Context Learning"
_ICLR.cc/2025/Conference — ICLR 2025 Spotlight_

### Official Review · Reviewer_medY · 2024-11-02

**Soundness:** 4
**Presentation:** 3
**Contribution:** 4
**Rating:** 10
**Confidence:** 4

**Summary:**

With this paper the authors contribute the following:

* The authors propose a new sequence modelling setting, namely learning to
  predict sequences sampled from a Markov chain sampled from a finite mixture
  of Markov chains.
  * This is similar to the variable 'task-diversity' in-context linear
    regression setting from Raventos et al., but using Markov chains rather
    than a regression model to generate sequences following Edelman et al.

* The authors list four idealised algorithms for solving this sequence
  modelling setting, along two axes of variation, namely:
  * observing either unigram statistics (states) or bigram statistics
    (transitions), and then
  * making predictions by leveraging knowledge of the latent mixture of
    Markov chains, or based solely on the observed frequencies in the
    context.

* The authors devise a pair of tests for distinguishing a learned predictor
  along these two axes, and in doing so reveal that depending on the degree
  of task diversity, the sequence length, and the number of training steps,
  small transformers trained for this task will behaviourally resemble each
  of these four algorithms. This creates the titular 'algorithmic phases'.

* The authors propose a behavioural analysis method, "Linear Combination of
  Algorithms (LCA)" whereby they decompose a transformer's outputs (given
  in-distribution inputs) into a mixture of the outputs of the four idealised
  algorithms. The decomposition is made by projecting the outputs onto the
  probability simplex spanned by the four algorithms in function space.
  The authors track this decomposition over training and observe a close
  alignment between trends in the weight of certain algorithms and trends in
  out-of-distribution generalisation including the "transience" phenomenon.

* This phase isolation methodology and auxiliary metrics replicates a number
  of phenomena reported in prior works on ICL in similar and disparate
  settings, and some new phenomena. In the main text, the emphasis is on the
  following three phenomena:
  * The authors find a 'task diversity threshold' at which transformers
    switch from learning to leverage knowledge of the pre-training task
    distribution to learning to generalises to unseen Markov chains. This
    finding is analogous to that of Raventos et al. for in-context linear
    regression.
  * The authors find that in certain phases transformers develop statistical
    induction heads, replicating the finding of Edelman et al. in a similar
    setting based on infinite mixtures of Markov chains, but for finite
    mixtures of Markov chains.
  * The authors show that with increased training time individual
    transformers shift from initially adopting a generalising method to
    eventually preferring one that leverages information of the training
    mixture. This is somewhat analogous to the "transience" phenomenon
    reported by Singh et al.
  * The authors also claim to replicate other phenomena, but I am less
    familiar with these other phenomena and the details are excluded from the
    main text, so I have not been able to evaluate them.

* The authors argue that their finite mixture of Markov chains setting
  offers a unified setting in which to study the emergence of in-context
  learning, which has previously been studied in disparate settings.

**Summary of my review:**
I was impelled to write a long review. So, I also include a summary of my
review here.

* I think the authors have made a strong contribution in an important area of
  the science of deep learning. Their setting is elegant, expressive, and
  permits an interesting variety of idealised solutions. The results from
  their phase isolation and LCA analyses are interesting and informative.

* However, I think the claims made in the paper at times overstate the
  results or are overly confident given the limitations of the methodology,
  which are not adequately discussed. In particular:
  * The phase isolation methodology is incapable of ruling out plausible
    alternative algorithms undermining the authors' claims that they have
    knowledge of what algorithms the transformer is implementing.
  * The "linear combination of algorithms" methodology, while revealing
    interesting behavioural dynamics, does not seem to me to be worthy of
    being called "mechanistic" nor an "explanation" of the transience of
    particular algorithms or other phenomena.
  * The motivation for proposing a new setting is to unify disparate studies
    on the emergence of ICL, however the proposed setting is not uniquely more
    appropriate for this purpose than alternative settings.

* I believe that the framing is sufficiently misleading that I cannot
  recommend the paper for acceptance in its current state. However, if the
  authors are able to back up their confidence or if they commit to tempering
  their claims then I would be happy to recommend the paper for acceptance
  because I didn't note any major technical flaws and I think the work is
  important and interesting.

* In addition, I note a number of additional questions and more minor
  concerns including about the naming of the "-ICL" algorithms and some
  details of the setting and results, detailed in the questions section of
  this review.

I look forward to the discussion period.

**EDIT TO ADD:** Summary of discussion:

* The authors addressed all of my concerns about the paper's framing.
  * The authors no longer claim that the phase isolation methodology rules out alternative similar algorithms or identifies the precise mechanisms of the learned transformer implements.
  * The authors talk more precisely about the LCA (now: LIA) methodology as revealing only behavioural dynamics.
  * The authors now motivate their work in more moderate terms.
* In addition, the authors added new preliminary mechanistic experiments providing further interesting insight into learning dynamics in their rich setting.
* I believe that the framing is now acceptably accurate. Therefore because of the paper's strengths I am pleased to raise my rating from 5 to 10.

**Strengths:**

As I said I think this is a strong paper. I note at least the following
strengths.

1. The paper is well-motivated by the importance of of understanding the
   emergence of algorithmic structure inside transformer sequence models,
   which is a priority for the science of deep learning.

2. To this end the authors contribute a neat setting with a clean and flexible
   data generating process and an interesting and rich collection of idealised
   solutions.
   * This setting can also serve as a solid basis for future work, experimenting
     in the same setting or extending it to (for example) higher-order,
     non-Markovian sequence modelling tasks, hidden Markov models, etc., all
     of which are immediately suggested by the authors' framework.

3. The authors have also conducted a comprehensive study of the numerous axes
   of variation in this setting.
   * The replications of phenomena found in prior work is valuable.
   * The classification of the configuration space into qualitatively distinct
     behavioural 'phases' is quite striking.
   * The LCA analysis is thought-provoking.

4. The LCA technique, carefully interpreted, is an elegant idea for
   behavioural analysis of models in general, when plausible candidate
   algorithms are known. I like it and I think I would use it (I have noticed
   several opportunities to use this in my research since seeing it in this
   paper.)

Overall I think the paper makes a valuable contribution that enriches our
understanding of ICL phenomenology and creates a rich framework for future
research that can continue exploring this important topic.

**Weaknesses:**

### W1. Insufficient evidence for confidence in algorithmic phase identification

The authors chose to name their paper 'algorithmic phases of in-context
learning'. I am a believer in the importance of names and I think this gives
me grounds to assume that the authors view their labelled phase diagrams as a
major element of their contribution.

I consider this labelled phase diagram to have two distinct parts. The first
of which is the division of the hyperparameter space into regions where a
trained transformer displays distinct modes of behaviour as quantified by the
behavioural classification metrics outlined in section 4.2 'isolating
algorithmic phases'. I am impressed by this part of the contribution.

I am concerned about the second part, namely the labelling of phases with
specific algorithms. The authors confidently association of each of the four
behavioural phases with one of the four algorithms described in section 4.1
'the Bayesian and non-Bayesian solutions of finite Markov mixtures'. I am
concerned that the authors have not secured sufficient ground from which to
confidently present these algorithms as uniquely accurate descriptions of the
transformer's behaviour in each phase.

**Phase isolation is not exhaustive:**
The phase identification methodology appears to assume a priori that these
are the only plausible algorithms, as if that were the case, then it would be
sufficient to associate these algorithms to each phase. However, these
algorithms do not appear to me to be *uniquely* principled ideal models of
how a sequence model might solve the Markovian prediction task, so I do not
think it appropriate to rule out *a priori* the many other possible
algorithms that could explain similar behavioural patterns and would be a
better 'label' for each phase.

A short list of other algorithms I can think of is as follows:

* **Unigram- or Bigram-likelihood Bayesian Averaging with a different
  prior:** An algorithm that functions like Uni-Bayes or Bi-Bayes, but uses a
  different task prior probability vector as a starting point for formulating
  its posterior predictive distribution.

* **Unigram- or Bigram-likelihood MLE or MAP:** An algorithm that functions
  like Uni-Bayes or Bi-Bayes in weighing the likelihood of each task for
  describing the sequence in front of it, but then rather than performing
  Bayesian averaging to make predictions, simply predicts based on the
  most likely task (possibly accounting for the task prior).

* **Slight modifications of frequency counting approaches:** An algorithm
  that functions like Uni-ICL or Bi-ICL but uses slight modifications to
  frequency counting such as starting the count for each unigram from epsilon
  instead of zero or similar for bigrams, or increasing the predicted
  proportions non-linearly in the counts.

These algorithms are close in function space to the proposed algorithms so
that it were the case that these were more accurate descriptions of a
transformer's behaviour, this would not show up in the phase isolation
metrics (nor in the LCA analysis which is based on projections in function
space---the above algorithms would be nearby in function space and so
projections would be similar). The only ways to rule out these alternatives,
along with every other algorithm, would be to devise specific tests to probe
the behavioural differences between these algorithms and the proposed
algorithms, or to probe the internal computational structure of the
transformers to seek to distinguish the mechanisms underlying their
behaviour.

**On appendix E and mechanistic analysis:** I did notice that the authors
also include additional experiments in appendix E that investigate the
activation patterns in various locations inside the transformer. However,
these experiments are insufficient to rule out the alternatives to Bi-Bayes
and Uni-ICL I have listed above, since for example counting transitions is
also consistent with these alternative algorithms. There are no mechanistic
experiments into Uni-Bayes. It appears the authors have more evidence
consistent with Bi-ICL than the other algorithms including the presence of
statistical induction heads, but the broader point stands.

**Overall:**
To be clear, I believe that the algorithms proposed by the authors and their
experiments are novel and significant. Given the infeasibility of uniquely
associating a particular principled algorithm with a phase, it seems fine to
me to start with approximate behavioural resemblance. I think this is what
the authors have grounds to claim they have contributed---a 'fuzzy' label for
each phase that serves as an initial proxy for what the transformer is going,
and this is worth sharing with the research community.

However, I also strongly believe that the authors have an essential
responsibility to clearly articulate the status of their labels. I am
concerned that in the current form the reader will draw the as-yet
unwarranted conclusion that, for example, transformers perform Bayesian model
averaging (in the appropriate phases). While the algorithmic variations I am
proposing are also deviations from 'optimality' as the authors have framed
the problem, it seems an open question to what extent transformers implement
'optimal' algorithms and to me, modelling the specific variations from
optimality in the pre-trained transformer is an important direction for
future work and it is not yet time to declare that the algorithms have been
definitively identified even in this simple setting.

**Requested revisions:**
I would be welcome the authors to correct my understanding of their
methodology and its limitations, or correct my reading of their confidence.
Otherwise, until this framing issue has been addressed the paper is unfit for
publication in my judgement. I would suggest that the authors commit to the
following revisions.

1. The authors should amend their presentation to refrain from claiming that
   the transformer implements these particular algorithms. Some examples of
   language I think is unwarranted include (not exhaustive):
   * Generally in the abstract and introduction any mention of the finding
     that the transformer uses specific mechanisms such as "use of unigram
     vs. bigram context statistics", though it is unavoidable that some
     qualifications must be left to subsequent pages and this is not
     necessarily a sufficient priority in the introduction.
   * Line 222 "we ... show ... the model selects between these solutions ..."
   * Line 295 "We now demonstrate the four algorithms proposed above
     perfectly delineate models trained on our task into explicit phases".
   * Line 345 (emphasis in original) "*Based on these results, we claim our
     proposed algorithmic solutions successfully explain the different
     mechanisms employed by the model to solve our finite mixture of Markov
     chains task.*"
   * Generally section 5 continues under the presumption that the transformer
     "is implementing" the algorithms from section 4.
   * Line 527/528: "to write down four algorithmic solutions ... and identify
     their existence in a trained model".

2. The authors should note prominently and explicitly in the paper (for
   example when introducing their methodology, in the conclusion, or in a
   dedicated 'limitations' section) that their methodology is insufficient to
   rule out other algorithms as being a more accurate description of each
   phase.



### W2. LCA is not mechanistic and does not offer a new explanation

The proposed LCA techniques is described in the abstract, introduction and
figure 1 caption as a "mechanistic analysis" offering a "new" "causal
explanation" of transient OOD generalisation. The terms "mechanistic" and
"explanation" are repeated throughout section 5. I do not see grounds for any
of the terms "mechanistic", "causal explanation", or "new explanation" to be
used in describing this methodology.

**On "mechanistic":** It is inappropriate to call LCA a mechanistic
decomposition. The tool is clearly based on analysing (in-distribution)
behaviour, and it does not reliably reveal internal mechanisms (though it may
do so, or approximately so, in this case). Accordingly, this analysis should
be called "behavioural" rather than "mechanistic".

I suppose the authors are using the word "mechanistic" because they believe
that LCA has identified the algorithm implemented by the transformer, and
that this is confirmed by the fact that the in-distribution behavioural
decomposition is predictive of out-of-distribution behaviour. It may very
well be the case that the transformer implements a coherent mechanism and
that LCA has, in this case, identified it (or approximately identified it).
However, LCA is incapable of revealing the mechanistic nature of a general
transformer, and so this reasoning is not sound.

1. As I argue in the previous section I do not believe they have grounds for
   this, and have only roughly categorised the behaviour rather than
   identified the precise algorithm. LCA will find a decomposition in terms
   of any list of algorithms given to it. If the true mechanisms inside the
   transformer are not represented, it will identify the behaviourally
   closest algorithms as comprising the behaviour.

2. LCA actually makes it clear that the transformer in some sense implements
   multiple algorithms, not one algorithm, at many points during training,
   since the mixtures are not concentrated. While the authors call this a
   'competition' between algorithms, they have not given any mechanistic
   model of this apparent competition. There are many possible internal
   mechanisms that would lead to such an apparent 'competition' appearing in
   the LCA decomposition, some of which would not even correspond to a proper
   competition but would rather be artefacts of the projection in function
   space.

3. In this case that the transformer's (lack of) reliance on pre-training
   task statistics is recoverable from in-distribution behaviour and then
   this prediction is validated on out of distribution behaviour, this does
   indicate that the tool happens to have uncovered part of the transformer's
   mechanism in this case. However, LCA would make the same predictions for a
   transformer constructed or trained to follow one mechanism in-distribution
   while following another mechanism out of distribution.

Because it only measures behaviour, LCA should not be called mechanistic. The
predictive power in this case is not coming from LCA but rather from the
apparent fact that the transformer happens to have roughly the mechanism the
authors expect.

**On "new explanation":** The authors also claim that tracking the LCA
decomposition over training reveals an explanation of the phenomenon of the
transience of out-of-distribution generalisation posed by Singh et al. This
explanation is described in the abstract as a "new insight".

As far as I can tell (the authors can correct me here if I have missed
something), by "explanation", the authors refer to their observation that
the shift in generalisation coincides with a shift in the transformer's
algorithm from predominantly resembling Bi-ICL to gradually increasingly
resembling Bi-Bayes later in training. The authors also observe that Bi-Bayes
is a better solution according to the training distribution, so this shift is
in turn explained by the learning algorithm pursuing better on-distribution
performance.

I think there are a number of problems with the framing of these observations
as a new explanation of Singh et al.'s transience derived from LCA.

1. First, it is not clear to me that the observations go beyond what was
   already reached in Singh et al.'s original paper. In that paper, they
   already talk about one algorithm, ICL, being replaced with a different
   algorithm, "in weight learning" (IWL) with worse generalisation
   performance. The authors of the present paper are pointing to a similar
   trend arising in their replication, but they now have slightly more
   specific models of the algorithms.

2. Second, note that actually there is an important difference between this
   setting and that of Singh et al.'s transience. Namely, in Singh et al.,
   the ICL and IWL algorithms by construction are equally performant on the
   training distribution. This is a crucial difference in this case. As the
   authors note, in this case, Bi-ICL giving way to Bi-Bayes is even less
   mysterious than ICL giving way to IWL in Singh et al., beacause this is
   driven by the in-distribution performance.

   A closer analogy to the setting proposed in the present paper comes from
   related work on in-context linear regression. Hoogland et al. (cited by
   the authors) and (slightly earlier and in more detail under the name of
   "forgetting") in the paper by Panwar et al. "In-context learning through
   the Bayesian prism" (from 2023, on arXiv before Singh et al. was published
   actually). These works observe that in the Raventos et al. setting, a
   generalising algorithm gives way in favour of a training-task-dependent
   algorithm, driven by improved in-distribution performance and with a
   concomitant drop in out-of-distribution generalisation performance.

3. Third, given this prior work, upon reading the abstract I was expecting
   perhaps some deeper explanation for what actually drives this phenomenon.
   For example, some insight into why the transition occurs at a given period
   during training or at a given rate, or why the generalising performance
   was preferred in the first place. I could not find any such deeper result
   in this paper.

4. Finally, I do not see a connection between LCA specifically and this
   explanation. The transition from Bi-ICL to Bi-Bayes would also show up as
   a shift in the proximity to Bayesian solution metric from the phase
   isolation methodology in section 4. I see no reason to attribute this
   explanation to LCA.

I certainly believe LCA offers a particularly crisp behavioural perspective
on this transition. But given all this, I think a more appropriate framing
for the authors' contribution would be to say that in addition to replicating
the transience phenomenon, the authors have used LCA to offer a new and
detailed perspective on the (behavioural) competition between algorithms
already thought to underlie the phenomenon of transient generalisation.

**On "causal explanation":** As far as I can tell, "causal" is only mentioned
in the introduction. I don't see any grounds for its inclusion here at all. I
suggest it should be removed.

**Overall:** I believe LCA is an interesting and useful behavioural
technique, but unless the authors can persuade me otherwise I am strongly
opposed to the framing in the introduction and throughout section 5 that the
technique is capable of offering mechanistic insights or that it has offered
a new explanation of the phenomenon of transient generalisation in in-context
learning.


### W3. The motivation in terms of unifying and generalising phenomena is inaccurate

Upon first reading the abstract and introduction I understood this paper to
be claiming that there is a need for:

1. an assessment of whether existing phenomenology are specific to the
   settings in which they were first found or whether they generalise to
   other settings and are thus more likely to be inherent features of ICL,
   and
2. a unified setting in which the various phenomena of ICL that have been
   studied in prior work can all be studied in one place.

The authors don't seem to be positioning this as a motivating vision toward
which they are contributing a 'first step' or something 'in this direction'.
Rather, they say for example in the abstract that their work "enables a
unified framework for studying" ICL. In the introduction they claim that
findings from prior work may be "disparate findings that manifest in specific
scenarios".

I think this motivating story is not an accurate description of the state of
the field and I think the degree of unification achieved in the paper is
strong but not as strong as the authors claim on this first page.

**On generalisation:** It appears to me that the authors have indeed provided
a novel contribution by 'generalising' several prior phenomena by exhibiting
them in a new setting. This is indeed a valuable contribution. However, it
appears that most of these phenomena have already been exhibited in multiple
settings, so it is inaccurate to claim that the phenomena were previously
only known to hold in isolated settings.

* In the case of the task diversity threshold, the authors have generalised
  this to the Markovian sequence modelling setting. Previously, it was shown
  for in-context linear regression (Raventos et al.), along with image
  classification (Kirsch et al.) as cited by the authors. The same phenomenon
  has recently been shown in a multi-task modular addition setting by He at
  al. in a recent preprint "Learning to Grok: Emergence of in-context
  learning and skill composition in modular arithmetic tasks".

* In the case of the emergence of statistical induction heads, there is only
  a very minor generalisation taking place since statistical induction heads
  were already shown by Edelman et al. to arise in Markovian sequence
  modelling with infinite mixtures of Markov chains. The generalisation to
  finite mixtures of Markov chains is, in my opinion, valuable but not very
  surprising especially given that the original exhibition of induction heads
  by Elhage et al. and Olsson et al. was in the completely distinct setting
  of language modelling.

  Moreover, I am not aware of induction heads being found in other settings
  such as in-context linear regression. If any ICL phenomenon is isolated to
  certain settings, this suggests that induction heads represent such a
  phenomenon.

* In the case of transience, it's true that Singh et al. study a specific
  scenario that is designed specifically to isolate this phenomenon, but it's
  not true that their findings have not been replicated in other settings.
  In the previous section I already noted work noting transient
  generalisation for in-context linear regression including Hoogland et al.
  (already cited by the authors) and the earlier work by Panwar et al.
  Beyond in-context linear regression, He et al. "Learning to grok"
  (mentioned above) have also demonstrated transience in in-context modular
  arithmetic.

  These examples actually seem more relevant to the present work than Singh
  et al., as they concern transitions between two different ICL algorithms,
  rather than between ICL and pure memorisation as in Singh et al.

* (I have not evaluated the novelty of generalising the other phenomena, as I
  am less familiar with these parts of the literature, and moreover the above
  three phenomena are the ones the authors discuss in the main text.)

Once again, I believe the authors have made a strong contribution, but my
concern is that they have not accurately described it in their abstract and
introduction, and they need to reframe their contribution in a more accurate
manner in my opinion.

**On unification:** Moreover, I did not find the motivation in terms of
unification compelling.

To me it seems like the extent of a 'unification' achieved by the authors is
that they have combined multiple interesting axes of experimental variation
(e.g. studying models at varying task diversity and training time) previously
varied individually in prior work. They have created one setting rich enough
to include these axes of variation, and they have demonstrated that along
each of these axes previously studied phenomena are replicated in their
experiments. I will repeat that I find this comprehensive investigation of
the axes and their combinations is a strong contribution.

However, in describing this contribution, 'unification' seems too strong of a
word. When I read the word unification, I think the prior settings should be
recoverable as special cases of a more general setting. Operationally, the
field should be able to continue forward by discarding previous settings in
favour of using a truly 'unified' setting. I think this standard has not been
met by the proposed setting. Rather, if the field universally adopted the
proposed setting, at least the following research directions would be
precluded.

1. Different settings encourage different mechanistic solutions in the
   transformer's internals. For example, while the Markovian setting allows
   one to study statistical induction heads, one does not have the ability to
   study the particular mechanisms that emerge in order to perform in-context
   linear regression or modular arithmetic. For the field to make progress on
   mechanistic analysis of in-context learning in transformers, it seems
   useful to be able to take advantage of the various constructions that have
   been proposed for specific implementations of in-context linear
   regression, for example. I don't see why we shouldn't try to keep our
   range of interesting synthetic settings as broad as possible.

2. Another example comes from the quite specialised setting studied by Singh
   et al., where, by construction, ICL and IWL are equally performant on the
   training distribution. Nevertheless, there is still an algorithmic
   transition between these algorithms at some point in training. This gives
   rise to questions that can't be asked or answered in a setting where the
   main four algorithms achieve quite different performance in-distribution,
   such as what drives the transition even in this case (when the need for
   better in-distribution performance is ruled out as an explanation for
   driving this transition).

The authors have included appendix G with some discussion on the perceived
benefits of studying Markovian sequence modelling tasks rather than modular
arithmetic or linear regression due to the lack of "sequence space structure"
in these alternative settings.
I must admit unfortunately I did not follow the discussion despite trying to
see their point of view. It is not immediately clear to me what sequence
space structure means. But if the authors refer to the fact that in Markovian
sequence modelling in order to count bigrams the transformer must look at
pairs of tokens, I note that even though modular arithmetic and linear
regression are usually formulated in-context using an i.i.d. sequence of
inputs, it is still necessary for example in Raventos et al.'s setting for
the transformer to look at pairs of sequence items (one containing the x and
the next the corresponding y). It is not clear to me how this is less rich
than looking at bigrams.

**Overall:** I can't emphasise enough, I really like the setting and the
comprehensive analysis along multiple axes. I am only concerned that the
introduction does not provide an accurate motivation for the work, and I
would like to respectfully challenge the authors to lay out a stronger case
for their contributions in their introduction. Doing so, in my opinion,
should not be too hard, because the authors have made some strong
contributions on an important topic.

**Questions:**

I collect various questions, minor concerns, or suggestions that occurred to
me while reading the paper. Given the length of my review I don't expect the
authors to respond to all of these questions (though I would be happy for
them to do so). If the authors are interested in me revising my decision and
have limited capacity in the discussion period I would recommend that they
engage with me on the three weaknesses before the contents of this section.
More importantly I hope that they might consider my questions and suggestions
and consider revising the paper to improve the presentation as they find most
appropriate.

**Q1. The four algorithms have unclear and misleading names.**
As I said, I am a believer in the importance of names. I felt strongly that
the choice to use '-ICL' as a suffix in the names of 'Uni-ICL' and 'Bi-ICL'
is a mistake that undermines the quality of the paper. By selecting these
names the authors have at the same time created the following two problems.

1. They have conflated one half of their list of algorithms with the concept
   of in-context learning, entrenching a false connotation that these two
   algorithms are more exemplary instances of ICL algorithms than others, in
   fact that the Bayesian solutions are not classified as ICL.
   This contradicts the authors' stated message in the abstract and
   conclusion, that ICL is an umbrella concept that encompasses multiple
   concrete algorithmic instantiations. Recalling again the paper's title,
   all four phases are supposedly 'of in-context learning'.
   I do not think that the authors mean to hold up Uni-ICL and Bi-ICL as
   'truer' examples of ICL than the 'Bayesian' algorithms, yet this is what
   their naming choice achieves.

2. They have missed an opportunity to communicate what is unique about these
   particular ICL algorithms. This is the role played by 'Bayes' in the names
   'Uni-Bayes' and 'Bi-Bayes' (in my reading this helpfully conveys that the
   methods use Bayesian averaging). I leave it to the authors to decide what
   would be an appropriate analogue of 'Bayes', but I invite them to consider
   using the term 'frequencies' or 'induction' and I urge them to avoid
   anchoring on induction *heads* themselves (a feature of transformers
   rather than ideal algorithms) or using the misleading 'non-Bayesian'
   terminology of Raventos et al.

I respect the right of the authors to name the algorithms. I can't say this
concern alone would prevent me from recommending the paper's acceptance.
However, in this case I feel strongly enough to register my protestation
about the names given my fresh perspective on the algorithms and the authors'
chosen takeaway message.


**Q2. Inconsistent summary of algorithms between main text and figure 4
caption:** In the figure caption, the Uni-Bayes and Bi-Bayes descriptions
talk about selecting a 'closest task' from the mixture. Based on the main
text, my understanding is that they do not select a single task but rather
they form a posterior distribution over all tasks and use the posterior
predictive distribution to make their prediction. Using the 'closest
distribution' sounds more akin to using a maximum likelihood distribution,
rather than using the posterior predictive distribution.

**Q3. Questionable choice to use a nonuniform task prior:** If I understand
correctly from the setting description, the authors sample a prior vector
from a uniform Dirichlet distribution over task priors. The resulting prior
will be almost certainly non-uniform and with high likelihood for high task
diversity it will have a small number of tasks with quite large
probabilities and a large number of tasks with very small probabilities. It
follows that after a reasonable amount of pre-training there may be some
tasks that have barely been sampled at all.

This seems to me to be a significant departure from the use of task diversity
by of Raventos et al., for whom, if I remember correctly, the task prior is
always uniform. I believe that having a skewed task distribution may confound
experiments since what I would call the 'effective task diversity', meaning
roughly the number of tasks the transformer has to 'memorise' in order to get
good performance on in-distribution evaluation (assuming tasks are sampled
from the same skewed prior for ID evaluation) will be smaller than the
specified task diversity, since the transformer can get away with not
remembering low-probability tasks.

The inclusion of this additional complication appears to be unjustified by
any particular argument in the manuscript. I would be curious if the authors
have a strong reason for including this detail. Of course, it is a virtue of
the setting that one can consider different distributions of tasks, since
this might be an interesting direction for future work, but such work would
surely involve *systematically* varying the prior rather than abdicating
control over the prior by sampling it from a high-dimensional Dirichlet
distribution.

Finally I note that it is regrettable that this detail appears to be
documented only in the appendix.

**Q4. Convoluted phase isolation tests:** These test seems very intricate. It
is not clear to me that they are the clearest ways of isolating phases, and I
wonder if you have considered and ruled out simpler alternatives.

1. For the bigram utilisation test, it occurs to me that the predictions of a
   model paying attention exclusively to unigram patterns would not vary much
   depending on the current state (at least late in the sequence). On the
   other hand, bigram-based models would vary their prediction based on the
   state. Therefore, I wonder if you have considered a simpler test of
   somehow quantifying uniformity in the set of rows of the revealed
   transition matrix?
2. For the proximity test, I wasn't able to think of a simpler test, beyond
   the idea that perhaps something with generalisation performance could be
   used.

**Q5. Missing details for proximity test:** I was left wondering about some
details of the proximity test. Unless I missed something, I would recommend
clarifying the following points, if possible in the main text or otherwise by
expanding on the 'additional details' in the appendix.

1. How is distance to a set of tasks defined? Is it the distance to the
   closest member of the set?
2. Does the measurement of distance to the training task set account for the
   task prior at all? I am concerned that for example if the closest task in
   the training set happens to be a task with very low prior probability,
   then this task will not draw the model's posterior towards that closest
   task, and the posterior may be more likely to be falsely detected as
   closer to a random task than if the closest training task happened to be
   one with higher prior probability.
3. How is the chain that is not part of the training set or the control set
   sampled?
4. I think I can guess how you turn the procedure you outlined into a single
   number used to colour your phase diagram, perhaps it involves repeating
   this procedure several times, and estimating an empirical probability of
   the closest task being from the training set, giving a number between 0
   and 1. If this is correct, I think it is worth spelling out in the text,
   as well as noting somewhere how many trials you take. If this is wrong
   then it's definitely worth spelling out in the text.

**Q6. Missing details about figure colour schemes:**
The phase isolation methodology and the LCA analysis are two distinct methods
for colouring a point on a phase diagram. I realised that it is not always
clear which of these methods you are using in each of the diagrams. It is
clear in Figure 4 and figure 5 where these techniques are explicated. For the
remaining figures, I am not sure which methodology you use, and I couldn't
find it documented anywhere. Please consider clarifying this in each figure's
caption.


**Q7. Why is LCA formulated in terms of L2 distance for probabilities?** It
would seem more natural to minimise cross entropy or KL as is used elsewhere
in the paper, and this would allow a clearer comparison to other quantities
such as model KL vs. LCA KL and so on. I don't think this is necessarily a
major issue but I just wondered if the authors had a good reason for it.


**Q8. What is the relationship between LCA and delta metrics from Raventos et
al.?** Raventos et al. consider two setting-specific metrics, they denote
them 'delta ridge' and 'delta dMMSE', measuring the L2 distance between the
predictions of their pre-trained transformer and those of their idealised
linear regression algorithms (ridge regression and dMMSE). These are
essentially measures of how close in function space the transformer is to one
of the algorithms. Have you thought about the relationship between these
metrics and LCA?


**Q9. How close is the LCA fit?** LCA weights are defined via a least squares
optimisation problem. In figures 6 and 7 the authors plot the argmin weights.
What is the min? In other words, how large is the irreducible component of
the least squares loss representing the distance of the transformer from the
simplex spanned by the four algorithms in function space?

It is important that this metric remains low in order to believe that the LCA
has captured something meaningful about the behaviour, rather than a very
lossy projection of the behaviour. Therefore I believe the authors should
report this metric in the paper, if not in the main text then at least in an
appendix.

A somewhat related metric appears to be the comparison between the LCA KL and
the Model KL in figure 6. However, if two models have similar KL from the
ground truth sequence that does not necessarily imply that they have low KL
between them. It would be informative to add the KL between the LCA and the
model to the two lines in these figures. This would play a similar role to
the residual.


**Q10. Non-ICL algorithms early in training:** Figure 6 shows shifts in the
model's development across training. One lesson from Hoogland et al. (a paper
cited by the authors in the appendix) is that the choice of algorithms early
in training might be even more unsophisticated than those that eventually
arise at convergence for a given task diversity and sequence length
configuration.

For example, early in training, I hypothesise that some of these transformers
might behave in a way that is well-described by an algorithm that does not
involve any in-context learning at all. Some particular algorithms that I
would consider searching for include the following:

* **Unigram prior:** Learning the average stationary distribution from all
  tasks and predicting tokens based on this distribution without looking at
  context.
* **Bigram prior:** Learning the average transition matrix and predicting
  based on this without looking at context.

Have the authors considered adding such non-ICL algorithms into the LCA
analysis? I would be curious whether doing so reduces the residual at all.


**Q11. Questions about evaluation:** Two small questions about the methodology
for evaluation.

1. During ID evaluation, what prior do you use for sampling the tasks from
   the set of tasks? Do you use the training prior or a uniform prior? You
   just say you 'choose one from a set.'
2. During evaluation (for both ID evaluation and OOD evaluation), how many
   tasks do you sample?

**Q12. Sample implementation of data generating process:**
Could the authors please clarify the relationship between the sample
implementation of the data generating process and the actual code used in
experiments?

I take the description of the implementation to imply that this is not the
implementation used in the experiments. This opens the possibility that it
may actually differ in important ways from the actual methodology used in the
experiments. There are certain details, such as the fact that the sequences
drawn from each chain are initialised with a sample from the stationary
distribution of the chain, do not appear to be noted in the
manuscript in any form other than in the reference implementation.

If the authors intend to open source their codebase after the peer review
process as noted in appendix J then I wonder if they intend to keep this
reference implementation in the paper?

Have the authors considered mentioning any details such as the initialisation
of sequences in text form as well as in code form?



**Q13. Paper structure:** On first read I found it slightly difficult to
follow the paper's first few sections. There is a lot going on in the paper,
between the problem, the phenomena, the phases, and the explanations.

I personally found understanding the phases helpful to my understanding of
the remainder of the paper. I wonder if the authors have considered promoting
section 4 to come before section 3? Of course, this is up to the authors.


**Q14. Terminology and notation:** A small number of minor notes.

1. Have you thought about whether the 'phases' are indeed phases in the sense
   of physics?
2. A bold "1" is overloaded as both a vector of ones (when describing the
   configuration of the Dirichlet distribution) and also an indicator
   function (line 232). I wonder if the authors have considered for example
   using blackboard bold for indicators, to avoid any potential confusion,
   not that I think the risk of confusion is particularly severe.

**Q15. Typos:** (just the ones I happened to notice):

1. Line 186/187: I think there is a stray closing parenthesis.
2. Line 422: "more experiments on this sorts".
3. Line 870--: The variables in this list look like they should be typeset in
  math mode.
4. Line 1638: unfinished sentence.
5. Line 2024: "BICL", is this meant to be Bi-ICL?

---

> ### Author Response · Authors · 2024-11-28
> **Reply is coming!**
>
> Dear Reviewer medY,
>
> We thank you for your time and effort in giving us such extremely detailed and thoughtful review. We have just updated our manuscript, and **your thoughtful feedback has really reshaped our paper, both in framing and technical analysis.** We have spent a huge amount of time and effort these past two weeks running new experiments (e.g., see Appendices E and F) and performing rewrites of several parts of our paper (e.g., see Abstract and Introduction). We truly believe our paper is in a better position because of this.
>
> However, since the window for paper revisions expires in a few hours, we have now updated our manuscript with all the changes we could implement, and several that we could not. In particular, we tried several variants in our writing that could incorporate your suggestions as much as possible. This process required a lot of subtlety and effort, and we would appreciate if you could allow us additional time to craft a detailed response specially for you that summarizes our extensive changes in the context of your recommendations.
>
> We'll come back soon!

---

> > ### Comment · Reviewer_medY · 2024-11-28
> >
> > Dear authors,
> >
> > Thank you for the update. From a quick skim of the revision and the other responses, I am already very impressed by the effort you've marshalled during the discussion period to improve the paper, and very thankful that you are taking my suggestions into account. I am looking forward to seeing your full response, whenever it is ready.

---

> ### Author Response · Authors · 2024-11-29
> **Rebuttals (1/9)**
>
> We thank the reviewer for their detailed feedback, extremely precise comments, and a multitude of suggestions to help improve our work! We are very happy to see the reviewer’s excitement about our work, for example that they found our work to be “well-motivated…priority for science of deep learning”, that our proposed task offers “a neat setting…with interesting and rich collection of idealised solutions”, that our experiments form “a comprehensive study of the numerous axis of variations”, that our LCA analysis is “thought-provoking” and “an elegant idea for behavioural analysis of models in general”, and that our work overall “makes a valuable contribution that enriches our understanding of ICL phenomenology”.
>
> Below, we respond to specific comments by the reviewer. Before we begin, we want to emphasize that this review was, at least for us, unprecedented along several axes: in its detail, in its precision, and in its length. We truly appreciate the amount of time and effort the reviewer invested in giving us this detailed feedback and providing us concrete recommendations that have already improved our work. Indeed, we stress that we have worked very hard these past two weeks to accommodate these recommendations and address the reviewer’s concerns. Specifically, we want to highlight that there was a lot in the reviewer’s comments that we agreed with and tried to address: e.g., we agree that we could have been more precise on our use of the word “mechanistic”, especially given the term’s modern connotations in deep learning; we have hence tried to remove any phrases that could have suggested we offer a complete mechanistic interpretation of our models. We have also made concrete attempts at actually reverse engineering our models and providing further (but nevertheless preliminary) experimental evidence for our claims.
>
> We hope our edits and efforts sufficiently address your concerns, and we look forward to engaging with you to further identify paths for improving the quality of our paper!
>
> ---
> ---
>
> ## Response to weaknesses
>
> > **W1. Insufficient evidence for confidence in algorithmic phase identification**
>
> We first note that we are glad to see the reviewer found our contribution of creating a phase diagram of ICL to be impressive—we found this result to be fascinating ourselves, and it updated how we (i.e., the authors) think about in-context learning.
>
> As we understand, the reviewer’s primary apprehension with our analysis is the possibility of alternative algorithms that can describe individual phases in the diagram, and, relatedly, insufficient mechanistic evidence for the algorithms we do associate with said phases (please correct us if we are overly simplifying this!). We completely understand the reviewer’s concern, which we believe stems from a mixture of imprecise writing on our end and some concrete experimental evidence that we could have offered to further back our claims.
>
> Below, we summarize the efforts we have made to revise our manuscript to address the reviewer’s concerns, including both (i) writing changes and (ii) new experiments.
>
>
> ### (i) Summary of our rewrites to be more precise
>
> - **Emphasis on behavioral equivalence and removal of remarks on “implementation”.** The first change we have made to our writeup is the removal of any phrases that suggested a model “implements” the four algorithms we use to define our phase diagram. Instead, we now say the four algorithms *explain the model’s **behavior*** in individual phases, where we define behavior as the next-token predictions. That is, we now *merely* claim that the algorithms can predict what output a model would produce, i.e., how it would behave, for a given input. Given that we are able to do this on both ID and OOD data, we feel confident the model likely implements some mechanism that is *behaviorally equivalent* to our predefined algorithms, but we neither claim to know nor do we try to speculate what this mechanism is. We put explicit remarks inline with these statements at the very beginning of our phase diagram analysis (see L252–L256).
>
>
> **[Continued below...]**

---

> ### Author Response · Authors · 2024-11-29
> **Rebuttals (2/9)**
>
> - **Acknowledged possibility of other algorithms.** We completely agree with the reviewer that our analysis does not guarantee that the algorithms we use to describe our phase diagram are “unique”. We have now made significant attempts at ensuring that our writing does not suggest so. Specifically, we now explicitly state in both the introduction and in analysis (see L358) that we are able to identify **at least** four algorithms that can **explain** the model’s **behavior**. That is, we do not rule out the possibility of there existing other algorithms that can explain the model’s behavior, but we weakly suggest they’ll likely be able to do so because they are in some sort of a class of behavioral equivalence with our considered algorithms: i.e., these alternative algorithms would also produce the same next-token predictions as our considered algorithms, but they may arrive at that solution via a different set of rules or a different implementation. Due to time constraints, we could not weaken this phrasing further yet, but by the final revision we intend to tone the claim down to suggest that algorithms which **generally** produce the same next-token prediction (i.e., they are approximately in the class of behavioral equivalence) could also be considered for describing our phase diagrams.
>
> ### (ii) New set of experimental evidence to further validate our algorithmic picture:
> - **KL divergence between model and algorithms associated with individual phases: Fig. 5(d).** Having said the above, we note we still label our phases using our algorithms’ names. We believe this is justified because (i) we do not claim these algorithms are a mechanistic explanation of the model or the phase diagram itself corresponds to a mechanistic decomposition of the model, and (ii) because we are in fact able to almost perfectly predict the model’s next-token predictions via our procedurally defined algorithms on OOD data. For this latter point, we have now added a new panel to Figure 5 to make the argument quantitatively, showing the KL between our model and predefined algorithms’ next-state transition probabilities is very low in phases associated with these algorithms.
>
>
> - **Effects of alternative algorithms on LCA: Figs. 40, 41.** Arguably, one could consider the possibility of alternative algorithms that are completely out of our considered algorithms’ “equivalence class”, i.e., they produce somewhat reasonable next-token predictions to take away some mass in our phase diagram analysis, yielding a new phase. To address this situation and demonstrate the robustness of our analysis to (at least) naively defined ad-hoc algorithms, we have now added new experiments in Appendix H (Figs. 40, 41). Our results show that such algorithms in fact do not get assigned their own individual phases. We emphasize that we do not claim these experiments are sufficient to suggest one could not come up with alternative algorithms to adversarially attack our analysis, but it does provide further evidence towards our claims’ validity.
>
> **Further (but nevertheless preliminary) mechanistic analysis.** As the reviewer noted, our mechanistic analysis demonstrating mechanistic evidence towards the considered algorithms being “implemented” by the model was relatively weak. We completely agree with this assessment! As noted above, we have partially tried to address this issue by rephrasing several parts of the paper. *However, we note that we have also **substantially expanded** our mechanistic analysis now*, yielding evidence towards the model at least possessing relevant information to implement an algorithm akin to the ones we consider in this work. We emphasize we have made efforts to ensure our writing reflects this evidence is still preliminary though, i.e., we do not claim this evidence is sufficient to make outright claims about models implementing algorithms considered in our analysis.
>
>
> **[Continued below...]**

---

> ### Author Response · Authors · 2024-11-29
> **Rebuttals (3/9)**
>
> - **Summary of new experiments in mechanistic analysis: Appendix E.** In Appendix E.1, we demonstrate that we can reconstruct transition matrices from the model’s MLP neurons in “fuzzy retrieval” phases (we note that is the name we now use for the phases we used to call “Bayesian”; we have made this revision to ensure further precision in our intended meaning). In Appendix E.2, we define a quantitative measure of memorization and evaluate it throughout training; as we find, if a model undergoes transience, memorization is first high, then it goes down when the model enters the Bi-ICL phase (or what’s now called the Bi-Inf phase), and then comes back again! We have also significantly expanded our attention heads analysis in Appendix E.3 and E.4, showing in fact that the attention maps at boundaries of different phases are a direct linear interpolation of attention heads from individual phases. We believe these latter set of results in fact provide *some* grounding to our approach of linearly combining individual algorithms in the LCA analysis.
>
>
> ### Overall Summary of Response to W1
>
> Overall, we note that we have made substantial edits to the paper to address reviewer’s comments: we took their suggestions seriously and have put in an honest effort to modulate our claims in accordance with our offered experimental evidence. Meanwhile, we have also run new experiments to add further evidence towards our claims, as summarized above. We note there are likely still parts of the writing that should have been addressed, but slipped through due to time constraints: we know of at least one such occurrence, i.e., the title of Section 4.2, which we plan to retitle “Analyzing OOD performance with LIA”. We promise to have these slips addressed by the time of final paper revision!
>
> ---
> ---
>
> > **W2. LCA is not mechanistic and does not offer a new explanation**
>
> We believe the reviewer has raised three broad points in their comment: (i) whether LCA (which we have now renamed to LIA) is a technique that can offer mechanistic insights, (ii) whether our explanation of transience via LCA is in fact a “new explanation”, and (iii) whether our explanation is causal. We mostly agree with the reviewer on points (i) and (iii), though we slightly diverge on point (ii). Especially, a framework can provide an explanation of a phenomena at different levels and we can deepen our understanding without being fully mechanistic. We explain our reasoning in the discussion below, and summarize what changes we have made to address the reviewer’s concerns.
>
> - **On causality.** Since this point is easiest to clarify, we discuss it first. We note we have **removed** the mention of the term “causal explanation” from introduction (the only place this term occurred). We believe this was mostly a slip during writing, and we agree it was unnecessary.
>
> - **On LCA being mechanistic.** We agree that claiming LCA (or what is now called LIA in the paper) offers a “mechanistic decomposition” is inaccurate. We have hence taken active efforts to rephrase parts of the writing that suggested LCA decomposes the model into different mechanisms at any given experimental configuration (except in Section 4.2’s title, which we mistakenly forgot to edit). Instead, we now claim that LCA offers a protocol for explaining the model’s **behavior** at any given configuration via an interpolation of individual algorithms we claim to underlie our phase diagram. That is, we now try to pitch LCA as a reliable and useful tool that can help gain insights into the model’s learning dynamics **if** we can preemptively speculate which algorithms explain the model’s behavior in different phases.
>
>     - Despite these writing edits, we do want to emphasize that given LCA’s OOD performance, *we believe* the technique can offer insights into a model’s behavior by offering an “effective substitute”. That is, one can think of LCA as a modeling technique that captures a trained model’s behavior, and can hence serve as an effective substitute for doing theoretical analysis or developing hypotheses to explain its behavior (e.g., why a model makes a decision for a given input). To the extent this modeling is accurate, which, as the reviewer noted, is likely true in our case, the developed insights will likely be effective too. We intend to write a longer note summarizing this in the final version of the paper; due to time constraints, we could not do so just yet.
>
>
> **[Continued below...]**

---

> ### Author Response · Authors · 2024-11-29
> **Rebuttals (4/9)**
>
> - **New explanation for Transience.** Our explanation for transience builds on our demonstration that a specific algorithmic behavior slowly, but steadily, supersedes other ones; in particular, this is the algorithm that achieves better loss on the training data over time (what we used to call Bi-Bayes). As the reviewer noted, the contribution here is relatively simple; however, our novelty lies in the technique used to demonstrate it and the dynamic underneath it. To address reviewer’s concerns that we may perhaps have overclaimed this contribution, we note we have made the following changes to the paper. We also compare our contribution to the related work noted by the reviewer below.
>     - **Removing the adjective “new”.** We agree that the precise dynamic here is driven by the fact that there is a certain
> algorithmic behavior that yields a better loss than others on the training dataset. Hence, to not state this contribution overly strongly, we have removed unnecessary adjectives and merely stated that we “offer **an** explanation for the transient nature of ICL” (e.g., see Contribution 3 in introduction).
>
>     - **Mechanistic evidence for a competition picture of Transience.** As the reviewer pointed out, we chose to use LCA to offer an explanation for the transient nature of ICL. The use of this tool actively pits different algorithms in competition with each other, but we did not sufficiently justify why one should take this competition lens. To further back up this picture, we have now added a preliminary mechanistic analysis, wherein we demonstrate that at the Bi-ICL to Bi-Bayes phase boundary, we can precisely see that attention maps have a structure that is akin to an interpolation of the structures achieved in those two phases individually! As training further proceeds, these maps increasingly look like the Bi-Bayes phase’s maps. Furthermore, we find the model shows an intriguing memorization dynamic: the ability to reconstruct transition matrices from MLP neurons first increases during training, then decreases in the Bi-ICL phase, and then starts to increase again! These results, in combination, are highly suggestive (but do not completely vindicate) the idea that the different algorithmic behaviors are in competition with each other to explain the model.
>
>     - **Loss vs. accuracy comparison to Singh et al.** Finally, we would like to highlight a subtlety that we slightly disagreed with in the reviewer’s comments. The reviewer mentioned that Singh et al.'s setting for studying transience likely has a different dynamic than our work’s, since their system is constructed to have equally performant in-weights learning (IWL) and in-context learning (ICL) solutions. However, we emphasize that performance in Singh et al.’s paper is defined via **accuracy** on the task, not via **loss**. That is, the two solutions to their in-context classification task are equally performant in terms of accuracy! In a recent work released after the ICLR deadline, it was in fact shown that IWL finally overtakes ICL because it yields a better final loss than ICL! The underlying learning dynamic uncovered in that work is exactly what we propose as well. In fact, we would also like to highlight a parallel ICLR submission that has made the same argument as ours, that competitive dynamics underlie transience, but in the setting similar to Singh et al.’s [2].
>
>
> [1] https://arxiv.org/pdf/2410.23042
> [2] https://openreview.net/forum?id=INyi7qUdjZ
>
> ---
> ---
>
> **[Continued below...]**

---

> ### Author Response · Authors · 2024-11-29
> **Rebuttals (5/9)**
>
> > **W3. The motivation in terms of generalizing and unifying phenomena is inaccurate**
>
> We first note that we are excited to see the reviewer’s appreciation of our attempt at capturing several ICL phenomena in a singular setting. While we understand the primary concern the reviewer has flagged, we want to emphasize that getting most (if not all) known results around ICL in a single setup truly was a guiding motivation for our work. This motivation stemmed from wanting to take a step towards developing a unified or normative theory of ICL. Specifically, we believed that given the use of rather disparate setups for understanding ICL, and phenomenology that is at least seemingly in conflict (e.g., task-retrieval vs. task-learning notions of ICL), we should first develop a setup that is rich enough to capture most such known results in a single setting, but also simple enough to be amenable to modeling. Such a setting can then become fertile grounds for attempting a unified account of ICL.
>
> The motivation above is what led to our proposed task: learning to simulate a finite mixture of Markov chains. However, we note that our final pitch in the abstract and introduction ended up getting awkwardly phrased, and hence did not accurately reflect our intentions faithfully. We have now rewritten several parts of the paper (especially abstract and introduction) to address this problem. While we strongly encourage the reviewer to read the abstract and introduction, we specifically highlight that we now explicitly state the following (paraphrased for brevity): our paper’s goal is to **enable a step towards** a unified account of ICL; to this end, we propose a setup that captures most (if not all) known phenomenology of ICL, and is simple enough to be amenable to modeling.
>
> ---
> ---
>
> ## Questions
>
> We thank the reviewer for the questions below. Before we answer them, we note that in the pursuit of making changes according to their comments above and running relevant experiments, we could not necessarily accommodate all requested edits. We have highlighted such cases below, and promise to perform relevant changes in the final version of the paper!
>
>
> ---
> > **Q1. The four algorithms have unclear and misleading names. As I said, I am a believer in the importance of names. I felt strongly that the…**
>
> We thank the reviewer for this question! Based on this question and W1, we have made significant edits to the manuscript. In particular, we have changed the names of our algorithms from Uni-Bayes, Bi-Bayes, Uni-ICL, Bi-ICL to Uni-Ret, Bi-Ret, Uni-Inf, Bi-Inf, respectively. Below, we reply to the reviewer’s sub-questions.
>
> **A1.1:** Indeed, the model is doing some form of in-context learning throughout all four phases. We have now changed the solution names so that they do not seem to suggest that some solutions are ICL and some are not. We thank the reviewer for motivating this important change!
>
> **A1.2:** We thank the reviewer for this comment as well. We have now replaced “Bayes” to “Retrieval”, and we have also clarified the use of “Bayes” in App. B.1. For a replacement for “ICL”, after a long discussion, we converged on “Inference” based on the fact that these algorithms involve inference of the stationary distribution or the transition matrix from the context. We hope the new names better illustrate each algorithm without causing confusion.
>
>
> ----
> > **Q2. Inconsistent summary of algorithms between main text and figure 4 caption: In the figure caption, the Uni-Bayes and Bi-Bayes descriptions talk about selecting a 'closest task' from the mixture. Based on the main text, my understanding is that they do not select a single task but rather they form a posterior distribution over all tasks and use the posterior predictive distribution to make their prediction.…**
>
> We thank the reviewer for pointing this out! The reviewer’s understanding is correct, our Bayesian (now renamed fuzzy retrieval) algorithms indeed perform a Bayesian averaging operation, i.e. an average of every hypothesis’ prior weighted by their likelihoods. We have now made major edits to the main text so that this point is clear. Furthermore, we removed the boxes in Figure 4 which used to describe selection. However, embarrassingly, editing Figure 4’s caption itself escaped our attention and the current version still mentions selection. (We have internally tagged this as an edit to make.) We should also fix this in Fig. 37’s caption as well.
>
> While we will make these edits, we clarify two things partially alleviating this wording:
>
> - In practice, Bayesian averaging and selection turns out to be extremely similar at context length of 15+ (which is most of our sequence of length 512).
>
> - For the above reason, we concisely use the word “retrieval” in our task for “fuzzy retrieval” to make intuitive explanations of phenomena, especially in App. C.
>
> ---
>
> **[Continued below...]**

---

> ### Author Response · Authors · 2024-11-29
> **Rebuttals (6/9)**
>
> > **Q3. Questionable choice to use a nonuniform task prior: If I understand correctly from the setting description, the authors sample a prior vector from a uniform Dirichlet distribution over task priors. ... This seems to me to be a significant departure from the use of task diversity by of Raventos et al., for whom, if I remember correctly, the task prior is always uniform…..**
>
> We agree to this point for the purpose of the conclusions drawn in this paper. Here is the honest reason: at the experimental design phase, we also aimed to investigate how models treat rare chains vs frequent chains. For this reason, we also explored uniform and Poisson priors on the tasks.
>
> However, we did not deeply investigate into the effect of this prior since:
> - The setup was rich enough that this question didn’t make it to our priority list.
>
> - In general, the effect of the prior term turned out to be very weak. This is because the posterior is heavily dominated by the likelihood for retrieval solutions and the prior has minor effects to the inference solutions.
>
> - Our initial experiments didn’t show significantly different results when we changed the prior.
>
> - With a uniform prior, we expected at best a small translation of the *phase diagram* towards low data diversity as we make all data distributions effectively more diverse.
>
> - We observed that the random seed on the Dirichlet *prior* had minor effects on the phase diagram.
>
> We are currently constrained by space to describe this in the main text, but we will try our best to fit these points in future edits to the paper.
>
> ---
>
> > **Q4. Convoluted phase isolation tests: These test seems very intricate. It is not clear to me that they are the clearest ways of isolating phases, and I wonder if you have considered and ruled out simpler alternatives...**
>
> 1. We have indeed considered this test! However, simply comparing the rows of the transition matrix will not fully reveal whether the model is able to use bigrams. The model could very well be using bigram statistics to *retrieve a chain*, but output next state probabilities from the stationary distribution of that chain irrespective of the last state. Please see Appendix B.2 for a discussion of these “unigram posterior” solutions. In other words, our test tries to quantify the presence of bigram modeling in the whole model and not just whether the output probabilities are last token dependent.
>
> 2. For proximity, we have also considered simpler tests, e.g., solely based on the KL divergence of the model and the training set matrices. However, we converged to this method as this method provided *a clear expectation value* when the method is operating in a fully retrieval mode vs. a fully inference mode. Motivated by the reviewer’s comment we added a detailed explanation of this normalization procedure in Appendix A.3.3.
>
> ---
>
> > **Q5. Missing details for proximity test: I was left wondering about some details of the proximity test. Unless I missed something, I would recommend clarifying the following points, if possible in the main text or otherwise by expanding on the 'additional details' in the appendix.**
>
> We answer the sub-questions below.
>
> 1. Yes, it is defined at the distance (KL divergence) to the closest member! We have now clarified this in Appendix A.3.3, and wrote down the formula for proximity explicitly in Eq. 11.
>
> 2. As mentioned above in response to Q3, the effect of the prior is minimal, especially since we uniformly average over sequence length, and thus most sequences are in the long context limit (15+ states).
>
> 3. Each row of this chain is sampled from the Dirichlet prior described in Appendix A.1, similarly from the training set and the random (control) set.
>
> 4. We thank the reviewer for this comment! We have added the formula and evaluation details in Appendix A.3.3.
>
> ---
>
> > **Q6. Missing details about figure colour schemes: The phase isolation methodology and the LCA analysis are two distinct methods for colouring a point on a phase diagram. ... Please consider clarifying this in each figure's caption.**
>
> All figures except Fig. 5 use the LCA (now LIA) phases. We will clarify this in the manuscript.
>
> ---
>
> > **Q7. Why is LCA formulated in terms of L2 distance for probabilities? ....**
>
> We used numerical optimization to minimize Eq. 7, and when using KL divergence we encountered small fitting problems and thus used the L2 distance. We do not expect this to make a big difference. We also point out that this metric was recently used in scaling studies of ICL as well [Anil et al 2024 https://www.anthropic.com/research/many-shot-jailbreaking], [Arora et al 2024 https://arxiv.org/abs/2410.16531].
>
> ---
>
> **[Continued below...]**

---

> ### Author Response · Authors · 2024-11-29
> **Rebuttals (7/9)**
>
> > **Q8. What is the relationship between LCA and delta metrics from Raventos et al.? ... Have you thought about the relationship between these metrics and LCA?**
>
> This is a great question! Unfortunately, we haven’t dived deep into making a metric level comparison with Raventos et al. recently. We would have ideally found a chance to re-read the paper and identify a relation during the rebuttals stage, but the paper edits kept us extremely occupied. We promise to attempt a reconciliation of their metrics and LCA before the final version of the paper is to be submitted.
>
> ---
>
> > **Q9. How close is the LCA fit? LCA weights are defined via a least squares optimisation problem. In figures 6 and 7 the authors plot the argmin weights. What is the min? In other words, how large is the irreducible component of the least squares loss representing the distance of the transformer from the simplex spanned by the four algorithms in function space?...**
>
> Thank you for these suggestions! We have now added these quantifications to the manuscript.
>
> 1. The residual probability space L2 (the argument of argmin in Eq. 7) is shown in Fig. 38 in Appendix H.1. The residual L2 is at the 0.001 scale, which is very small given that probability vectors are normalized to sum to unity. Thus, our four algorithms indeed span a simplex which can model the Transformer accurately. Interestingly, we find the phase boundaries show up as regions with increased—but still very low in an absolute sense—L2.
>
> 2. This is another important quantification, and we now show this in Fig. 39 in Appendix H.1. We show that the KL divergence between the model and the LCA predictions are very low in either direction of KL computation, compared to the KL divergence seen in Fig. 3, 5, 6, 7. Here again, we find a small increase of KL at the phase boundaries.
>
> We thus conclude that the four solutions do provide a good basis to model the trained Transformer.
>
>
> ---
>
> > **Q10. Non-ICL algorithms early in training: Figure 6 shows shifts in the model's development across training. One lesson from Hoogland et al. ... Have the authors considered adding such non-ICL algorithms into the LCA analysis? I would be curious whether doing so reduces the residual at all.**
>
> This is a great question and we would have been really curious to attempt it! Unfortunately, there was only so much time and we ended up prioritizing other experiments for now. However, we would like to note this parallel ICLR submission that has performs an analysis that may be inline with what the reviewer is suggesting (it is not LCA exactly, but the authors demonstrate a competition dynamics w.r.t. a memorization solution that can dictate learning dynamics earlier in training).
>
> ---
>
> > **Q11. Questions about evaluation: Two small questions about the methodology for evaluation.**
> > **1. During ID evaluation, what prior do you use for sampling the tasks from the set of tasks? Do you use the training prior or a uniform prior? You just say you 'choose one from a set.’**
> > **2. During evaluation (for both ID evaluation and OOD evaluation), how many tasks do you sample?**
>
>
> 1. We sample from the training prior to keep the distribution exactly the same. We have clarified this in Appendix A.3.2.
>
> 2. We sample 30 tasks and k (=10) sequences from each task. This detail is available in Appendix A.3.2.
>
>
> ---
> > **Q12. Sample implementation of data generating process: Could the authors please clarify the relationship between the sample implementation of the data generating process and the actual code used in experiments?...**
>
>
> The sample implementation and the actual DGP used in experiments produces *exactly* the same data. The DGP we actually used simply has more class hierarchy, methods, attributes, tokenizing tools, etc. We also clarify that Appendix A.1 does mention the initialization of samples from the stationary distribution. This code block is mostly added since our DGP is very simple and some readers would prefer to read code instead of equations. The codebase will be open sourced, but we intend to keep this code block in the manuscript for clarity.
>
> ---
>
> **[Continued below...]**

---

> ### Author Response · Authors · 2024-11-29
> **Rebuttals (8/9)**
>
> > **Q13. Paper structure: On first read I found it slightly difficult to follow the paper's first few sections. ... I personally found understanding the phases helpful to my understanding of the remainder of the paper. I wonder if the authors have considered promoting section 4 to come before section 3? Of course, this is up to the authors.**
>
> This is a fair question. Our main motivation for the paper structure stemmed from the intuition that, despite its commonplace use in literature, ICL does not have a formal definition. Hence, to study the concept, we should develop a setup that captures as much phenomenology as possible, using the phenomenology itself as the definition of ICL: i.e., if there is a setup that yields a model that captures most known results on ICL, then that setup becomes worth of studying the concept of ICL in a unified manner. Motivated by this then, our paper layout became: (i) propose our task, (ii) demonstrate it reproduces phenomenology of ICL to argue that the setup is worth studying, and (iii) develop insights into ICL.
>
> We have tried to make this narrative more fluid in the updated version of the paper; we’d be grateful if the reviewer can take a look.
>
> ---
> >**Q14. Terminology and notation: A small number of minor notes.**
> > **1. Have you thought about whether the 'phases' are indeed phases in the sense of physics?**
> > **2. A bold "1" is overloaded as both a vector of ones (when describing the configuration of the Dirichlet distribution) and also an indicator function (line 232)....**
>
> **Response to 14.1.** Thank you for this question! As our title suggests, we indeed drew inspiration from the term "phases" in physics. While establishing an exact analogy is challenging due to thermodynamics' focus on "equilibrium states," careful examination of this analogy was the key inspiration that shaped our work. Below, we elaborate on this point, which may bring further clarity to the issues we discussed earlier, such as behavioral metrics, mechanisms, and the nature of explanations.
>
> The first key insight from physics is how scientists defined phases in the early days of thermodynamics, even without knowing that atoms existed! Notably, in physics, **new invention of "order parameters" defines new phases.**
>
> - **Macroscopic Order Parameters (Thermodynamics):** Historically, thermodynamics developed through the identification of macroscopic variables like pressure, temperature, and volume, and their relationships in describing systems' bulk properties. Phase transitions were identified by discontinuities or abrupt changes in these variables or their derivatives. For instance, water's volume changes abruptly when it freezes into ice at 0°C. **Our use of behavioral metrics to differentiate algorithmic solutions in Transformer models parallels the employment of macroscopic order parameters in thermodynamics.** These evaluations detect changes in the model's output behavior without examining internal mechanisms. Thus, we acknowledge that our analysis is not "mechanistic" at this stage!
>
>
> - **Microscopic Order Parameters (Statistical Mechanics):** With the emergence of atomic theory, statistical mechanics provided a microscopic "mechanistic" understanding of thermodynamic phenomena. Microscopic order parameters, such as molecular arrangement and orientation in crystal lattices, offered mechanistic insights into phase transitions. To achieve a similar mechanistic understanding, indeed, we need to examine the network's internal variables, including weights, activations, and neural representations. These internal variables function as microscopic order parameters, and it would  Their analysis could reveal how different algorithmic strategies are implemented within the network's architecture.
>
> To contextualize in the modern deep learning landscape, for example, a remarkable recent sequence of developments can be seen in the behavioral characterization of hidden progress measures [1], followed by the formulation and discovery of mechanistic analogues [2]. Overall, the parallel between studying physical systems and neural networks has helped us understand the reasoning behind your comments on naming and framing. Like all early scientific endeavors, considerable work remains to develop these ideas further, and your thoughtful feedback has brought clarity to our thinking about this topic!
>
> **Response to 14.2.** Thank you! We changed this to the Kronecker delta.
>
> [1] Hidden Progress in Deep Learning: SGD Learns Parities Near the Computational Limit
>
> [2] Progress measures for grokking via mechanistic interpretability
>
> ---
> > **Q15. Typos**
>
> Thank you for highlighting these typos; we have fixed them in the revised manuscript!
>
> ---
>
> **[Continued below...]**

---

> ### Author Response · Authors · 2024-11-29
> **Rebuttals (9/9)**
>
> ### Summarizing rebuttals
> **Summary.** We would like to again and show our sincere gratitude to the reviewer for their detailed feedback and the immense amount of time they must have put in to create such a precise review. We have tried our best to put in a proportionate amount of effort, and we are extremely excited to see the results of the same: our paper now uses a much more accurate terminology throughout, is noticeably more precise with its claims, and has several new experiments that provide further evidence to all claims. We hope our responses above address the reviewer’s concerns, and, if so, that they would consider consider raising their score to support the acceptance of our work!

---

> > ### Comment · Reviewer_medY · 2024-11-29
> > **The revised paper addressed all of my concerns, I raised my score to 10**
> >
> > Thank you for responding to each point of my review in detail, and for your diligent and continued efforts to improve the paper. Your substantial revisions have largely addressed my concerns about the mismatch between the paper's (strong) contributions and its (overly strong) framing/claims. Based on your concessions and revisions so far I trust that you will follow through on the remaining changes you mentioned for which there was not time in the revision period. Accordingly, I have increased my rating of the soundness and presentation and my overall rating (from 5 to 10). I think this paper should be accepted to and indeed highlighted at the conference based on my overall impression that it contributes (1) a useful technique (LCA/LIA) and (2) a rich setting (accompanied by (3) very thorough analysis) all of which I believe will provide a firm basis for future work in the science of deep learning.
> >
> > (I have a couple of notes and follow-up questions to your detailed response that I'll post soon; I wanted to start by communicating this update to my rating.)

---

> > > ### Comment · Reviewer_medY · 2024-11-29
> > > **Notes and follow-up questions**
> > >
> > > Your response completely addressed most of my concerns and questions. I would like to thank you for each of your replies, clarifications, and revisions, but for the sake of brevity given the length of the discussion so far, I will offer my overall thanks and list only the small number of places in which I have follow-up comments or questions. I don't think it's necessary to respond to the comments, but I would appreciate a reply to the questions if you find the time.
> > >
> > > Comments:
> > >
> > > * **W3:** I feel like this concern of mine has been largely addressed by the new abstract and introduction, but not entirely addressed. The new introduction is much improved compared to my recollection of the original. In particular, I noticed a de-emphasis on the goal of showing that prior findings generalise, and I noticed the switch you mentioned to "a unifying setting" (language that does not preclude the possible existence of other unifying settings). I still think these weaker mentions of the claims of generalisation and unification are subject to (parts of) the critique in my original review. On the plus side, the inclusion of the idea that the work is 'a small step towards a unified model of ICL' is appropriate and compelling. I think the introduction could be further improved by leaning into this thread. Anyway, overall the revisions to these and other sections are sufficient to flip my recommendation and I leave the final introduction wording to your judgement.
> > > * **Q8:** The new experiments you added in fig. 5(d) (the KL deltas from each phase) appear to me to be analogous to Raventós et al.'s delta metrics. They used L2 distance between function outputs, which makes sense for regression, but your algorithms and models output distributions, so using KL here makes sense.
> > > * **Q13:** The structure makes sense to me now as does your justification. Alas, I am unable to revisit the state of mind prior to reading and reflecting on the paper in detail. Of course, you could consider showing the paper to new readers if you are interested in further feedback.
> > > * **Q15:** I spotted these additional typos while re-reviewing the draft. I understand you are still updating the draft and submitted the revision in a hurry. You may already have seen some of these, I thought it's worth mentioning them in case they slip through your proof reading pipeline.
> > >   * Line 20 and line 90: "a competitive dynamics"
> > >   * Line 1579: "$\alpha = 1$---should that be a bold 1 with a dimensionality subscript?
> > >   * Line 1668: unfinished sentence in figure caption.
> > >   * Line 2006: missing space after figure link.
> > >   * Line 2650: "tying ... increase the ... error" ("increases"?)
> > >   * Line 2668: "when letting them independent".
> > >
> > > Follow-up questions:
> > >
> > > * **W2:** Regarding point (ii) on the comparison to Singh et al., this is a point I am personally stuck on, and I would appreciate further discussion from the authors, as at least one of us (quite possibly me) appears to be misreading Singh et al.. I ran out of space to write my question here, let me carry it over to the next OpenReview comment.
> > > * **Q9:** I think you intended to link me a paper in your response but there was no reference.

---

> > > > ### Comment · Reviewer_medY · 2024-11-29
> > > > **Reagrding comparison to Singh et al.**
> > > >
> > > > So. I have always thought that the interesting part of the phenomenon Singh et al. shows is that "transience" can arise even when 'it reduces the loss' is ruled out as a generic explanation.
> > > >
> > > > Quoting from Singh et al. section 3:
> > > >
> > > > > Chan et al. [10] found some transient ICL in other data regimes, such as when some small percentage of sequences are non-bursty, where the context does not contain the query (Figure 3). In these settings, the eventual rise of IWL is to be expected, as ICL cannot minimize the loss on the non-bursty sequences. Notably, our results (Figure 1) show surprisingly that ICL transience extends even to cases where ICL is “sufficient” to fully solve the training task (such as when the context always contains a relevant exemplar-label mapping that can be used).
> > > >
> > > > And then from their discussion section:
> > > >
> > > > > **Given that ICL does emerge, why does it fade in favor of IWL?** On our training data, either ICL or IWL can reach good performance. While we do not know for certain why ICL emerges in a network, the question remains of why it fades if it is a viable strategy to solve the training task. Our results in Section 6 suggest that part of the explanation may lie in a competition between IWL and ICL circuits for resources in the transformer residual stream [16]. Under the assumption that IWL is
> > > > asymptotically preferred, this competition would explain why ICL fades after emerging.
> > > >
> > > > > **Why is IWL asymptotically preferred over ICL, when both solve the task?** Induction heads [9] are a possible circuit that may be responsible for ICL, and comprise of two interacting components: finding a match in-context, then copying some token forward. The match operation is fundamentally limited by the softness of the attention operation [41–43]. On the other hand, IWL relies on learning the exemplar-label mapping in-weights, which we show is feasible in Figure 10d. We postulate—and hope to investigate in future work—the possibility that although ICL and IWL can achieve perfect accuracy on the task, this “imperfect match” to prior context asymptotically incentivizes solutions (when training with cross-entropy loss, as is standard) that do not rely as much on context and instead learn in-weights.
> > > >
> > > > They mention the (potentially limited) capacity for the architecture to implement each hypothetical solution perfectly. You point to the difference between loss and accuracy (though I didn't quite understand the distinction). You also pointed to two papers:
> > > >
> > > > * The paper "Towards understanding in-context vs. in-weight learning" (note, this paper is [also under parallel review](https://openreview.net/forum?id=aKJr5NnN8U)), about which I have the same concern that these authors are explaining a *different phenomenon* than the one presented in Singh et al.
> > > > * The other paper you linked, which I have seen but haven't had a chance to study yet.
> > > >
> > > > I am left pretty unclear about whether "transience" does (or should) include the setting where the ideal IWL algorithm reduces the loss compared to the ICL algorithm, or whether that should be considered something else. Question 1: Could you please share your thoughts on this matter?
> > > >
> > > > Finally, Question 2: Did you have a chance to compare to the "forgetting" phenomenon in Panwar et al. that I mentioned in my original review? I think this is a concurrent example (concurrent to Singh et al.) that much more clearly aligns with the phenomenon I think you have reproduced and are (in some sense) explaining, because (1) it involves two ideal algorithms with distinct loss and (2) they are both in-context learning algorithms, rather than ICL vs. IWL as in Singh et al.

---

> > > > > ### Author Response · Authors · 2024-12-02
> > > > > **Response to follow-up**
> > > > >
> > > > > We thank reviewer medY again for their enormous efforts dedicated to improving our work!
> > > > >
> > > > > > **Q8: Raventós et al’s delta metric.**
> > > > >
> > > > > We thank the reviewer for pointing out this similarity. We have added a line to the Appendix describing LIA: “We note that analyzing the deviation of model outputs to a single analytic solution has been explored in a linear regression setting (Raventos et al., 2023). In contrast, LIA extracts the weights of each member solution in a set of solutions using Eq. 7.“
> > > > >
> > > > > > **Q10.**
> > > > >
> > > > > We think the reviewer was referring to Q10 instead of Q9. Apologies for missing the citation there! It should be: https://openreview.net/forum?id=INyi7qUdjZ .
> > > > >
> > > > > ---
> > > > > ---
> > > > >
> > > > > > **Remanent of W2: Comparison with Singh et al.**
> > > > >
> > > > > **Question 1:**
> > > > >
> > > > > While we have ourselves not tried Singh et al.'s setup to elicit Transience, we got in touch with the colleagues who have a paper under review at ICLR that analyzes an abstraction of the setup by Singh et al. to study what causes Transience (we note since the other paper is under review as well, we will avoid naming them or their work). Our colleagues confirmed that in their abstraction, the loss achieved by the ICL solution is worse than the IWL solution on the training data! Both the solutions will given you the same accuracy on the data from the training distribution, but IWL has a lower loss ultimately and hence gradient descent will sooner or later arrive at it.
> > > > >
> > > > > Building on this, we also tried diving deeper into Singh et al.'s paper again. If we understand correctly, Singh et al.'s Fig. 1 shows that the training loss is indeed decreasing during the rise of IWL (please see Fig. 1c at steps>1e7). That is, there is a decrease in loss as IWL re-emerges. We note that since loss is the optimization objective, and only serves as a surrogate for model performance (i.e., accuracy), loss can in fact keep decreasing even if the performance is saturated at 100% (see [1, 2] for a related discussion).
> > > > >
> > > > > Overall, then, we agree with the reviewer that Singh et al.'s results are showing that ICL can be transient even if its *performance* (defined as accuracy in their work) is not worse than IWL. However, we note that since performance is not the optimization objective, if there is an alternative solution with a lower loss, the model will eventually arrive at it. This latter dynamic is what is driving IWL over ICL in Singh et al.'s setup too.
> > > > >
> > > > > In fact, we can see the essence of our argument is present in the three quotes you pointed out from Singh et al.:
> > > > >
> > > > > From Discussion Quote 1:
> > > > > > “either ICL or IWL can reach good performance”
> > > > >
> > > > > From Discussion Quote 2:
> > > > > > “when both solve the task”
> > > > > > “although ICL and IWL can achieve perfect accuracy on the task”
> > > > >
> > > > > And in fact they *seem* to know that the loss, the optimization target indeed prefers IWL, as they say they suggest that the loss has to do with what solution is incentivized:
> > > > >
> > > > > From Discussion Quote 2:
> > > > > > “asymptotically incentivizes solutions (when training with cross-entropy **loss**”
> > > > >
> > > > >
> > > > > **Summary** We still think ICL is only transient if IWL has a better training loss. We don’t think Singh et al.’s results disagree with this, as they only demonstrate that ICL and IWL are *able to perform* equally well on the task. Most importantly, they do not show that IWL’s loss is the same or higher than ICL. In fact, they make a few comments supporting that IWL indeed has a lower loss!
> > > > >
> > > > > [1] https://arxiv.org/abs/2211.08422
> > > > >
> > > > > [2] https://arxiv.org/pdf/2310.06110
> > > > >
> > > > > ---
> > > > > ---
> > > > >
> > > > > **[Continued below...]**

---

> > > > > > ### Comment · Reviewer_medY · 2024-12-02
> > > > > > **Thanks, sounds good, and my final attempt to articulate difference from Singh et al.**
> > > > > >
> > > > > > Thanks for your response. This all sounds good to me, including your comments clarifying the observations of Singh et al. regarding loss vs. accuracy. Based on your clarifications I have achieved a more nuanced understanding of the empirical phenomena that Singh et al. demonstrates. Thank you again for this.
> > > > > >
> > > > > > However, I reserve a minor doubt that this is a fully accurate interpretation of the situation *as it relates to your work and claimed explanation.* Let me try one last time to articulate my final remaining doubt on this topic.
> > > > > >
> > > > > > ---
> > > > > >
> > > > > > This time let me explicitly draw a distinction between the following two pairs of algorithms:
> > > > > >
> > > > > > 1. First, there are the **idealised algorithms** `IWL` and `ICL`, these are the equations we would write if we were designing algorithms by hand for sequence prediction on the Omniglot dataset.
> > > > > >     * These algorithms are akin to your four idealised algorithmic phases, `Uni`/`Bi`-`Ret`/`Inf`, given by equations for posterior predictive inference or pure frequency-counting based on the context.
> > > > > >     * In particular I would define idealised `ICL` for Singh et al.'s data as a function that looks up the query in the context and returns the in-context completion. Assume that there is always a unique completion from the data, as in Singh et al.
> > > > > >     * As for idealised `IWL`, it would simply look at the query and apply the true look-up table underlying the data generating process.
> > > > > > 2. Aside from these idealised algorithms, there are **the solutions (partially) learned by Singh et al.'s transformers,** let me call these `~IWL` and `~ICL`. These algorithms are, hypothetically, approximations of the idealised `IWL` and `ICL`, but they are constrained to be expressible by their transformer architecture (and indeed they are further constrained to be reachable by gradient-based learning).
> > > > > >     * Note that all observations in Singh et al.'s experiments apply to these approximations of idealised algorithms, not necessarily to the idealised algorithms themselves.
> > > > > >
> > > > > > Now, given this distinction, I can articulate my concern as follows: *The loss advantage of ICL over IWL applies to the transformer-implemented algorithms `~ICL` and `~IWL` but **not** to the idealised algorithms `ICL` and `IWL`.*
> > > > > >
> > > > > > * Even if measuring cross entropy loss, on the given training data set, idealised `IWL` has no cross entropy advantage over idealised `ICL` (or vice versa), because both predict the same Dirac distribution over the true next outcome. *In this setting the idealised algorithms are not distinguished by training loss.*
> > > > > > * In contrast, empirically, there is a loss advantage observed by Singh et al., but it is observed in the context of transformer training, meaning it is a difference between the transformer's attempts to implement the ideal algorithms, in other words it is between `~IWL` and `~ICL`. These algorithms don't return Dirac distributions, so even if they make the same modal predictions and therefore achieve perfect accuracy, they can have different cross entropy from the true Dirac distribution.
> > > > > >
> > > > > > **Compare this to other ICLR submissions:** The concurrent work I am familiar with is ["Toward Understanding In-context vs. In-weight Learning"](https://openreview.net/forum?id=aKJr5NnN8U), which you mentioned earlier, and is currently under review, and studies an abstract model of ICL vs. IWL in a somewhat similar setting to Singh et al.'s. I am not sure if this is the same as the paper of your colleagues to which you refer, but it doesn't matter, let me use it to make a concrete point. These authors give an idealised model of ICL, but their model includes mechanistic assumptions about how ICL could be implemented by a transformer and this assumption is what gives rise to a training loss disadvantage compared to their idealised IWL solution. My point is simply that in order to demonstrate such a training loss advantage in the idealised solutions, they had to change the idealised solution to be more like `~ICL` than `ICL` as I defined them, and this underlies the extent of their explanatory power.
> > > > > >
> > > > > > **Compare this to your setting:** You observe transience in a setting where the competing idealised algorithmic solutions are already distinguished by training loss, and then claim that *this distinction* explains transience as in Singh et al. I disagree with this because you are pointing to a loss advantage between idealised algorithms where in Singh et al. there is no such advantage between between `ICL` and `IWL` (even though there is empirically a distinction between `~ICL` and `~IWL`).
> > > > > >
> > > > > > In my opinion, your setting does not capture the same phenomenon---transience even when the idealised `ICL` and `IWL` algorithms are not distinguished by the loss. The true explanation of this phenomenon must lie in the difference between `ICL`/`IWL` and `~ICL`/`~IWL`, that is, in studying how transformers implement/differ from idealised algorithms. This is a direction that your work has not yet explored as far as I can tell.

---

> ### Author Response · Authors · 2024-12-02
> **Response to follow (continued...)**
>
> **Question 2:**
>
> First, we thank the reviewer for pointing out this paper again! We had missed it during our initial submission, and agree that this paper is indeed more similar to our proposed task. We now cite it in Appendix C and Figure 27 (will be reflected in the next version of the paper).
>
> **Main answer.** We certainly think the "forgetting" (term used by Panwar et al. for transience) is a much more similar phenomenon to what we see, modulo the sequence modeling aspect that we have, which adds another dimension of analysis: solution complexity (unigram vs bigram). And with our study, we can now interpret there are competing solutions underlying their observations!
>
> Quoting Panwar et al.’s Section 6.1:
> > We group them into the following 4 categories based on OOD loss and describe the most interesting one in detail (full classification in §D.3): (1) 2 1 to 2 3 : no generalization; no forgetting; (2) 2 4 to 2 6 : some generalization; no forgetting; (3) 2 7 to 2 11: full generalization and forgetting [...] (4) 2 12 to 2 20: full generalization; no forgetting.
>
> These results can be almost one-to-one related to:
> (1) no generalization; no forgetting -> Fig. 33 a (A Uni-Ret solution emerges which doesn’t generalize and thus cannot show transience)
> (2) some generalization; no forgetting -> Fig. 33 b (From a mixture of Uni-Ret and Uni-Inf, we observe an initial increase of KL followed by a decrease)
> (3) full generalization and forgetting -> Fig. 33 d, e (Bi-Inf solution being overtaken by Bi-Ret)
> (4) full generalization; no forgetting -> Fig. 33 f (The Bi-Ret solution not appearing within the computation budget)
>
> We again note we have added a discussion of this paper in Appendix C, where we discuss the similarity of our results to Panwar et al.'s, and also discuss the relation between their setup and ours. We thank the reviewer for bringing this paper to our attention!

---

> ### Author Response · Authors · 2024-12-02
> **On difference from Singh et al.**
>
> Thank you for continuing the discussion!
>
> We would first like to note that your argument makes perfect sense to us, and we had not considered this perspective: the distinction between the idealized algorithms and the corresponding implementation a model arrives at is a really good idea! In fact, rereading Singh et al., we now realize that they explicitly state in their paper something to the effect of: (i) an induction head is (likely) needed for their task to be solved entirely in-context; (ii) since attention is used to implement this, and attention learned via gradient descent can likely implement induction in only a fuzzy way, the ICL implementation will always have some uncertainty (and hence non-trivial loss); (iii) the IWL solution, as it is implemented, does not need to be fuzzy and can hence have lower uncertainty / loss; and (iv) this likely leads to IWL overtaking ICL eventually. While this argument does make explicit the distinction between idealized and implemented algorithms, we really like this vocabulary and the nuance you are suggesting!
>
> **Overall, then, we see your point now!** We are more than happy to revise the next version of the paper to discuss in detail the difference with respect to Singh et al.'s setup for Transience and our own setup. Specifically, we will emphasize the distinction between the idealized and implemented algorithms, but also clarify that the dynamics of transience are led by one solution having a better loss than the other. This loss imbalance can emerge either because of how the problem is specified (as in our case) or how the solution is implemented (as in Singh et al.'s case).
>
> Finally, thank you again for this message! This was very a really clarifying note for us.

---

### Official Review · Reviewer_tKrP · 2024-11-03

**Soundness:** 3
**Presentation:** 3
**Contribution:** 3
**Rating:** 8
**Confidence:** 3

**Summary:**

The authors introduce a new ICL synthetic task that consists of sampling from a finite mixture of Markov chains. Their setting reproduces several ICL phenomena, and through extensive analyses, the authors also report results on phase transitions between different algorithms.

**Strengths:**

I think this is a nice, comprehensive paper that introduces a simple setting that unifies many recent papers on ICL and captures analogous phenomena (task diversity thresholds, transience of ICL). For example, Figure 3 basically reproduces the results of two previous papers.

The experimental protocols for assessing bigram utilization and proximity to the Bayesian solution were thoughtful and creative.

I like the idea of approximating the transformer’s behavior as a mixture of algorithms. It’s interesting that this can be used to predict OOD and that it highlights a sort of “persistent competition” between implementing different algorithms. This potentially provides a compelling explanation for phenomena like transience.

There’s also a comprehensive set of ablations that study the impact of width and data complexity etc…

Overall, the paper was fairly clear.

**Weaknesses:**

First of all, I want to emphasize that I think this line of work is valuable and scientifically meaningful. However, it would be nice to discuss how these insights translate into practical design choices (e.g., predicting OOD performance using a similar approach to this paper).

I think one major premise of the paper is that there’s a transition between different algorithms.
I think this is convincing. However, I think I want some kind of control condition where you use 3 or 4 different “silly” algorithms just to confirm that these kinds of phase transitions aren’t some artifact of fitting some linear combination of algorithms and that, if your algorithms are not related to the task, you don’t always see these interesting algorithm phase transitions.

I think I also wanted a bit of better understanding on how well LCA predicts the performance of the transformer relative to some naive baselines just to confirm that LCM well approximate the transformer.

**Questions:**

This paper identifies a number of these interesting phenomena, but it would be nice if the authors could discuss a bit more about why these phase transitions occur. For example in the LCA analysis, why is there rich structure in these phase transitions when you might (naively) expect a smooth increase in the weights on the optimal solution (e.g., Bi-ICL) and roughly uniform weight on the other solutions? Can you study the properties of the loss function (e.g., Hessian) at these different phase transition points?

---

> ### Author Response · Authors · 2024-11-28
> **Rebuttals (1/3)**
>
> We thank the reviewer for their careful reading of the paper and valuable feedback! We are excited to see they found our work to be "thoughtful and creative", “nice, comprehensive”, and that they "like the idea of approximating the transformer's behavior as a mixture of algorithms", which "provides a compelling explanation for phenomena like transience."
>
> We have made significant changes to our manuscript:
> - A major revision of the writing, revising some terminologies, improving the flow and highlighting contributions.
> - A large set of mechanistic experiments to confirm the existence of the algorithms we delineate. (Fig. 15-26)
> - Additional quantifications of experimental results to verify our findings. (Fig. 5d, 35, 38-42)
> - Additional visualizations of results for completeness. (Fig. 32-34)
>
> Below, we address specific comments from the reviewer.
>
> ----------------------------
> > **First of all, I want to emphasize that I think this line of work is valuable and scientifically meaningful. However, it would be nice to discuss how these insights translate into practical design choices (e.g., predicting OOD performance using a similar approach to this paper).**
>
> Thank you for this comment! While we do believe our work suggests directions that may be worth pursuing for practical design choices, we would like to first emphasize that the very fact we are engaging in this discussion indicates our study has offered an important practical takeaway: our results suggested that models may be undergoing a competition of algorithms when performing ICL, updating our perspective on how one should think about ICL, and also prompting the reviewer to ask how one can tame the competition. We believe this discussion could not have been had before, and hence the opening of this discourse can likely lead to useful work down the line. For example, we may find research work motivated towards design of contexts that bias the model towards an algorithm over another one in the competition.
>
> That said, we summarize what we believe are the most crucial takeaways for practice.
>
> - **Doing better evals: Effects of context scaling.** One of the crucial axes that our work explores is context-size scaling, demonstrating that a larger vs. shorter context results in very different algorithms succeeding the competition. Most current benchmarks primarily focus on use of 2–32 exemplars for few-shot capabilities evaluation, but we believe this may be underestimating model capabilities, e.g., by focusing on retrieval / Bayesian algorithms. There is a chance that merely scaling the context and adding more exemplars will significantly improve model performance on downstream tasks, without further training advancements. Results in this spirit have started to emerge (e.g., see [1]), but our work offers a formal model for why context scaling should be considered more seriously in capability evaluations. In fact, the sudden learning of capabilities via context-scaling, as has been shown in some recent work (e.g., see [2, 3, 4]) may even be explainable by our competition of algorithms picture!
>
>
> - **Linear combination of algorithms may offer a tool for OOD performance prediction.** Assuming one can identify what algorithms are engaged in the competition to dictate model behavior during ICL, our results show that it is likely possible to predict model performance on OOD samples by merely using in-distribution inputs! We do not claim this finding will necessarily be of relevance to LLMs, where the pretraining data is too vast in its scope, but Transformers and their ability to perform ICL are nowadays used as backbones for several applications. For example, in the application of in-context RL [5, 6], one hopes to in-context learn a world model and then have the Transformer engage in goal-directed behavior on novel environments. We believe such tasks are sufficiently simple to speculate what policies (algorithms) the model may be following, decompose its behavior across them, and then predict behavior for novel environments!
>
>
> [1] https://arxiv.org/abs/2403.05530
>
> [2] https://www.anthropic.com/research/many-shot-jailbreaking
>
> [3] https://arxiv.org/abs/2310.17639
>
> [4] https://openreview.net/forum?id=pXlmOmlHJZ
>
> [5] https://arxiv.org/abs/2306.14892
>
> [6] https://arxiv.org/abs/2210.14215
>
>
> ---
>
> **[Continued below...]**

---

> ### Author Response · Authors · 2024-11-28
> **Rebuttals (2/3)**
>
> > **I think I want some kind of control condition where you use 3 or 4 different “silly” algorithms just to confirm that these kinds of phase transitions aren’t some artifact of fitting some linear combination of algorithms and that, if your algorithms are not related to the task, you don’t always see these interesting algorithm phase transitions.**
>
> This is a great suggestion! Following your comments, we have now added Fig. 40, 41 in Appendix H, where we run the following two experiments. These results provide validation to both: (i) the algorithms we claim to underlie our phase diagram, and (ii) the methodology of linearly interpolating these algorithms’ predictions to describe model behavior to identify which phase the model is in when using a given experimental configuration.
>
> - **Experiment 1:** We define four relatively ad-hoc algorithms and try to decompose model next-token probabilities across them using LCA (now renamed to LIA–linear interpolation of algorithms). Results, shown in Fig. 40, demonstrate that there are *no* interesting transitions when using the four ad-hoc algorithms for LIA! The largest weight for each algorithm simply remains roughly constant throughout checkpoints.
>
>
> - **Experiment 2:** We also perform another version of the above experiment, where we mix the four ad-hoc algorithms to our four core algorithms identified using tests on bigram utilization and Bayes proximity (yielding a total of 8 algorithms). We then perform our analysis on these 8 algorithms simultaneously. Results are shown in Fig. 41, and we find that our four core algorithms again dictate model behavior in precisely the same regions we claimed to be their corresponding algorithmic phases. The four ad-hoc algorithms are never assigned a phase (or for that matter any mass in the interpolation) of their own.
>
> Further, we also refer the reviewer to updated Fig. 5d, and the new Fig. 35 in Appendix F.1.4 These figures show KL divergence between our model’s next token predictions and the four algorithms that we claim to underlie our phases. We find the phase boundaries very closely match the boundaries of low KL regions in these diagrams, further validating our algorithms and phase diagram!
>
> ----------------------------
> > **I think I also wanted a bit of better understanding on how well LCA predicts the performance of the transformer relative to some naive baselines just to confirm that LCM well approximate the transformer.**
>
> This is also a great idea! Following your comment, we have now run the following experiment.
>
> **Defining the naive baseline.** In general, when doing LCA (now called LIA in the paper), we optimize the weights for linear interpolation at a given checkpoint. Now, to define our baseline, we run the optimization process over all checkpoints simultaneously––that is, we try to identify weights such that a linear interpolation of our algorithms would best predict the model’s behavior when averaged over time. We note that to induce some diversity and make this baseline somewhat difficult, we do allow the other control variable in our experiments (i.e., data diversity) to vary.
>
> **Results.** See Fig. 42 in Appendix H. We find that when we fix the weights across checkpoints, the error on the fit KL divergence increases substantially. Thus, we demonstrate that the “basis” of four solutions are indeed well modeling the transformer’s implemented solution.
>
> ---
> ---
>
> **[Continued below...]**

---

> ### Author Response · Authors · 2024-11-28
> **Rebuttals (3/3)**
>
> ### Questions
> > **This paper identifies a number of these interesting phenomena, but it would be nice if the authors could discuss a bit more about why these phase transitions occur. ...**
>
> Great question! Based on our current understanding of the learning dynamics, we believe the following picture is likely at play under these transitions.
>
> - **Transition with training time.** When the model begins to utilize bigram statistics of the context, we see a rather sharp drop in KL for bigram-based algorithms. Interestingly, this sharp transition correlates with the form of a precise attention head structure: specifically, the induction head. Please see the newly added Fig. 19, 20, 21 for a mechanistic analysis of the attention layers supporting this point. This structure allows the model to compute bigram statistics, and hence enables the learning of solutions that can achieve a better loss (specifically, a Bayesian solution that uses bigram statistics). We also note that this induction head formation takes place at generally the same number of training iterations in our phase diagrams, i.e., this transition is likely optimization-limited (with a weak dependence on data diversity, which must just be higher than some critical value). We note we have also added this discussion to Appendix J.
>
>
> - **Transition with data diversity.** As we increase data diversity, we find the model behavior is explained by an algorithm that involves inferring the transition matrix from context, instead of relying on chains seen during training (we now call this algorithm Bi-Inf; its earlier name was Bi-ICL). Our newly added mechanistic analysis provides hints for what is going on in this transition. Specifically, we find that at low data-diversity values, the model memorizes the transition matrices of Markov chains seen during training in its MLP neurons (see Appendix E.1). Then, as we increase data diversity, we find the model starts to represent these transitions across several neurons (i.e., in superposition). While this superposition works well at medium data-diversity values, at very high values, it is difficult for the model to achieve it quickly; i.e., it can likely still memorize all transition matrices, but it can take a long time to do so. Meanwhile, as the model goes through training, the ability to utilize bigram statistics kicks in. Since the model has still not memorized the transition matrices well by this point, this renders the Bi-Inf (i.e., what used to be called Bi-ICL) algorithm the best solution to the task! However, with further training, the Bi-Ret algorithm (i.e., what used to be called Bi-Bayes) takes over since it finally achieves a lower loss on the training set. Intriguingly, we find that we can again start to get decent reconstruction of the Markov chains in this phase! That is, this cross-over of Bi-Inf to Bi-Ret is driven by a persistent pressure to memorize the training distribution.
>
> We hope the above summary is helpful, and we are happy to expand further if needed!
>
> ----------------------------
> ----------------------------
>
> **Summary:** We thank the reviewer for their detailed feedback, which has helped us improve the rigor of our linear interpolation of algorithms (LIA) analysis by adding several interesting baselines, and also helped us contextualize the practical value of our work. We also appreciate the nudge to better justify the algorithmic phases seen in our results. We hope the responses above address the reviewer’s concerns, and, if so, they would consider championing our paper during the discussion period!

---

> > ### Comment · Reviewer_tKrP · 2024-11-29
> > **I will continue to champion and support this paper**
> >
> > Thanks for thoroughly addressing my comments!
> >
> > I'll maintain my positive score and strongly advocate for its acceptance to ICLR.

---

### Official Review · Reviewer_Pv6r · 2024-11-04

**Soundness:** 3
**Presentation:** 3
**Contribution:** 2
**Rating:** 6
**Confidence:** 3

**Summary:**

This paper investigates the mechanisms of ICL under a unified framework. The authors developed a synthetic sequence modeling task using finite mixtures of MCs and studied various phenomena in this setting, including the task retrieval vs. learning dichotomy and emergence of induction heads. They were able to simultaneously reproduce most of these well-known results. Additionally, the authors proposed that ICL is best understood as a combination of different algorithms rather than a single capability -- model configurations and data diversity impact transitions between these algorithms. This shows that ICL behavior can shift based on training data properties.

**Strengths:**

1. Very detailed and sound study of the ICL mechanism: varying task diversity, training steps, context length, and evaluating on various metrics. This is a very empirically rigorous study that verifies previous literature by reproducing various well-known results within a single setting.

2. Some analysis on how model design affects downstream performance on various metrics.

**Weaknesses:**

My main concern with this paper is its novelty: as the authors have correctly noted, many of the results presented here have already been explored in existing literature. While this paper offers a valuable unifying study that synthesizes and reproduces previous findings in a single framework, the setting itself has also been examined in prior work (e.g., Edelman et al.). Consequently, the overall message lacks new insights.

**Questions:**

The Bayesian vs. Non-Bayesian comparison: Generally, I think of Bayesian learning as a form of "learning to learn," where a posterior is updated based on observed data according to Bayes' rule, with a prior typically established during pre-training. Therefore, it is unclear why the comparison between ID and OOD performance specifically reflects a shift between Bayesian and non-Bayesian paradigms. In particular, the analogy between "task-retrieval" vs. "task-learning" and Bayesian vs. non-Bayesian, in my view, does not fully capture the essence of Bayesian inference.

In a Bayesian approach, one would ideally follow Bayes' rule to update an implicit or explicit posterior or posterior predictive distribution. Given this, shouldn’t the KL divergence be examined between the model prediction and the true posterior or posterior predictive distribution (according to Bayes' rule and the correct prior) at each context length (so given the same context of length $l$, examine at each step from $1,\dots, l$)?

I understand that you may be viewing task-retrieval vs. task-learning as a contrast between in-weight learning (IWL) and meta-learning. However, this distinction could also be seen as reflecting a finite, discrete prior versus a continuous or uniform prior. The true Bayesian vs. non-Bayesian distinction, perhaps, is whether the model has learned to correctly update a posterior (i.e., learned Bayes' rule) and has fitted an accurate prior.

---

> ### Author Response · Authors · 2024-11-28
> **Rebuttals (1/4)**
>
> We thank the reviewer for their thorough feedback! We are glad they found our study "very detailed and sound", "very empirically rigorous" and offering “a valuable unifying study that synthesizes and reproduces previous findings in a single framework”. We also appreciate the feedback for better clarifying our novel contributions, and we have made substantial improvements in our updated manuscript to address this.
>
> We have made significant changes to our manuscript:
> - A major revision of the writing, revising some terminologies, improving the flow and highlighting contributions.
> - A large set of mechanistic experiments to confirm the existence of the algorithms we delineate. (Fig. 15-26)
> - Additional quantifications of experimental results to verify our findings. (Fig. 5d, 35, 38-42)
> - Additional visualizations of results for completeness. (Fig. 32-34)
>
> Below, we respond to specific comments by the reviewer.
>
> ---
> > **My main concern with this paper is its novelty: ... this paper offers a valuable unifying study that synthesizes and reproduces previous findings in a single framework, the setting itself has also been examined in prior work (e.g., Edelman et al.)....**
>
>
> Thank you so much for raising this important point for clarification! We'd first like to emphasize that while reproducing ICL phenomena in a singular model system was a crucial step for validating our framework, our ultimate goal was to harness this system’s simplicity and richness to **analyze the mechanisms underlying ICL’s phenomenology and characterize their precise implementations in transformers**. Below, we clarify three key novel contributions of our work.
> - **Unified setup for studying ICL.** Our primary motivation for reproducing ICL’s phenomenology in a singular setup was to demonstrate the remarkable richness of our proposed task. Despite its simplicity, our task captures most (if not all) known phenomenology of ICL, providing a first step towards a unified understanding of the concept. *This is precisely why we spent only half a page on those experiments, instead devoting most of our paper to **our novel analysis** of linear interpolation of different ICL algorithms.* While Edelman et al. studied a setup similar to ours, i.e., learning to simulate Markov chains, their work involves learning with *infinite* chains. Our strategic choice of finite mixtures enabled us to capture an array of crucial ICL phenomena, including data diversity effects, transient nature of ICL, model width scaling effects, and several others listed in Appendix C. We emphasize these results *could not have* been shown in Edelman et al.’s setup, restricting the generality of their claims on how ICL works.
> - **Novel analysis of linear interpolation of different ICL algorithms.** Our thorough grounding of the system via reproduction of phenomenology was not an end in itself, but rather a crucial step toward deeper mechanistic insights. The interpretability gained through simplification led us to identify four distinct algorithmic phases with linear interpolation of different ICL algorithms. ***To the best of our knowledge, this analysis has never been attempted before!*** With this approach, we characterized the transitions between these algorithmic phases as a function of data diversity, optimization steps, and context size. This interpretability offered us a novel framework of "competitive dynamics of algorithms", revealing how different algorithmic strategies are promoted or suppressed by varying factors of data diversity, optimization steps, and context size. Through this lens, we have been able to not just reproduce, but also refine our understanding of six distinct phenomena of ICL previously reported in the literature. For example, *we can now explain the origin of the non-monotonic nature of ICL as a process where a certain algorithm gets promoted during training, but then suppressed by another algorithm* driven by the training loss. We can also explain the impact of model width scaling, tokenization, and other architectural design choices, and we in fact demonstrate that *there was a confounding factor in the effect of model width scaling in prior work by Kirsch et al.!* (See Appendix C.2.2 and C.2.3 for discussion on the latter.)
>
>
> **[Continued below...]**

---

> ### Author Response · Authors · 2024-11-28
> **Rebuttals (2/4)**
>
> - **Analyzing implementation of behavioral algorithms (New Appendix E).** To further address your feedback on how to best leverage our system for extracting novel insights, we have added a comprehensive new section with four key analyses (See Appendix E.1-E.4). First, we successfully reconstruct training transition matrices from MLP weights in low data diversity settings, demonstrating a direct neuronal implementation of retrieval solutions (Appendix E.1). Second, we track neuron memorization dynamics by measuring KL divergence between neuron outputs and transitions, revealing systematic changes in memorization that align precisely with our algorithmic phase transitions (Appendix E.2). Third, we analyze attention maps to demonstrate distinct architectural signatures for each algorithm - uniform attention for Uni-Inf, induction head patterns for Bi-Inf, and patterns for Bi-Ret (Appendix E.3) combines information from two subsequent tokens, in line with our behavioral observations. Finally, we validate our Linear Interpolation of Algorithms (LIA) framework by showing that attention patterns at phase boundaries smoothly interpolate between adjacent algorithms (Appendix E.4). Together, these analyses provide mechanistic evidence for how identified behavioral algorithms are implemented and compete, providing a mechanistic foundation and novel insights!
>
> We thank you for your valuable feedback, which prompted us to thoroughly reconsider and enhance how we present our work's goals, approach, and insights. We have expanded our discussion of specific contributions in Appendix C, newly created Appendix E, and revised the manuscript to better emphasize how our empirically grounded system advances beyond simple reproduction. The revised manuscript now offers refined insights into the algorithmic nature of In-Context Learning, including new results that analyze how transformers implement these algorithms through our minimal, yet rich experimental setup.
>
>
> ---
>
>
> **[Continued below...]**

---

> ### Author Response · Authors · 2024-11-28
> **Rebuttals (3/4)**
>
> ### Questions
> ---
> > **The Bayesian vs. Non-Bayesian comparison: Generally, I think of Bayesian learning as a form of "learning to learn," where a posterior is updated based on observed data according to Bayes' rule, with a prior typically established during pre-training. Therefore, it is unclear why the comparison between ID and OOD performance specifically reflects a shift between Bayesian and non-Bayesian paradigms...**
>
>
> Thank you for raising this point! We first emphasize that we used the terms Bayesian vs. non-Bayesian primarily to be consistent with prior work by Raventos et al., who deem these two approaches to be different in nature. While we had mentioned this motivation in Footnote 3 in the paper, i.e., that both approaches can in fact be understood as Bayesian inference with different forms of prior, we realize now that this terminology actually induced confusion and the footnote did not adequately address the issue. To this end, we have now significantly improved the paper’s writing to address this ambiguity. Specifically, we made the following updates.
>
> - **Update names to remove the Bayesian vs. Non-Bayesian dichotomy.** We have now updated the names of our algorithms to remove any ambiguity that one algorithm is Bayesian and another is not. Specifically, we call the two approaches “fuzzy retrieval” versus “inference” of the relevant transition matrices. We have updated the discussion around these terms to emphasize, however, that they are both Bayesian inference approaches under the hood (as discussed below).
>
>
> - **Updated Section 3 to emphasize that both approaches involve a Bayesian averaging operation.** We have updated Section 3’s writing to strongly emphasize in the main text itself that, in fact, both the retrieval and inference approaches involve a Bayesian averaging operation: the precise prior involved in this operation is quite different though, limiting the generalization abilities of one family of algorithms over another. Specifically, in the fuzzy retrieval approaches, the prior covers merely the Markov chains seen during training, while in the latter it is a relaxed prior that uniformly covers the entire Dirichlet distribution. This affects their ID and OOD generalization abilities, as discussed next.
>
>
> - **Improving the discussion for why Retrieval approaches underperform on OOD inputs.** As the reviewer correctly points out, in Bayesian inference the posterior is indeed updated based on observed data (i.e., at inference time) and the prior over training chains is established during pre-training. However, our results show that the form of these priors can be drastically different depending on the data diversity in the experimental setup: this can be seen, e.g., in Eq. 2 and 3 where the likelihood calculation (and thus the posterior calculation) requires knowledge over the training set matrices. The model is still "learning to learn" in the sense that it is trying to ascertain which seen Markov chains best explain the observed data (i.e., the context), and using a weighted average of those chains to continue the sequence (hence performing “learning”). However, the two inference solutions (i.e., what used to be called “non-Bayesian solutions”) in Eqs. 5, 6 do not involve any information about the training set, and thus will have no performance gap between ID and OOD. That is, since their prior covers the entire training distribution (i.e., the Dirichlet prior), they guarantee the same performance on both ID and OOD data. We have updated Section 3.1 to emphasize the above.
>
>
> - **Mechanistic evidence to justify poor OOD performance of retrieval algorithms.** While the low (and at times literally zero) KL to our predefined algorithms indicates the model is behaviorally equivalent to them, we note that we have now provided crucial mechanistic evidence to justify these claims. Specifically, in Appendix E, we now demonstrate that we can literally reconstruct transition matrices from MLP neurons when the model is engaging in behavior similar to retrieval algorithms (see App. E.1)! We show that this does not work when the model is in “inference” phases, i.e., is relying on a relaxed prior.
>
> We hope the above changes help address the reviewer’s concerns, and please let us know if there are any further clarifications that can be of help!
>
> ---
>
> **[Continued below...]**

---

> ### Author Response · Authors · 2024-11-28
> **Rebuttals (4/4)**
>
> > **In a Bayesian approach ... Given this, shouldn’t the KL divergence be examined between the model prediction and the true posterior or posterior predictive distribution (according to Bayes' rule and the correct prior) at each context length...?**
>
> Thank you for this suggestion! We certainly agree that behavioral equivalence between our model and predefined algorithms will be better demonstrated by computing the KL between model predictions and algorithms’ posterior predictive distribution. To this end, we have now made the following updates to the paper.
>
> - **KL between model and algorithms in Figure 5d.** We have now updated Figure 5 by adding a panel (d) that shows the KL between our model’s predictions and the algorithms’ posterior predictive distributions. We find a clear validation of our claims: specifically, we see the identified algorithmic phases for different experimental configurations (Figs. 5a–5c) perfectly match the regions of low KL with respect to the corresponding algorithms in Fig. 5d!
>
>
> - **KL with context-scaling.** We have now added Figure 34 where we chose 4 checkpoints in the phase space where the model is expected to be dominated by one algorithmic solution and show the model and solution’s expected KL divergence as we change the input context length. We find a clean verification of our claims: the model prediction robustly follows the KL expected from each algorithmic solution for different context sizes, both for ID chains and OOD chains. This figure is actually very convincing, thank you very much for suggesting this!
>
> We thank the reviewer again for suggesting this experiment!
>
> ---
> > **I understand that you may be viewing task-retrieval vs. task-learning ... this distinction could also be seen as reflecting a finite, discrete prior versus a continuous or uniform prior. ...**
>
> As we note in our previous answers, we used terms that indicate a distinction between task-retrieval vs. task-learning primarily to stay consistent with prior work (especially the paper by Raventos et al.). We had a footnote in the paper explaining, however, that we do not believe this dichotomy is real. To repeat, we had written: “We refer to these solutions as non-Bayesian, in line with Ravento ́s et al. (2023), while acknowledging that they can be interpreted as a form of Bayesian inference with a relaxed prior (see App. F.1).” We then clarified in the referenced appendix that in fact the non-Bayesian solutions have a continuous or uniform prior, and the Bayesian ones had a discrete prior over chains seen during training.
>
> Overall, we believe that this consistency with prior work has not been useful and caused confusion. To this end, we have now updated the names of our algorithms and emphasized in Section 3 that, in fact, they can both be deemed as Bayesian inference with different forms of priors: one over just the seen training chains, and one a continuous prior over the entire data distribution.
>
>
> ---
> ---
>
> **Summary:** We thank the reviewer for their detailed feedback that has motivated us to contextualize our work with respect to prior results, further emphasizing its novelty. We are also grateful for reviewer’s comments to better clarify the distinction between families of algorithms considered in this work, helping us significantly update Section 3 and add several new experiments mechanistically grounding these algorithms in Appendix E. We hope our responses have addressed the reviewer’s concerns, and, if so, they would consider raising their score to support the acceptance of our work!

---

> > ### Comment · Reviewer_Pv6r · 2024-11-28
> >
> > Dear authors,
> >
> > I have read the rebuttal and thank the authors for their detailed responses. I have raised my score.

---

### Official Review · Reviewer_vTGM · 2024-11-09

**Soundness:** 4
**Presentation:** 4
**Contribution:** 2
**Rating:** 6
**Confidence:** 4

**Summary:**

This paper introduces a new synthetic task, namely finite Markov mixtures, to study the properties of in-context learning (ICL) in Transformers. The authors present several analyses on top of finite Markov mixtures. First they show that this task alone can effectively recover known phenomena which were originally discovered on several different tasks like linear regression, classification, and finite automata. Then they contribute new insights, and argue that ICL is best thought of as a convex combination of several learning algorithms which emerge or vanish during training.

Overall I found this work a well-written and interesting read, with several novel observations which are very relevant to the ICLR community. In particular, I appreciated the insight that ICL is best thought of as a mixture of learning algorithms, and that these algorithms are transient throughout training. It’s main weakness is that It lacks a clear takeaway message for readers to act upon — see my questions below.

**Strengths:**

- **Well-written**. This expository work is well written and enjoyable to read. The authors do a good job at holding the reader by the hand through their analyses and provide adequate context even for non-ICL experts. The figure are easily understandable (mostly, see below for a nit) and the flow of the paper makes sense.
- **Sound, novel, and unifying benchmark.** The benchmark is novel in the context of ICL studies and it is soundly derived (eg, using the expected KL as quality measure in Eq. 1). Some of the choices are arguable (eg, why Dirichlet?) but that’s splitting hair. What’s more interesting is that it reproduces known phenomena from the literature while also uncovering novel insights.
- **Strong analysis revealing new insights**. The insight that ICL can be decomposed into several algorithms *at any point in training* is novel and deserves further thinking. Some may argue that the transient nature of ICL was known (the authors say so themselves) but this work goes one step further, showing the algorithms are not forgotten but rather superposed with different weighting coefficients (Fig. 6.b).

**Weaknesses:**

- **Lack of actionable takeaway message**. Because the benchmark is synthetic, it’s unclear how much it says about ICL for real-world LLMs. This is the main flaw of the paper. For example, l.81: it’s good to know that Transformers can learn different algorithms — but which is it for LLMs / VLMs? Certainly not the unigram or bigram of l. 301, right? Similarly, I liked the flavor of paragraph on l. 301 but it’s unclear how to replicate this study on real-world LLMs. This limits the insights into how real-world ICL really works, and so we’re left with the question: what do we do with the discovery that ICL is best thought of as a mixture of algorithms? This work doesn’t say which are the algorithms for real-world ICL, so we can’t, e.g., prescribe which algorithm to surface and when.
- **The Bayesian vs non-Bayesian discussion is unclear**. On l. 226 the authors argue that ICL first has a Bayesian and then a non-Bayesian behavior (they intentionally don’t use the term frequentist), and that the Bayesian overfits to the training Markov mixtures. I think this distinction could be explained better. For example, one could make the argument that ICL is always Bayesian — otherwise what is transferred? — but the prior weakens as the model sees more diverse chains. In fact, the experiments support this line of thought: Fig. 5.b shows that as we increase the number of training chains, the prior weakens. In fact, I think believe this weakening of the prior is a property of how the Markov mixtures are constructed, and we would see a different behavior if the test chains where chosen differently. Note that this point  doesn’t take away from the superposition analysis but it needs clearer explanations.
- **Minor presentation issues**. The paragraph on l. 341 is not detailed enough, and doesn’t explain what the Fig. 5 shows. It also doesn’t help that Fig. 5 doesn’t have clear legend on the heat map scales — is darker more Bayesian or less Bayesian for 5.b? Finally, some parameter choices in the analysis are weird: l. 187, why plot at train step 839 specifically? l. 192, why at n = 2^7? Can we have similar figures for later training steps and for larger n’s in the Appendix?

**Questions:**

See my questions in the weaknesses section above.

---

> ### Author Response · Authors · 2024-11-28
> **Rebuttals (1/4)**
>
> We thank the reviewer for their thorough review and feedback! We are glad that they found our paper “well-written", our experiments “sound, novel and unifying”, and our analysis “strong; revealing new insights”! We also appreciated reviewer’s emphasis on what we believe to be our main contributions: 1) unification of ICL phenomenology with a novel setup and 2) downstream insights from algorithmic decomposition.
>
> We have made significant changes to our manuscript:
> - A major revision of the writing, revising some terminologies, improving the flow and highlighting contributions.
> - A large set of mechanistic experiments to confirm the existence of the algorithms we delineate. (Fig. 15-26)
> - Additional quantifications of experimental results to verify our findings. (Fig. 5d, 35, 38-42)
> - Additional visualizations of results for completeness. (Fig. 32-34)
>
> Below, we respond to specific comments by the reviewer. Please note that we changed the names of algorithms from Uni-Bayes, Bi-Bayes, Uni-ICL, Bi-ICL to respectively Uni-Ret, Bi-Ret, Uni-Inf, Bi-Inf in the manuscript. However, in our responses below we will continue using the *original names* to ensure ease of readability.
>
> ---
> ---
>
> >  **(eg, why Dirichlet?)**
>
> Great question! We note that the Dirichlet distribution is a commonly used prior for categorical or Multinomial variables in Bayesian inference. Given we are working with discrete state spaces, a Dirichlet prior thus makes intuitive sense for our problem setting as well. This is also likely the reason for the prior’s common use in recent work exploring Transformers trained on Markov chains [1, 2]. That said, we emphasize we did *not* see any noticeable effects in our preliminary experiments with a uniform prior and a Poisson prior. Hence, to stay consistent with prior work, we preferred to use a Dirichlet prior in our experiments.
>
> [1] https://arxiv.org/abs/2402.11004
>
> [2] https://arxiv.org/abs/2407.17686v1
>
> ---
> **[Continued below...]**

---

> ### Author Response · Authors · 2024-11-28
> **Rebuttals (2/4)**
>
> > **Lack of actionable takeaway message. Because the benchmark is synthetic, it’s unclear how much it says about ICL for real-world LLMs. ... I liked the flavor of paragraph on l. 301 but it’s unclear how to replicate this study on real-world LLMs. ... What do we do with the discovery that ICL is best thought of as a mixture of algorithms?**
>
> Thank you for this comment! While we do believe our work suggests directions that may be worth pursuing from a practical standpoint, we would like to first emphasize that the very fact we are engaging in this discussion indicates our study has offered an important takeaway: our results suggested that models may be undergoing a competition of algorithms when performing ICL, updating our perspective on how one should think about ICL, and also prompting the reviewer to ask how one can tame the competition! We believe this discussion could not have been had before, and hence the opening of this discourse can likely lead to useful work down the line. For example, we may find research work motivated towards design of contexts that bias the model towards an algorithm over another one in the competition.
>
> That said, we summarize what we believe are the most crucial and actionable takeaways of our results:
>
>
> - **Doing better evals: Effects of context scaling.** One of the crucial axes that our work explores is context-size scaling, demonstrating that a larger vs. shorter context results in very different algorithms succeeding the competition. Most current benchmarks primarily focus on use of 2–32 exemplars for few-shot capabilities evaluation, but we believe this may be underestimating model capabilities, e.g., by focusing on retrieval / Bayesian algorithms. There is a chance that merely scaling the context and adding more exemplars will significantly improve model performance on downstream tasks, without further training advancements. Results in this spirit have started to emerge (e.g., see [1]), but our work offers a formal model for why context scaling should be considered more seriously in capability evaluations. In fact, the sudden learning of capabilities via context-scaling, as has been shown in some recent work (e.g., see [2, 3, 4]) may even be explainable by our competition of algorithms picture!
>
>
> - **Linear combination of algorithms may offer a tool for OOD performance prediction.** Assuming one can identify what algorithms are engaged in the competition to dictate model behavior during ICL, our results show that it is likely possible to predict model performance on OOD samples by merely using in-distribution inputs! We do not claim this finding will necessarily be of relevance to LLMs, where the pretraining data is too vast in its scope, but Transformers and their ability to perform ICL are nowadays used as backbones for several applications. For example, in the application of in-context RL  [5, 6] , one hopes to in-context learn a world model and then have the Transformer engage in goal-directed behavior on novel environments. We believe such tasks are sufficiently simple to speculate what policies (algorithms) the model may be following, decompose its behavior across them, and then predict behavior for novel environments!
>
>
> [1] https://arxiv.org/abs/2403.05530
>
> [2] https://www.anthropic.com/research/many-shot-jailbreaking
>
> [3] https://arxiv.org/abs/2310.17639
>
> [4] https://openreview.net/forum?id=pXlmOmlHJZ
>
> [5] https://arxiv.org/abs/2306.14892
>
> [6] https://arxiv.org/abs/2210.14215
>
>
> ---
> **[Continued below...]**

---

> ### Author Response · Authors · 2024-11-28
> **Rebuttals (3/4)**
>
> > **The Bayesian vs non-Bayesian discussion is unclear.**
>
> Thank you for raising this point! The distinction between these two sets of approaches (Bayesian vs. non-Bayesian) is subtle and indeed worthwhile being clear over. To this end, we have now significantly updated Section 3, highlighting how one set of algorithms involves Bayesian averaging with a prior defined over the set of Markov chains seen during training (we now call this set the *fuzzy retrieval* algorithms), while another set of algorithms involves Bayesian averaging with a relaxed prior, i.e., the Dirichlet distribution (we now call this set the *inference* algorithms). We note we have also improved the discussion in Appendix B.1, where we discuss in more detail how these two sets of approaches are related.
>
> In regards to the effect of choosing test chains differently, we highlight our preliminary results in Appendix G on test chains that do not come from a Dirichlet prior: we find that models in the Bi-ICL phase in fact generalize perfectly well to an entirely OOD chain!
>
>
> ---
> ---
> **[Continued below...]**

---

> ### Author Response · Authors · 2024-11-28
> **Rebuttals (4/4)**
>
> **Minor comments**
>
> > **The paragraph on l. 341 is not detailed enough, and doesn’t explain what the Fig. 5 shows. It also doesn’t help that Fig. 5 doesn’t have clear legend on the heat map scales — is darker more Bayesian or less Bayesian for 5.b?**
>
> Fair point! To address your comments, we have now made the following updates to Fig. 5 and the corresponding paragraph.
>
> - Figure 5: We now clearly mark the meaning of individual colorbars to help identify what a brighter or darker color means for a given heatmap. We have also added another panel to this figure, wherein we demonstrate that our identified algorithmic phases in panel (a)–(c) yield very low KL from the model’s next-token probabilities. This helps further ground our analysis.
> - Paragraph corresponding to L.341 (now found at L.345): We have expanded the discussion in this paragraph substantially, discussing first the algorithmic phases identified using our evaluations, then the impact of different control axes (data diversity and training) that drive transitions between the different algorithmic phases, and then emphasizing several other experiments that corroborate this analysis. These additional experiments are available in Appendix E,F Fig 15~35.
>
> ---
>
> > **Some parameter choices in the analysis are weird: l. 187, why plot at train step 839 specifically? l. 192, why at n = 2^7? Can we have similar figures for later training steps and for larger n’s in the Appendix?**
>
>
> We apologize for any confusion caused by missing details! To address your comments, we have now added notes on why we use specific values for the main results, and added plots ablating those values to show our claims remain consistent regardless of the precise value chosen.
>
> - **Why step 839:** Since our goal at L.187 (and hence Fig. 3.b) was to demonstrate the data diversity threshold needed for OOD generalization similar to Raventos et al. 2023, we chose the checkpoint at step=839 as the threshold is most easy to visualize there. This is because step=839 is a checkpoint where we can observe the data diversity dependent transition without confounding factors from the transient nature of ICL. However, we note that the plots look qualitatively similar for checkpoints from step 600 to 2200. To this end, we have now added plots for checkpoints from these ranges in App. F.1.3 Fig 32.
> - **Why n=2^7:** (the manuscript now uses $N$ instead of $n$) Similar to above, we chose $𝑛=2^7$ as the data diversity to plot since that value best illustrates the transient nature of ICL. Again, plotting for $𝑛=2^6$ or $𝑛=2^8$ also yield us the same qualitative effect, but the the Bi-ICL phase is respectively too short and too long for these values, hence not yielding the clearest visualization of transience. However, to emphasize that our results are robust, we have now added additional plots in App. F.1.3 Fig. 33 for different values of $n$ and noted in the main paper that results for alternative settings can be found in the appendix.
>
> ---
> ---
> **Summary:** We thank the reviewer for their detailed feedback, which has helped us significantly improve the clarity of our writing and helped us demonstrate the robustness of our claims via addition of several new plots that ablate control variables! We hope our responses address the reviewer’s concerns and, if so, they would consider championing our paper during the discussion period.

---

> ### Author Response · Authors · 2024-12-01
> **Summary of updates before discussion deadline**
>
> Dear Reviewer vTGM,
>
> Thank you for your thoughtful and insightful feedback which has helped us significantly improve our manuscript. As the extended discussion period ends tomorrow, we wanted to follow up on our response and the substantial updates we've made to address your concerns.
>
> We recognize that we have made numerous changes which may take time to digest. Thus, we would like to concisely summarize the key updates addressing your main points:
>
> 1. **Regarding the "lack of actionable takeaway message"**:
>    - We have added extensive mechanistic experiments **(new Fig. 15-26)** that validate the existence of the algorithms we identified
>    - New quantitative results (new Fig. 5d, 35, 38-42) provide concrete evidence for the importance of context-scaling effects and potential for OOD performance prediction
>
> 2. **On the "Bayesian vs non-Bayesian discussion"**:
>    - We have significantly revised Section 3 to clarify the distinction between fuzzy retrieval algorithms and inference algorithms, updating these terminologies throughout the paper
>    - Added preliminary results in Appendix G demonstrating perfect generalization to OOD chains in the Bi-ICL (now called Bigram Inference) phase
>
> 3. **Addressing presentation concerns**:
>    - Added clear colorbar annotations to Fig. 5 and expanded the corresponding discussion
>    - Included comprehensive ablation studies (new Fig. 32-34) showing robustness across different parameter choices
>    - Added extensive explanations in Appendix E,F justifying our parameter selections
>    - Demonstrated the consistency of our findings across different model architectures and sizes (new Fig. 38-42)
>
> We would greatly appreciate if you could review these updates and consider updating your score if you find that we have adequately addressed your concerns. If you have any remaining questions or would like clarification on any point, we are happy to respond promptly.

---

### Meta-Review · Area_Chair_Sc5Q · 2024-12-24

**Metareview:**

This paper provides a new perspective of understanding in-context learning (ICL). By synthetically constructing sequence modeling tasks that simulate a finite mixture of Markov chains, the authors were able to train transformers on the simulated data and reproduce phenomena of ICL. One nice insight is that ICL can be thought of as a combination of algorithms, and their competition determines the behavior of ICL for different settings of model, data and context.

The reviewers found the work well-written, sound, comprehensive and insightful. During rebuttal, the authors addressed many concerns the reviewers had around clarity, for example about the Bayesian vs non-Bayesian discussion. The downsides the reviewers pointed out including novelty, indications to real-world models and ICL tasks, and a lack of actionable items beyond the new insights on how ICL works for synthetic problems. The authors made clarifications on those and it'd be great to systematically write these up and integrate into the paper.

All reviewers suggested acceptance and I concur.

**Additional Comments On Reviewer Discussion:**

see above: The reviewers found the work well-written, sound, comprehensive and insightful. During rebuttal, the authors addressed many concerns the reviewers had around clarity, for example about the Bayesian vs non-Bayesian discussion. The downsides the reviewers pointed out including novelty, indications to real-world models and ICL tasks, and a lack of actionable items beyond the new insights on how ICL works for synthetic problems. The authors made clarifications on those and it'd be great to systematically write these up and integrate into the paper.

All reviewers suggested acceptance and I concur.

---

### Decision · Program_Chairs · 2025-01-22

Accept (Spotlight)